# Momentum-Driven Adaptivity: Towards Tuning-Free Asynchronous Federated Learning

Wenjing Yan [1]  Xiangyu Zhong [1]  Xiaolu Wang [2]  Ying-Jun Angela Zhang [1]

## Abstract

Asynchronous federated learning (AFL) has emerged as a promising solution to address system heterogeneity and improve the training efficiency of federated learning. However, existing AFL methods face two critical limitations: 1) they rely on strong assumptions about bounded data heterogeneity across clients, and 2) they require meticulous tuning of learning rates based on unknown system parameters. In this paper, we tackle these challenges by leveraging momentum-based optimization and adaptive learning strategies. We first propose **MasFL**, a novel momentum-driven AFL framework that successfully eliminates the need for data heterogeneity bounds by effectively utilizing historical descent directions across clients and iterations. By mitigating the staleness accumulation caused by asynchronous updates, we prove that MasFL achieves state-of-the-art convergence rates with linear speedup in both the number of participating clients and local updates. Building on this foundation, we further introduce **AdaMasFL**, an adaptive variant that incorporates gradient normalization into local updates. Remarkably, this integration removes all dependencies on problem-specific parameters, yielding a fully problem-parameter-free AFL approach while retaining theoretical guarantees. Extensive experiments demonstrate that AdaMasFL consistently outperforms state-of-the-art AFL methods in runtime efficiency and exhibits exceptional robustness across diverse learning rate configurations and system conditions.

[1]Department of Information Engineering, The Chinese University of Hong Kong, Hong Kong SAR; [2]Software Engineering Institute, East China Normal University, China. Correspondence to: Xiangyu Zhong <xyzhong@ie.cuhk.edu.hk>, Xiaolu Wang <xiaoluwang@sei.ecnu.edu.cn>.

*Proceedings of the $42^{st}$ International Conference on Machine Learning*, Vancouver, Canada. PMLR 267, 2025. Copyright 2025 by the author(s).

## 1. Introduction

Recent years have witnessed an explosive growth in edge computing and mobile applications, leading to massive amounts of data being generated on distributed devices. Federated learning (FL) has emerged as a transformative paradigm that enables model training across these decentralized data sources while preserving privacy and addressing data sovereignty concerns (McMahan et al., 2017; Kairouz et al., 2021; Li et al., 2020). By keeping data locally and sharing only model updates, FL has become increasingly crucial in numerous applications, from mobile services to smart cities (Abreha et al., 2022; Pandya et al., 2023).

However, the practical deployment of FL faces significant challenges due to system heterogeneity among participating clients. Devices vary substantially in their computing capabilities, network conditions, and availability patterns (Assran et al., 2020; Zhong et al., 2022). Traditional synchronous FL protocols, such as the well-known FedAvg (McMahan et al., 2017; Karimireddy et al., 2020b), which require all clients to complete their updates before proceeding to the next round, suffer from the "straggler effect" where the system's progress is bottlenecked by the slowest participants. This synchronization barrier severely impacts training efficiency and system scalability.

Asynchronous federated learning (AFL) has emerged as a promising solution to address the system heterogeneity challenge by allowing clients to communicate with the server independently of other participants (Xie et al., 2019; Chen et al., 2020; Zakerinia et al., 2022; Nguyen et al., 2022; Wang et al., 2023; Fraboni et al., 2023; Wang et al., 2024c;a; Leconte et al., 2024a;b). Despite its effectiveness in mitigating the straggler effect, existing AFL approaches face two critical challenges: 1) **Data Heterogeneity:** In AFL, faster clients contribute updates more frequently than slower ones, resulting in a biased global model dominated by the data distributions of those faster clients. This bias is exacerbated by local training, where clients perform multiple updates before communicating with the server, leading to the phenomenon of "client drift" (Karimireddy et al., 2020b). To address this issue, existing AFL methods often rely on strong assumptions of bounded data heterogeneity, restricting their applicability in real-world scenarios with highly

diverse and arbitrary client distributions. 2) **Algorithm Tuning:** The theoretical convergence of AFL algorithms relies heavily on the correct selection of hyperparameters, such as learning rates. However, in existing AFL approaches, these hyperparameters are often functions of multiple unknown parameters, including problem-specific constants (e.g., smoothness constants), algorithmic variables (e.g., initial suboptimality gap, gradient variance bounds), and system-dependent factors (e.g., client staleness). Since these parameters are typically unavailable *a priori*, tuning learning rates becomes a highly challenging task. Furthermore, the problem-specific learning rate configurations limit the adaptability of these approaches to diverse and dynamic AFL environments.

In light of the key challenges associated with AFL, it is crucial to address the following questions:

*(i) Can we design an AFL algorithm that removes the restrictive bounded heterogeneity assumption while preserving theoretical convergence guarantees?*
*(ii) Building upon such a framework, can we further eliminate the need for manual hyperparameter tuning by introducing adaptive mechanisms that automatically adjust to diverse system characteristics and data distributions?*

These questions form the foundation of our systematic effort to develop more robust and practical AFL methods.

### 1.1. Main Contributions

In this paper, we provide affirmative answers to the above two questions by exploring the effects of momentum-based optimization and adaptive learning strategies. Our main contributions are summarized as follows:

- We develop two novel training approaches for AFL: **MasFL** and **AdaMasFL**. MasFL incorporates a two-level (both client- and server-side) momentum into the AFL framework to mitigate client drift and accelerate convergence, effectively addressing the challenges posed by data heterogeneity. Building on this, AdaMasFL further integrates local gradient normalization with local momentum, enabling automatic adaptation to diverse system characteristics and data distributions without manual hyperparameter tuning.

- We establish strong theoretical guarantees for both algorithms, achieving state-of-the-art convergence rates with linear speedup with respect to the number of participating clients and local updates per round. Notably, MasFL eliminates the restrictive bounded data heterogeneity assumption commonly required in existing AFL approaches, making it applicable to scenarios with arbitrarily heterogeneous data distributions. Beyond this theoretical advancement, AdaMasFL achieves convergence without relying on any unknown system or algorithmic

parameters, eliminating the need for tedious hyperparameter tuning.

- We conduct comprehensive empirical evaluations on deep learning tasks using real-world datasets. The numerical results demonstrate that our approaches consistently outperform state-of-the-art AFL algorithms in runtime efficiency, even when their hyperparameters are well-tuned. This improvement can be attributed to the acceleration provided by our two-level momentum mechanism and the per-step adaptation of our effective stepsizes. Notably, AdaMasFL exhibits exceptional robustness across a wide range of learning rates and asynchrony levels, maintaining nearly consistent performance despite varying degrees of delays. In contrast, the performance of state-of-the-art baselines deteriorates significantly under the same conditions.

### 1.2. Related Work

Our approaches extend the buffered asynchronous aggregation mechanism introduced by FedBuff (Nguyen et al., 2022) to enhance training efficiency and mitigate the adverse effects of outdated gradients. To address the "client drift" problem caused by data heterogeneity in FedBuff, Wang et al. (2023) proposed $CA^2FL$, which maintains cached updates for each client to calibrate global aggregation. These cached updates function similarly to the control variates in our methods. However, $CA^2FL$ still relies on the bounded data heterogeneity assumption to ensure convergence. To accommodate arbitrary heterogeneous data in asynchronous learning, a recent study (Wang et al., 2024b) proposed DuDe-ASGD, which reuses the most recent gradients from all clients in each round of global aggregation. While effective, DuDe-ASGD does not account for multiple local updates. In contrast, our MasFL framework eliminates the need for data heterogeneity bounds while achieving the best-known communication efficiency, demonstrating an efficient utilization of distributed resources.

Yu et al. (2024) identified that asynchrony introduces implicit bias in momentum updates and proposed momentum approximation for AFL, which optimally weights historical model updates to approximate synchronous momentum behavior. FedAC (Zang et al., 2024) addresses both model staleness and client drift through adaptive server updates and corrective client training. Although these works demonstrate empirical improvements, their algorithms lack theoretical convergence guarantees. FADAS (Wang et al., 2024c) introduces a global adaptive AFL framework by applying adaptive optimization techniques at the server-side aggregation while supporting asynchronous client updates. However, FADAS relies on stringent assumptions on bounded data heterogeneity and bounded gradient norms. Moreover, it still requires careful hyperparameter tuning. In contrast,

our method AdaMasFL integrates adaptive stepsizes with a two-level momentum, eliminating these constraints while maintaining theoretical convergence guarantees without problem-specific hyperparameter tuning. Recent progress in parameter-free FL algorithms has been marked by the development of PAdaMFed (Yan et al., 2025), a pioneering algorithm that eliminates the need for manual tuning of problem-specific parameters. While PAdaMFed represents a significant breakthrough, it is designed for the synchronous setting and cannot be directly extended to address the AFL problem studied in this work. A comprehensive literature review is presented in Appendix A.

## 2. Problem Setup

Consider a distributed learning system consisting of $N$ clients (edge devices) jointly optimizing a model parameterized by $\boldsymbol{\theta} \in \mathbb{R}^d$ under the coordination of a central server. Each client $i$ maintains a local dataset characterized by a distribution $\mathcal{D}_i$, from which samples $\boldsymbol{\xi}_i$ are drawn. The local objective at client $i$ is defined as the expected loss: $f_i(\boldsymbol{\theta}) := \mathbb{E}_{\boldsymbol{\xi}_i \sim \mathcal{D}_i}[F(\boldsymbol{\theta}; \boldsymbol{\xi}_i)]$. The global optimization problem is formulated as:

$$\min_{\boldsymbol{\theta} \in \mathbb{R}^d} f(\boldsymbol{\theta}) := \frac{1}{N} \sum_{i=1}^{N} f_i(\boldsymbol{\theta}),$$

where each client's raw data remains localized, inherently preserving data privacy.

In real-world federated learning (FL) environments, clients' data distributions often exhibit significant heterogeneity ($\mathcal{D}_i \neq \mathcal{D}_j$ for $i \neq j$) due to factors such as diverse user behaviors, geographical disparities, and device-specific characteristics. This heterogeneity manifests across multiple dimensions, including feature distributions, label spaces, and variations in local dataset sizes and qualities, leading to substantially different optimization landscapes across clients. Conventional approaches handle this heterogeneity by imposing the following gradient dissimilarity bounds:

$$\frac{1}{N} \sum_{i=1}^{N} \|\nabla f_i(\boldsymbol{\theta})\|^2 \leq B\|\nabla f(\boldsymbol{\theta})\|^2 + \sigma_g^2. \quad (1)$$

However, such bounded heterogeneity assumptions are often restrictive and fail to capture real-world scenarios. In practice, client gradient norms can vary significantly, and the degree of heterogeneity may fluctuate unpredictably during training. Moreover, existing nonconvex FL methods rely on precise knowledge of system parameters (e.g., $L$, $\sigma^2$, $B$, $\sigma_g^2$) for stepsize tuning. Accurate estimation of these parameters, especially smoothness constants and heterogeneity bounds, requires global data access—a process that contradicts the fundamental privacy principles of FL and becomes increasingly intractable as data distributions evolve.

In this paper, we aim to address these challenges by leveraging momentum-based optimization and adaptive learning strategies. Our theoretical analysis relies only on the following assumptions.

**Assumption 2.1** ($L$-Smoothness). For any $i \in \{1, \cdots, N\}$, the local loss function $f_i$ is $L$-smooth that

$$\|\nabla f_i(\boldsymbol{\theta}) - \nabla f_i(\boldsymbol{\theta}')\| \leq L\|\boldsymbol{\theta} - \boldsymbol{\theta}'\|, \quad \forall \boldsymbol{\theta}, \boldsymbol{\theta}' \in \mathbb{R}^d.$$

**Assumption 2.2** (Stochastic Gradient). The stochastic gradients are unbiased estimators with bounded variance $\sigma^2$, such that

$$\mathbb{E}_{\boldsymbol{\xi}_i \sim \mathcal{D}_i}[\nabla F(\boldsymbol{\theta}; \boldsymbol{\xi}_i)] = \nabla f_i(\boldsymbol{\theta}),$$
$$\mathbb{E}_{\boldsymbol{\xi}_i \sim \mathcal{D}_i}\|\nabla F(\boldsymbol{\theta}; \boldsymbol{\xi}_i) - \nabla f_i(\boldsymbol{\theta})\|^2 \leq \sigma^2, \quad \forall i.$$

**Assumption 2.3** (Bounded Delay). Let $\tau_i^t$ represent the delay of client $i$ at the $t$th global round. Specifically, $\tau_i^t$ is defined as the time interval between the current global round $t$ and the global round when client $i$ began its local updates. We assume that the maximum update delay is bounded, i.e., $\tau_{\max} := \max_{i,t} \tau_i^t < \infty$. Correspondingly, the average delay is also bounded, i.e., $\bar{\tau} := \frac{1}{NT} \sum_{i,t} \tau_i^t < \infty$.

## 3. Momentum-Driven AFL

### 3.1. Algorithm Design

Momentum techniques have demonstrated remarkable effectiveness in mitigating data heterogeneity and accelerating convergence in synchronous FL. In the synchronous setting, (Cheng et al., 2024) showed that momentum helps remove data heterogeneity bounds (i.e., bound gradient dissimilarity) for nonconvex FL, marking a first in the literature. However, integrating momentum into AFL poses significant challenges due to fundamental conflicts between momentum's historical gradient accumulation and the inherent staleness of asynchronous updates. In asynchronous settings, clients' delayed updates compromise the integrity of momentum calculations, as stale gradients introduce biases into the optimization trajectory, as demonstrated in Yu et al. (2024). This issue is further exacerbated by the heterogeneity of client data distributions, where the global model may become biased toward the data distributions of faster clients. Moreover, asynchronous updates disrupt the maintenance of a coherent global momentum across all clients, making it challenging to accurately capture the true collective optimization trajectory of the distributed system.

To unlock the potential of momentum in asynchronous environments, we propose a **M**omentum-driven **as**ynchronous **F**ederated **L**earning (**MasFL**) framework, which effectively harnesses the benefits of momentum while systematically managing staleness accumulation. The complete server-side and client-side procedures of MasFL are presented in

**Algorithm 1** MasFL: Procedures at Central Server

1: **Require:** Initial model $\boldsymbol{\theta}^0$, control variates $\boldsymbol{c}_i^0 = \frac{1}{K} \sum_{k=0}^{K-1} \nabla F\left(\boldsymbol{\theta}^0; \boldsymbol{\xi}_i^{-1,k}\right)$ for any $i$, $\boldsymbol{c}^0 = \frac{1}{N} \sum_i \boldsymbol{c}_i^0$, momentum $\boldsymbol{g}^0 = \boldsymbol{c}^0$, global learning rate $\gamma$, local learning rate $\eta$, and momentum parameter $\beta$
2: **for** $t = 0, \cdots, T-1$ **do**
3:     Randomly selected a set of clients $\mathcal{S}_t$
4:     Update

$$\boldsymbol{c}_i^{t+1} = \begin{cases} \widetilde{\boldsymbol{c}}_i^{t+1}, & \text{if } i \in \mathcal{S}_t \\ \boldsymbol{c}_i^t, & \text{otherwise} \end{cases} \quad (2)$$

5:     Aggregate momentum

$$\boldsymbol{g}^{t+1} = \beta \left( \frac{1}{S} \sum_{i \in \mathcal{S}_t} \left( \boldsymbol{c}_i^{t+1} - \boldsymbol{c}_i^t \right) + \boldsymbol{c}^t \right) + (1-\beta)\boldsymbol{g}^t$$

6:     Update global model $\boldsymbol{\theta}^{t+1} = \boldsymbol{\theta}^t - \gamma \boldsymbol{g}^{t+1}$
7:     Aggregate control variate

$$\boldsymbol{c}^{t+1} = \boldsymbol{c}^t + \frac{1}{N} \sum_{i \in \mathcal{S}_t} \left( \boldsymbol{c}_i^{t+1} - \boldsymbol{c}_i^t \right)$$

8:     Send $\boldsymbol{\theta}^{t+1}$ and $\beta \boldsymbol{c}^{t+1} + (1-\beta)\boldsymbol{g}^{t+1}$ to all clients
9: **end for**

---

**Algorithm 2** MasFL: Procedures at Client $i$

1: Receive $\boldsymbol{\theta}^{t-\tau_i^t}$ and $\beta \boldsymbol{c}^{t-\tau_i^t} + (1-\beta)\boldsymbol{g}^{t-\tau_i^t}$ from server. Set $\boldsymbol{\theta}_i^{t,0} = \boldsymbol{\theta}^{t-\tau_i^t}$
2: **for** $k = 0, \cdots, K-1$ **do**
3:     Compute

$$\boldsymbol{g}_i^{t,k} = \beta \left( \nabla F\left( \boldsymbol{\theta}_i^{t,k}; \boldsymbol{\xi}_i^{t,k} \right) - \widetilde{\boldsymbol{c}}_i^t + \boldsymbol{c}^{t-\tau_i^t} \right) + (1-\beta)\boldsymbol{g}^{t-\tau_i^t}$$

4:     Update local model $\boldsymbol{\theta}_i^{t,k+1} = \boldsymbol{\theta}_i^{t,k} - \eta \boldsymbol{g}_i^{t,k}$
5: **end for**
6: Send $\widetilde{\boldsymbol{c}}_i^{t+1} = \frac{1}{K} \sum_{k=0}^{K-1} \nabla F\left( \boldsymbol{\theta}_i^{t,k}; \boldsymbol{\xi}_i^{t,k} \right)$ to the server

---

Algorithms 1 and 2, respectively. To address system asynchrony and data heterogeneity, MasFL introduces several key innovations, described as follows.

**Asynchronous Local Training.** Unlike synchronous FL where the system waits for all participating clients to complete their local updates before proceeding to the next round, MasFL allows the server and clients operate asynchronously, with the server maintaining a buffer for each client to store their latest updates. At each global round $t$, the server randomly selects a subset of clients, $\mathcal{S}_t$, to participate in global model aggregation. The aggregation uses the buffered updates, ensuring that it is independent of the current progress of the selected clients. Simultaneously, all clients performs local updates at their own pace. Upon completing $K$ local iterations, clients send their updates, $\widetilde{\boldsymbol{c}}_i^{t+1}$, to the server and immediately begin the next round of local updates based on the latest global model. This design naturally accommodates variations in client computational speeds and communication delays while maintaining overall learning efficiency.

Our asynchronous training is inspired by the FedBuff scheme proposed in (Nguyen et al., 2022), where the server randomly selects an active client set of size $M_c$ to perform local updates simultaneously and accumulates the fastest $S$ clients for the global model update. This scheme assumes *uniform arrivals of gradient computation* to ensure equal

chances of client participation. In contrast, our scheme maximally leverages clients' parallel computing capabilities while maintaining the uniformity of client selection. When $M_c = N$ and the uniform arrival assumption holds, both schemes achieve equivalent effects. Nevertheless, we emphasize that our asynchronous algorithm design is compatible with both buffer schemes.

**Stale Control Variates.** MasFL adopts a local control variate $\boldsymbol{c}_i^t$ to track the gradient information of each client $i$. At each round $t$, if client $i$ is selected for global aggregation, $\boldsymbol{c}_i^{t+1}$ is updated using its latest buffered value $\widetilde{\boldsymbol{c}}_i^{t+1}$; Otherwise, it retains its previous value. The global control variate, $\boldsymbol{c}^{t+1}$, is computed as the aggregation of all local control variates (Line 7, Algorithm 1), ensuring $\boldsymbol{c}^{t+1} = \frac{1}{N} \sum_{i=1}^N \boldsymbol{c}_i^{t+1}$ throughout the algorithm iteration. The difference between global and local variates, $\boldsymbol{c}_i^t - \boldsymbol{c}^t$ (Line 5, Algorithm 1), serves as a correction term to mitigate "client drift" caused by data heterogeneity. This design extends the control variate mechanism introduced in SCAFFOLD (Karimireddy et al., 2020b), but it is specifically adapted for asynchronous settings. On the client side, this correction term becomes $\widetilde{\boldsymbol{c}}_i^t - \boldsymbol{c}^{t-\tau_i^t}$, where the global control variate $\boldsymbol{c}^{t-\tau_i^t}$ is outdated due to system asynchrony. This inconsistency between the client- and server-side quantities introduces additional challenges in our theoretical analysis.

**Two-Level Momentum.** MasFL employs both global and local momentum to effectively handle asynchronous updates: i) *Global Momentum (Line 5, Algorithm 1)*. The server maintains a momentum $\boldsymbol{g}^{t+1}$, which aggregates client updates with a parameter $\beta$. This term stabilizes the global optimization trajectory by incorporating historical information. The difference in control variates, $\boldsymbol{c}_i^{t+1} - \boldsymbol{c}_i^t$, captures the progress of each client $i$. Since the participating clients at each round are selected uniformly, for any $t$, we have

$$\mathbb{E}_{\mathcal{S}_t} \left[ \frac{1}{S} \sum_{i \in \mathcal{S}_t} \left( \boldsymbol{c}_i^{t+1} - \boldsymbol{c}_i^t \right) + \boldsymbol{c}^t \right] = \frac{1}{N} \sum_{i=1}^N \boldsymbol{c}_i^{t+1} = \boldsymbol{c}^{t+1}.$$

Thus, $\boldsymbol{g}^{t+1}$ represents a weighted average of the global

**Algorithm 3** AdaMasFL: Procedures at Central Server

1: **Require:** Initial model $\boldsymbol{\theta}^0$, control variates $\boldsymbol{c}_i^0 = \frac{1}{K}\sum_{k=0}^{K-1}\nabla F\left(\boldsymbol{\theta}^0; \boldsymbol{\xi}_i^{-1,k}\right)$ for any $i$, $\boldsymbol{c}^0 = \frac{1}{N}\sum_i \boldsymbol{c}_i^0$, momentum $\boldsymbol{g}^0 = \boldsymbol{c}^0$, global learning rate $\gamma$, local learning rate $\eta$, and momentum parameter $\beta$.

2: **for** $t = 0, \cdots, T-1$ **do**

3:    Randomly selected a set of clients $\mathcal{S}_t$

4:    Update

$$\boldsymbol{c}_i^{t+1} = \begin{cases} \widetilde{\boldsymbol{c}}_i^{t+1}, & \text{if } i \in \mathcal{S}_t \\ \boldsymbol{c}_i^t, & \text{otherwise} \end{cases} \quad (3)$$

5:    Aggregate local updates $\overline{\boldsymbol{g}}^t = \frac{1}{S}\sum_{i\in\mathcal{S}_t}\boldsymbol{\Delta}_i^t$

6:    Update global model $\boldsymbol{\theta}^{t+1} = \boldsymbol{\theta}^t - \gamma\overline{\boldsymbol{g}}^t$

7:    Aggregate momentum

$$\boldsymbol{g}^{t+1} = \beta\left(\frac{1}{S}\sum_{i\in\mathcal{S}_t}\left(\boldsymbol{c}_i^{t+1} - \boldsymbol{c}_i^t\right) + \boldsymbol{c}^t\right) + (1-\beta)\boldsymbol{g}^t$$

8:    Aggregate control variate

$$\boldsymbol{c}^{t+1} = \boldsymbol{c}^t + \frac{1}{N}\sum_{i\in\mathcal{S}_t}\left(\boldsymbol{c}_i^{t+1} - \boldsymbol{c}_i^t\right)$$

9:    Download $\boldsymbol{\theta}^{t+1}$ and $\beta\boldsymbol{c}^{t+1} + (1-\beta)\boldsymbol{g}^{t+1}$ to all clients

10: **end for**

---

**Algorithm 4** AdaMasFL: Procedures at Client $i$

1: Receive $\boldsymbol{\theta}^{t-\tau_i^t}$ and $\beta\boldsymbol{c}^{t-\tau_i^t} + (1-\beta)\boldsymbol{g}^{t-\tau_i^t}$ from server. Set $\boldsymbol{\theta}_i^{t,0} = \boldsymbol{\theta}^{t-\tau_i^t}$

2: **for** $k = 0, \cdots, K-1$ **do**

3:    Compute

$$\boldsymbol{g}_i^{t,k} = \beta\left(\nabla F\left(\boldsymbol{\theta}_i^{t,k}; \boldsymbol{\xi}_i^{t,k}\right) - \widetilde{\boldsymbol{c}}_i^t + \boldsymbol{c}^{t-\tau_i^t}\right) + (1-\beta)\boldsymbol{g}^{t-\tau_i^t}$$

4:    Update local model

$$\boldsymbol{\theta}_i^{t,k+1} = \boldsymbol{\theta}_i^{t,k} - \eta\frac{\boldsymbol{g}_i^{t,k}}{\|\boldsymbol{g}_i^{t,k}\|}$$

5: **end for**

6: Send $\boldsymbol{\Delta}_i^t = \frac{1}{\eta K}\left(\boldsymbol{\theta}^{t-\tau_i^t} - \boldsymbol{\theta}_i^{t,K}\right)$ and $\widetilde{\boldsymbol{c}}_i^{t+1} = \frac{1}{K}\sum_{k=0}^{K-1}\nabla F\left(\boldsymbol{\theta}_i^{t,k}; \boldsymbol{\xi}_i^{t,k}\right)$ to the server

---

satisfied, i.e., $\eta \leq \frac{\sqrt{T - \sqrt{SKT} - 4SK\tau_{\max}^2}}{4\sqrt{5}eKL\sqrt{T}}$, then it holds that

$$\frac{1}{T}\sum_{t=0}^{T-1}\mathbb{E}\|\nabla f(\boldsymbol{\theta}^t)\|^2 \leq \mathcal{O}\left(\frac{L\Delta + \sigma^2}{\sqrt{SKT}} + \frac{\sigma^2}{T} + \frac{L^2\sigma^2}{NKT}\right)$$

for sufficiently large $T$, where $e$ denotes Euler's number and $\Delta := f(\boldsymbol{\theta}^0) - f^*$ represents the initial optimality gap with $f^* = \min_{\boldsymbol{\theta}} f(\boldsymbol{\theta}) > -\infty$.

Theorem 3.1 demonstrates that MasFL achieves a convergence rate of $\mathcal{O}\left(\frac{1}{\sqrt{SKT}}\right)$ in terms of $\mathbb{E}\|\nabla f(\boldsymbol{\theta})\|^2$ for sufficiently large $T$. This rate matches the convergence rates of traditional synchronous FL baselines (Karimireddy et al., 2020b), demonstrating that the delays inherent in asynchronous training do not degrade the asymptotic convergence guarantees.

To maintain the convergence order of $\mathcal{O}\left(\frac{1}{\sqrt{T}}\right)$ as in the synchronous case, MasFL requires the number of global rounds to satisfy $T \geq \mathcal{O}(\tau_{\max}^2)$. This condition is less restrictive than that required by CA$^2$FL (Wang et al., 2023) and FADAS (Wang et al., 2024c), which impose $T \geq \mathcal{O}(\tau_{\max}^4)$ (since the convergence rates of both papers are $\mathcal{O}(\frac{1}{\sqrt{T}} + \frac{\tau_{\max}\bar{\tau}}{T})$).

Following the standard definition (Arjevani et al., 2023), a point $\boldsymbol{\theta} \in \mathbb{R}^d$ is said to be an $\epsilon$-*stationary point* of a function $f : \mathbb{R}^d \to \mathbb{R}$ if $\|\nabla f(\boldsymbol{\theta})\| \leq \epsilon$. Under this definition, MasFL finds an $\epsilon$-stationary point in expectation within $\mathcal{O}\left(\frac{1}{SK\epsilon^4}\right)$ global rounds, indicating linear speedup with respect to both the number of participating clients $S$ and the number of local steps $K$. Notably, this convergence guarantee holds without requiring bounded gradient dissimilarity assumptions, such as (1), benefit from the carefully designed two-level momentum structure.

---

control variate $\boldsymbol{c}^{t+1}$ and its previous value $\boldsymbol{g}^t$. ii) *Local Momentum (Line 3, Algorithm 2).* Each client computes its local momentum $\boldsymbol{g}_i^{t,k}$ at every step of its local training. This local momentum combines the client's current gradient with the control variate correction, mitigating client drift and promoting alignment with the global objective. This two-level structure isolates all staleness representations on the client side, ensuring that the global momentum is computed based on the latest accumulators ($\boldsymbol{c}^t$ and $\boldsymbol{g}^t$), which is critical for our theoretical analysis.

### 3.2. Theoretical Analysis of MasFL

The following Theorem 3.1 presents the convergence rate of MasFL, whose proof is deferred to Appendix C.

**Theorem 3.1.** *Suppose that Assumptions 2.2 and 2.1 holds. Let $\{\boldsymbol{\theta}^t\}_{t=1}^T$ be the global iterates generated by MasFL. Set $\beta = \sqrt{\frac{SK}{T}}$ and $\gamma = \frac{1}{4L}\sqrt{\frac{SK}{T}}$. Define $a := \tau_{\max}^2\beta^2 + 20e^2\eta^2K^2L^2$. If the condition $1 - 4a - \sqrt{\frac{SK}{T}} \geq 0$ is*

## 4. Adaptive Momentum-Driven AFL

Although MasFL effectively leverages momentum in AFL to accelerate convergence while eliminating data heterogeneity bounds, it requires careful tuning of the learning rates that are theoretically dependent on the unkonwn smoothness parameter $L$ and delay bound $\tau_{\max}$. Estimating these parameters poses significant challenges and compromises the fundamental privacy guarantees of FL. To address this limitation, we further propose **Ada**ptive **M**omentum-driven **as**ynchronous **F**ederated **L**earning (AdaMasFL), which preserves the benefits of MasFL while achieving problem-parameter-free convergence.

### 4.1. Algorithm Design

AdaMasFL builds upon the framework of MasFL by seamlessly integrating momentum with adaptive learning rates. The complete procedures of AdaMasFL are presented in Algorithms 3 and 4. Compared to MasFL, the key innovation of AdaMasFL lies in its adoption of momentum-driven gradient normalization with a different server-side aggregation.

**Momentum-Driven Adaptivity.** AdaMasFL retains the momentum design of MasFL while introducing normalized gradients for local model updates (Line 4, Algorithm 4). This gradient normalization mitigates sensitivity to learning rate choices by automatically adjusting the effective stepsizes based on local optimization landscape. Normalized gradients ensure constant-magnitude updates regardless of gradient scales, satisfying

$$\left\| \eta \frac{\boldsymbol{g}_i^{t,k}}{\|\boldsymbol{g}_i^{t,k}\|} \right\| = \eta, \forall i, k, t.$$

By adapting the stepsize to gradient magnitudes, AdaMasFL eliminates the need for problem-dependent tuning based on unknown quantities such as the smoothness constant $L$.

However, gradient normalization alone is insufficient to achieve convergence in stochastic optimization, as it discards magnitude information. Recent studies (Hazan et al., 2015; Yang et al., 2023) have shown that normalized gradients can hinder convergence due to the randomness of stochastic gradient directions, even in simpler centralized convex settings. This issue is further exacerbated in AFL, where clients face heterogeneous gradient noise and delays. To address this, the momentum term plays a critical role by preserving consistent descent directions across clients and iterations.

**Server-Side Aggregation.** Unlike MasFL, where the server updates the global model based on the momentum $\boldsymbol{g}^{t+1}$ (Line 6, Algorithm 1), AdaMasFL updates the global model using the aggregation of local update directions (Line 6, Algorithm 3) that $\overline{\boldsymbol{g}}^t = \frac{1}{\eta SK} \sum_{i \in \mathcal{S}_t} \left( \boldsymbol{\theta}^{t-\tau_i^t} - \boldsymbol{\theta}_i^{t,K} \right)$. This

formulation ensures bounded global progress that

$$\|\gamma \overline{\boldsymbol{g}}^t\| = \left\| \frac{\gamma}{S} \sum_{i \in \mathcal{S}_t} \boldsymbol{\Delta}_i^t \right\| = \left\| \frac{\gamma}{SK} \sum_{i \in \mathcal{S}_t, k} \frac{\boldsymbol{g}_i^{t,k}}{\|\boldsymbol{g}_i^{t,k}\|} \right\| \leq \gamma.$$

Nevertheless, the global momentum is still retained in AdaMasFL to accelerate local model updates. These designs collectively enable AdaMasFL to achieve both normalization-induced stability and momentum-driven acceleration.

### 4.2. Theoretical Analysis of AdaMasFL

The following Theorem 4.1 presents the convergence rate of AdaMasFL, whose proof is deferred to Appendix D.

**Theorem 4.1.** *Suppose that Assumptions 2.1 and 2.2 holds. Let $\{\boldsymbol{\theta}^t\}_{t=1}^{T}$ be the global iterates generated by AdaMasFL. Set $\gamma = \frac{(SK)^{1/4}}{T^{3/4}}$, $\eta = \frac{1}{K\sqrt{T}}$, and $\beta = \sqrt{\frac{SK}{T}}$, then it holds that*

$$\frac{1}{T} \sum_{t=0}^{T-1} \mathbb{E} \left\| \nabla f(\boldsymbol{\theta}^t) \right\|$$

$$\leq \mathcal{O} \left( \frac{\Delta + L + \sigma + \sqrt{L\sigma}}{(SKT)^{\frac{1}{4}}} + \sigma \sqrt{\frac{SK}{T}} + \frac{\sqrt{L\sigma\tau_{\max}}}{T^{\frac{3}{8}}(SK)^{\frac{1}{8}}} \right.$$

$$\left. + \bar{\tau}\sigma \sqrt{\frac{S}{T}} + \tau_{\max} L \frac{(SK)^{\frac{1}{4}}}{T^{\frac{3}{4}}} + \bar{\tau} \sqrt{L\sigma\tau_{\max}} \frac{(SK)^{\frac{3}{8}}}{T^{\frac{7}{8}}} \right)$$

*for sufficiently large $T$.*

Theorem 4.1 demonstrates that AdaMasFL achieves a convergence rate of $\mathcal{O}\left(\frac{1}{(SKT)^{1/4}}\right)$ in terms of $\mathbb{E}\|\nabla f(\boldsymbol{\theta})\|$. Under the definition of $\epsilon$-stationarity, AdaMasFL finds an $\epsilon$-stationary point within $\mathcal{O}\left(\frac{1}{SK\epsilon^4}\right)$ global rounds, which aligns with the communication complexity of MasFL. Notably, Theorem 4.1 explicitly determines all hyperparameters, $\eta$, $\gamma$, and $\beta$ based solely on algorithm-specific constants: the number of participating clients $S$, local update iterations $K$, and communication rounds $T$. This eliminates the need for trial-and-error tuning or problem-parameter estimation, resulting in a completely problem-parameter-free AFL approach.

## 5. Comparisons with Existing AFL Methods

We now compare MasFL and AdaMasFL with several representative AFL approaches using various metrics, as listed in Table 1. Prior AFL algorithms face different theoretical and practical limitations. FedBuff (Nguyen et al., 2022), CA[2]FL (Wang et al., 2023), and DeFedAvg-nIID (Wang et al., 2024a) all require careful stepsize tuning based on the maximum delay $\tau_{\max}$. While CA[2]FL and DeFedAvg-nIID improve upon FedBuff by achieving a better convergence

Table 1: Comparisons of AFL algorithms for handling heterogeneous data.
(Convergence Rate = The convergence rate of different algorithms in terms of $\frac{1}{T}\sum_{t=0}^{T-1}\mathbb{E}\left\|\nabla f(\boldsymbol{\theta}^t)\right\|$; Additional Assumptions = Additional assumptions aside from Assumptions 2.1–2.3; BDH = Bounded data heterogeneity define in (1); BG = Bounded gradient that $\|\nabla f_i(\boldsymbol{\theta})\| \leq G, \ \forall i, \boldsymbol{\theta}$.)

| Algorithms | Convergence Rate[1] | Additional Assumptions | Stepsize Restrictions | Stepsize-related Problem-Parameters |
|---|---|---|---|---|
| FedBuff (Nguyen et al., 2022) | $\mathcal{O}\left(\frac{K^{1/4}\sigma_g}{(ST)^{1/4}} + \sqrt{\frac{\tau_{\max}\bar{\tau}}{T}}\right)$ | BDH | $\eta\gamma \leq \frac{1}{4K\tau_{\max}^{3/2}}$ | $\tau_{\max}$ |
| CA²FL (Wang et al., 2023) | $\mathcal{O}\left(\frac{1}{(SKT)^{1/4}} + \sqrt{\frac{\tau_{\max}}{T}}\right)$ | BDH | $\eta\gamma \leq \frac{S}{36K\tau_{\max}^2 L^2}, \eta \leq \frac{1}{36K\sqrt{\tau_{\max}}L}$ | $\tau_{\max}, L$ |
| DeFedAvg-nIID (Wang et al., 2024a) | $\mathcal{O}\left(\frac{1}{(SKT)^{1/4}} + \frac{1}{\sqrt{KT}}\right)$ | BDH, BG | $\eta\gamma \leq \frac{1}{4LK\tau_{\max}}, \eta \leq \frac{1}{4\sqrt{3}LK}$ | $\tau_{\max}, L$ |
| FADAS (Wang et al., 2024c) | $\mathcal{O}\left(\frac{1}{(ST)^{1/4}} + \sqrt{\frac{\tau_{\max}\bar{\tau}}{T}}\right)$ | BDH, BG | $\eta\gamma \leq \min\left\{\frac{\epsilon^2 S(N-1)}{180C_G^2 N(N-S)\tau_{\max}^2 KL}, \frac{\sqrt{\epsilon^3 S(N-1)}}{12\sqrt{C_G N(N-S)}\tau_{\max}^2 KL}\right\}, \eta \leq \frac{\sqrt{\epsilon}}{\sqrt{360C_G\tau_{\max}^2 KL}}^2$ | $\tau_{\max}, L, G$ |
| **MasFL** | $\mathcal{O}\left(\frac{\sqrt{\kappa}}{(SKT)^{1/4}} + \sqrt{\frac{\kappa}{T}}\right)^3$ | – | $\beta = \sqrt{\frac{SK}{T}}, \gamma = \frac{\beta}{4L}, \eta \leq \frac{\sqrt{T}-\sqrt{SKT}-4SK\tau_{\max}^2}{4\sqrt{5}eKL\sqrt{T}}$ | $\tau_{\max}, L$ |
| **AdaMasFL** | $\mathcal{O}\left(\frac{1}{(SKT)^{1/4}} + \frac{\sqrt{\tau_{\max}}}{T^{3/8}}\right)$ | – | $\beta = \sqrt{\frac{SK}{T}}, \gamma = \frac{(SK)^{1/4}}{T^{3/4}}, \eta = \frac{1}{K\sqrt{T}}$ | – |

[1] For the convergence rate defined in terms of $\frac{1}{T}\sum_{t=0}^{T-1}\|\nabla f(\boldsymbol{\theta}^t)\|^2$, we can readily obtained the corresponding rate with respect to $\frac{1}{T}\sum_{t=0}^{T-1}\|\nabla f(\boldsymbol{\theta}^t)\|$ by taking square root on both sides of the associated bound. This operator is verified by the following fact: $\frac{1}{T}\sum_{t=0}^{T-1}\mathbb{E}\|\nabla f(\boldsymbol{\theta}^t)\| = \frac{1}{T}\sum_{t=0}^{T-1}\mathbb{E}\sqrt{\|\nabla f(\boldsymbol{\theta}^t)\|^2} \leq \frac{1}{T}\sum_{t=0}^{T-1}\sqrt{\mathbb{E}\|\nabla f(\boldsymbol{\theta}^t)\|^2} \leq \sqrt{\frac{1}{T}\sum_{t=0}^{T-1}\mathbb{E}\|\nabla f(\boldsymbol{\theta}^t)\|^2}$, where the first and second inequalities utilizes Jensen's inequality as the square root function is concave.

[2] $C_G := \eta KG + \epsilon$ and $\epsilon > 0$ is an adaptive optimization parameter.

[3] $\kappa := \frac{3-4a}{1-4a-\sqrt{SK/T}}$ and $a := \tau_{\max}^2\beta^2 + 20e^2\eta^2 K^2 L^2$.

rate of $\mathcal{O}\left(\frac{1}{(SKT)^{1/4}}\right)$ and linear speedup with respect to $S$ and $K$, they still rely on bounded data heterogeneity assumptions and knowledge of the smoothness parameter $L$. FADAS (Wang et al., 2024c) employs the AMSGrad optimizer at the server-side to enable adaptive stepsize in AFL. However, its convergence guarantee relies on both bounded gradients and bounded data heterogeneity assumptions, and fails to achieve linear speedup with respect to local steps $K$.

In contrast, our algorithms achieve substantial improvements over existing approaches. Specifically, MasFL eliminates the bounded data heterogeneity assumption while attaining state-of-the-art communication efficiency. Building on this foundation, AdaMasFL maintains the best-known communication complexity while operating in a completely problem-parameter-free manner, with all learning rates explicitly determined by algorithm-specific constants $S$, $K$, and $T$. Collectively, these advancements mark a significant step forward in achieving robust and adaptive AFL.

## 6. Numerical Experiments

We evaluate the performance of our algorithms on the image classification task using two real-world datasets: CIFAR-10 (Li et al., 2017) and FMNIST (Xiao et al., 2017). For FMNIST, we utilize a convolutional neural network (CNN) consisting of three convolutional layers and two fully connected layers. For CIFAR-10, we adopt a ResNet-18 architecture (He et al., 2016). We compare our algorithms against two state-of-the-art AFL baselines: CA²FL (Wang et al., 2023) and FADAS (Wang et al., 2024c). We simulate the practical asynchronous conditions using FedBuff's delay mechanism. Specifically, at any given time, there are a total of $M_c$ clients performing local updates concurrently. The execution times of all clients are sampled randomly from a uniform distribution. These varying execution times naturally create different delays in global aggregation participation. Detailed experimental setups and additional simulation results are provided in Appendix E.

Figure 1 compares the test accuracy of various algorithms versus the number of communication rounds on both the CIFAR-10 and FMNIST datasets. For both datasets, we pro-

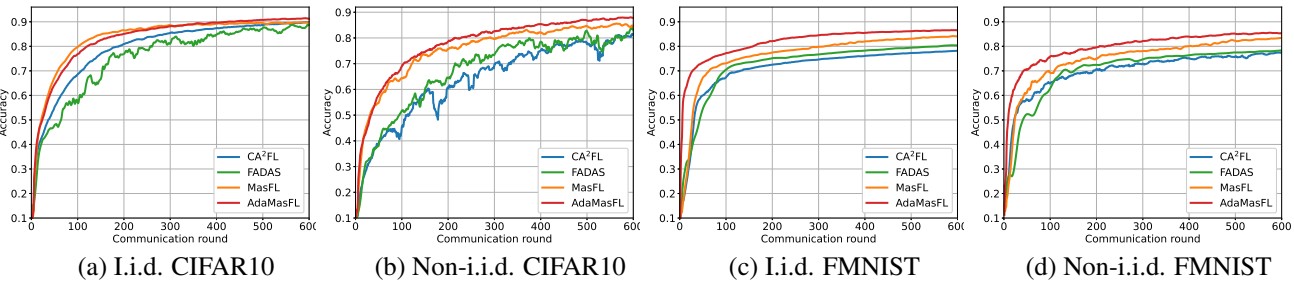

| (a) I.i.d. CIFAR10 | (b) Non-i.i.d. CIFAR10 | (c) I.i.d. FMNIST | (d) Non-i.i.d. FMNIST |

Figure 1: Test accuracy versus communication round on different datasets with i.i.d./non-i.i.d. data ($M_c = 20$).

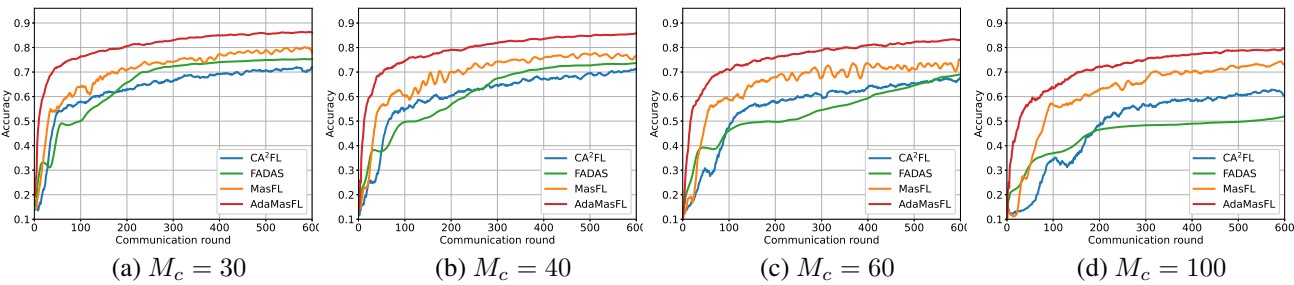

| (a) $M_c = 30$ | (b) $M_c = 40$ | (c) $M_c = 60$ | (d) $M_c = 100$ |

Figure 2: Test accuracy on the non-i.i.d. FMNIST dataset under varying levels of asynchrony.

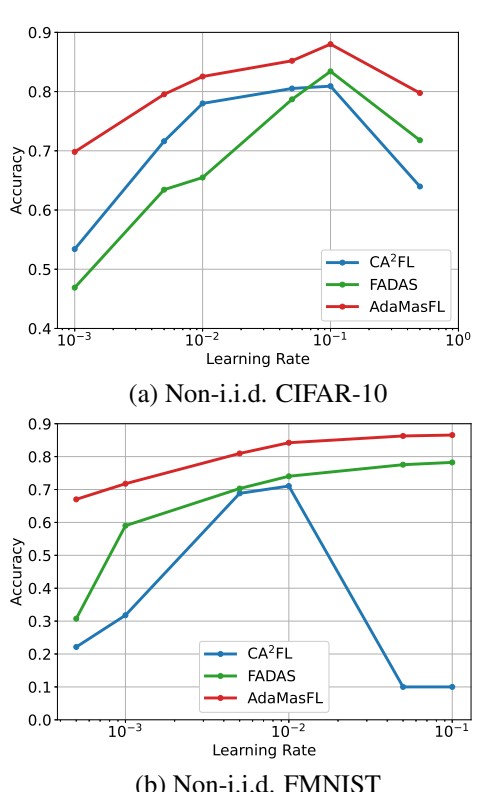

(a) Non-i.i.d. CIFAR-10

(b) Non-i.i.d. FMNIST

Figure 3: Test accuracy versus learning rate with non-i.i.d. data.

vide results for i.i.d. and non-i.i.d. data distributions. The learning rates of our algorithms, MasFL and AdaMasFL, are derived based on the theoretical guidance provided in Theorem 3.1 and Theorem 4.1, respectively. To ensure fair comparisons, the hyperparameters of the baseline methods, CA²FL and FADAS, are optimally selected according to the recommendations in their respective papers. The results show that our proposed methods, MasFL and AdaMasFL, significantly outperform the baselines, CA²FL and FADAS, in both convergence speed and test accuracy. This improvement can be attributed to our two-level momentum design, which accelerates optimization progress at each update. Additionally, the momentum mechanism stabilizes the optimization trajectory by accumulating historical gradients, as evidenced by the results in the non-i.i.d. setting. Notably, the advantages of our algorithms are particularly pronounced in the more challenging task, CIFAR-10 with non-i.i.d. data, as shown in Figure 1b.

Figure 2 illustrates the test accuracy of various algorithms on the non-i.i.d. FMNIST dataset under different asynchronous settings. Here, $M_c$ denotes the number of clients performing local updates concurrently. Since the server updates the global model only after collecting a total of $S$ client updates, a larger $M_c$ enables more frequent global aggregations, resulting in greater asynchronous delays. In Figure 2, we fix the learning rate settings of all algorithms to those used in Figure 1d and evaluate their performance as $M_c$ increases. We observe that AdaMasFL demonstrates exceptional robustness to varying levels of asynchrony, maintaining nearly consistent performance despite increasing

delays. In contrast, the performance of other algorithms deteriorates significantly, requiring reconfiguration to adapt to changing environments. Similarly, Figure 3 illustrates the test accuracy of various algorithms versus learning rate on both the CIFAR-10 and FMNIST datasets with non-i.i.d. data distributions. It is evident that AdaMasFL consistently outperforms the baselines, CA$^2$FL and FADAS, across the displayed learning rate regions.

## 7. Conclusions

This paper proposed two novel training approaches for AFL: MasFL and AdaMasFL. Specifically, MasFL introduced a two-level momentum mechanism that eliminates the requirement for data heterogeneity bounds in non-i.i.d. settings while achieving the best-known communication efficiency. Building on this foundation, AdaMasFL incorporated momentum-driven gradient normalization, removing all dependencies on problem-specific parameters in algorithm tuning while preserving the theoretical guarantees. Extensive numerical experiments demonstrated that our methods consistently outperform state-of-the-art AFL algorithms in runtime efficiency and exhibit exceptional robustness across a wide range of learning rates and asynchrony levels.

## Impact Statement

This paper presents work whose goal is to advance the field of Machine Learning. There are many potential societal consequences of our work, none which we feel must be specifically highlighted here.

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

# A. Related Works

Our work draws inspiration from and contributes to several active research areas in FL, including asynchronous optimization, momentum techniques, and adaptive stepsize methods.

**Asynchronous Training Paradigms.** Traditional asynchronous SGD methods (Mishchenko et al., 2022; Koloskova et al., 2022; Even et al., 2024) have established theoretical foundations for distributed optimization, particularly in addressing client synchronization challenges. While these methods have demonstrated linear speedup convergence under independent and indentically distributed (iid) settings and various objective functions, their effectiveness in non-iid scenarios has been limited. Moreover, these approaches typically focus on immediate gradient computation without local iterations, leading to increased communication overhead. Our work extends beyond these limitations by incorporating local training while maintaining theoretical guarantees in non-iid settings. Recent AFL frameworks like FedBuff (Nguyen et al., 2022), DeFedAvg-nIID (Wang et al., 2024a), and DuDe-ASGD (Wang et al., 2024b) have shown promising empirical results in handling heterogeneous data distributions. Methods such as QuAFL (Zakerinia et al., 2022) and FAVANO (Leconte et al., 2023) have explored different asynchrony models, including client interruption strategies, but face limitations in theoretical guarantees or practical implementation. Our framework distinguishes itself by achieving linear speedup without requiring bounded heterogeneity assumptions, while simultaneously providing robust performance across diverse learning rate configurations.

**Momentum-Based Techniques.** Momentum has emerged as a powerful technique for addressing data heterogeneity in FL. Various approaches, such as FedAvgM (Hsu et al., 2019) and SlowMo (Wang et al., 2019), have incorporated server-side momentum to enhance convergence rates. More sophisticated methods like MIME (Karimireddy et al., 2020a) and FAFED (Wu et al., 2023) have explored the combination of client- and server-side momentum with adaptive techniques. While these methods have achieved improved convergence properties, they generally require careful tuning of multiple hyperparameters. Recent work like FedSPS (Sohom Mukherjee, 2024) has attempted to minimize hyperparameter dependence but still relies on restrictive assumptions about gradient bounds and data heterogeneity. Yu et al. (2024) identify that asynchrony introduces implicit bias in momentum updates and propose momentum approximation for AFL, which finds optimal weights for historical model updates to approximate synchronous momentum behavior. FedAC (Zang et al., 2024) addresses both model staleness and client drift issues through adaptive server updates and corrective client training, though its empirical improvements lack theoretical convergence guarantees. Our approach advances this line of research by providing a fully adaptive framework that operates asynchronously without bounded heterogeneity assumptions or the need for problem-specific parameter tuning.

**Adaptive Learning Schemes.** Adaptive learning rates have been extensively studied and proven highly successful in single-machine learning scenarios (Duchi et al., 2011; Hazan et al., 2015; Reddi et al., 2018; Cutkosky & Mehta, 2020; Yang et al., 2023), where they effectively address the challenges of learning rate tuning and gradient scaling. In recent years, the critical need for hyperparameter-efficient training in federated learning has led to growing interest in extending these adaptive methods to FL settings. Server-side adaptive methods, such as FedOpt (Reddi et al., 2020) and its variants (FedAdaGrad, FedAdam, and FedYogi), have demonstrated enhanced convergence properties compared to traditional fixed learning rate approaches. Recent developments in local adaptive methods, including FAFED (Wu et al., 2023) and FedDA (Li et al., 2023), have explored client-side adaptation strategies. FADAS (Wang et al., 2024c) introduces a global adaptive AFL framework that applies adaptive optimization techniques at the server-side aggregation while supporting asynchronous client updates, and further proposes a delay-adaptive learning rate adjustment strategy to enhance resilience against large client delays. While these approaches have shown promise, they typically require bounded data heterogeneity assumptions and careful calibration of various hyperparameters. In contrast, our method integrates adaptive stepsizes with client-side momentum in an asynchronous setting, eliminating these restrictions while maintaining theoretical convergence guarantees.

# B. Technical Lemmas

Throughout the analysis, we use the following notation for summations:

- $\sum_i$ denotes summation over all clients $i \in \{1, \ldots, N\}$;

- $\sum_{i \in \mathcal{S}_t}$ denotes summation over selected clients in multiset $\mathcal{S}_t$;

- $\sum_k$ denotes summation over local steps $k \in \{0, \ldots, K-1\}$;

- $\sum_t$ denotes summation over global rounds $t \in \{0, \ldots, T-1\}$.

**Lemma B.1.** *Given vectors $\boldsymbol{\omega}_1, \cdots, \boldsymbol{\omega}_N \in \mathbb{R}^d$ and $\overline{\boldsymbol{\omega}} = \frac{1}{N} \sum_{i=1}^{N} \boldsymbol{\omega}_i$, if we sample $\mathcal{S} \subset \{1, \cdots, N\}$ uniformly randomly such that $|\mathcal{S}| = S$, then it holds that*

$$\mathbb{E}\left[\left\|\frac{1}{S}\sum_{i\in\mathcal{S}}\boldsymbol{\omega}_i\right\|^2\right] \leq \|\overline{\boldsymbol{\omega}}\|^2 + \frac{1}{SN}\sum_{i=1}^{N}\|\boldsymbol{\omega}_i - \overline{\boldsymbol{\omega}}\|^2.$$

*Proof.* Letting $\mathbb{1}\{i \in \mathcal{S}\}$ be the indicator for the event $i \in \mathcal{S}$, we prove this lemma by direct calculation as follows:

$$\mathbb{E}\left[\left\|\frac{1}{S}\sum_{i\in\mathcal{S}}\boldsymbol{\omega}_i\right\|^2\right] = \mathbb{E}\left[\left\|\frac{1}{S}\sum_{i=1}^{N}\boldsymbol{\omega}_i\mathbb{1}\{i\in\mathcal{S}\}\right\|^2\right]$$

$$= \frac{1}{S^2}\mathbb{E}\left[\left(\sum_i\|\boldsymbol{\omega}_i\|^2\,\mathbb{1}\{i\in\mathcal{S}\} + 2\sum_{i<j}\boldsymbol{\omega}_i^{\top}\boldsymbol{\omega}_j\,\mathbb{1}\{i,j\in\mathcal{S}\}\right)\right]$$

$$= \frac{1}{SN}\sum_{i=1}^{N}\|\boldsymbol{\omega}_i\|^2 + \frac{1}{S^2}\frac{S(S-1)}{N(N-1)}2\sum_{i<j}\boldsymbol{\omega}_i^{\top}\boldsymbol{\omega}_j$$

$$= \frac{1}{SN}\sum_{i=1}^{N}\|\boldsymbol{\omega}_i\|^2 + \frac{1}{S^2}\frac{S(S-1)}{N(N-1)}\left(\left\|\sum_{i=1}^{N}\boldsymbol{\omega}_i\right\|^2 - \sum_{i=1}^{N}\|\boldsymbol{\omega}_i\|^2\right)$$

$$= \frac{N-S}{S(N-1)}\frac{1}{N}\sum_{i=1}^{N}\|\boldsymbol{\omega}_i\|^2 + \frac{N(S-1)}{S(N-1)}\|\overline{\boldsymbol{\omega}}\|^2$$

$$= \frac{N-S}{S(N-1)}\frac{1}{N}\sum_{i=1}^{N}\|\boldsymbol{\omega}_i - \overline{\boldsymbol{\omega}}\|^2 + \|\overline{\boldsymbol{\omega}}\|^2$$

$$\leq \frac{1}{SN}\sum_{i=1}^{N}\|\boldsymbol{\omega}_i - \overline{\boldsymbol{\omega}}\|^2 + \|\overline{\boldsymbol{\omega}}\|^2,$$

where the last inequality uses the fact that $\frac{N-S}{N-1} \leq 1$ for any nonempty set $\mathcal{S}$. $\qquad\square$

**Lemma B.2.** *For any $i, t$, define $\phi_i^t := \mathbb{E}\left\|\nabla f_i\left(\boldsymbol{\theta}^{t-\tau_i^t}\right) - \boldsymbol{c}_i^{t+1}\right\|^2$, $\phi^t := \frac{1}{N}\sum_i \phi_i^t$, $\omega_i^t := \frac{1}{K}\sum_k \mathbb{E}\left\|\boldsymbol{\theta}_i^{t,k} - \boldsymbol{\theta}^{t-\tau_i^t}\right\|^2$, and $\boldsymbol{\omega}^t := \frac{1}{N}\sum_i \omega_i^t$. We have*

$$\frac{1}{T}\sum_{t=0}^{T-1}\phi^t \leq \frac{4N\sigma^2}{SKT} + \frac{4L^2\boldsymbol{\omega}^0}{T} + \frac{4\sigma^2}{K} + \frac{4L^2}{T}\sum_{t=0}^{T-1}\boldsymbol{\omega}^t$$

$$+ \frac{4N^2\gamma^2L^2}{S^2T}\sum_{t=0}^{T-1}\left(\mathbb{E}\|\mathcal{E}^t\|^2 + \mathbb{E}\|\nabla f\left(\boldsymbol{\theta}^t\right)\|^2\right).$$

*where $\mathcal{E}^t := \nabla f(\boldsymbol{\theta}^t) - \boldsymbol{g}^{t+1}$.*

*Proof.* Since for any $t$, the $S$ elements in $\mathcal{S}_t$ are uniformly sampled from $\{1, \cdots, N\}$, we have

$$\boldsymbol{c}_i^{t+1} = \begin{cases} \boldsymbol{c}_i^t & \text{if } i \in \mathcal{S}_t \text{ (w.p. } 1 - \frac{S}{N}) \\ \frac{1}{K}\sum_k \nabla F\left(\boldsymbol{\theta}_i^{t,k}; \boldsymbol{\xi}_i^{t,k}\right) & \text{if } i \notin \mathcal{S}_t \text{ (w.p. } \frac{S}{N}). \end{cases}$$

Using Young's inequality repeatedly, we have

$$
\begin{aligned}
\phi_i^t &= \left(1 - \frac{S}{N}\right) \mathbb{E}\left\|\nabla f_i\left(\boldsymbol{\theta}^{t-\tau_i^t}\right) - \boldsymbol{c}_i^t\right\|^2 + \frac{S}{N}\mathbb{E}\left\|\nabla f_i\left(\boldsymbol{\theta}^{t-\tau_i^t}\right) - \frac{1}{K}\sum_k \nabla F\left(\boldsymbol{\theta}_i^{t,k}; \boldsymbol{\xi}_i^{t,k}\right)\right\|^2 \\
&\leq \left(1 - \frac{S}{N}\right) \mathbb{E}\left\|\nabla f_i\left(\boldsymbol{\theta}^{t-\tau_i^t}\right) \mp \nabla f_i\left(\boldsymbol{\theta}^{t-\tau_i^t-1}\right) - \boldsymbol{c}_i^t\right\|^2 + \frac{2S}{N}\left(\frac{\sigma^2}{K} + L^2\boldsymbol{\omega}_i^t\right) \\
&\leq \left(1 - \frac{S}{N}\right) \mathbb{E}\left[\left(1 + \frac{S}{2N}\right)\phi_i^{t-1} + \left(1 + \frac{2N}{S}\right)L^2\left\|\boldsymbol{\theta}^{t-\tau_i^t} - \boldsymbol{\theta}^{t-\tau_i^t-1}\right\|^2\right] + \frac{2S}{N}\left(\frac{\sigma^2}{K} + L^2\boldsymbol{\omega}_i^t\right) \\
&\leq \left(1 - \frac{S}{2N}\right)\phi_i^{t-1} + \frac{4N}{S}\gamma^2 L^2\left(\mathbb{E}\left\|\mathcal{E}^{t-\tau_i^t-1}\right\|^2 + \mathbb{E}\left\|\nabla f\left(\boldsymbol{\theta}^{t-\tau_i^t-1}\right)\right\|^2\right) + \frac{2S}{N}\left(\frac{\sigma^2}{K} + L^2\boldsymbol{\omega}_i^t\right).
\end{aligned}
$$

Summing up the above inequality over $i$ and $t$ yields

$$
\begin{aligned}
\frac{S}{2N}\frac{1}{T}\sum_{t=0}^{T-1}\phi^t \leq &\frac{\phi^0}{T} + \frac{4N\gamma^2 L^2}{ST}\sum_{t=0}^{T-1}\left(\mathbb{E}\left\|\mathcal{E}^t\right\|^2 + \mathbb{E}\left\|\nabla f\left(\boldsymbol{\theta}^t\right)\right\|^2\right) \\
&+ \frac{2S\sigma^2}{NK} + \frac{2SL^2}{NT}\sum_{t=0}^{T-1}\boldsymbol{\omega}^t.
\end{aligned}
\tag{4}
$$

Since for any $i$, $\boldsymbol{c}_i^0 = \frac{1}{K}\sum_{k=0}^{K-1}\nabla F\left(\boldsymbol{\theta}^0; \boldsymbol{\xi}_i^{-1,k}\right)$ and $\boldsymbol{c}_i^1 = \frac{1}{K}\sum_{k=0}^{K-1}\nabla F\left(\boldsymbol{\theta}_i^{0,k}; \boldsymbol{\xi}_i^{0,k}\right)$. Then, we have

$$
\begin{aligned}
\phi_i^0 &= \left(1 - \frac{S}{N}\right)\mathbb{E}\left\|\nabla f_i\left(\boldsymbol{\theta}^0\right) - \boldsymbol{c}_i^0\right\|^2 + \frac{S}{N}\mathbb{E}\left\|\nabla f_i\left(\boldsymbol{\theta}^0\right) - \boldsymbol{c}_i^1\right\|^2 \\
&\leq \frac{2\sigma^2}{K} + \frac{2S}{N}L^2\boldsymbol{\omega}_i^0, \forall i.
\end{aligned}
$$

Thus, $\phi^0 = \frac{1}{N}\sum_i \phi_i^0 \leq \frac{2\sigma^2}{K} + \frac{2S}{N}L^2\boldsymbol{\omega}^0$. Plugging this inequality into (4) completes the proof. $\qquad\square$

**Lemma B.3.** *For any $i, t, k$, define $\boldsymbol{\omega}_i^t := \frac{1}{K}\sum_k\left\|\boldsymbol{\theta}_i^{t,k} - \boldsymbol{\theta}^{t-\tau_i^t}\right\|^2$, $\boldsymbol{\omega}^t := \frac{1}{N}\sum_i\boldsymbol{\omega}_i^t$, $\phi_i^t := \mathbb{E}\left\|\nabla f_i\left(\boldsymbol{\theta}^{t-\tau_i^t}\right) - \boldsymbol{c}_i^{t+1}\right\|^2$,*
*$\phi^t := \frac{1}{N}\sum_i\phi_i^t$, and $\boldsymbol{\zeta}_i^{t,k} := \mathbb{E}[\boldsymbol{\theta}_i^{t,k+1} - \boldsymbol{\theta}_i^{t,k}|\mathcal{F}_i^{t,k}] = -\eta\left(\beta\left(\nabla f_i\left(\boldsymbol{\theta}_i^{t,k}\right) - \widetilde{\boldsymbol{c}}_i^t + \boldsymbol{c}^{t-\tau_i^t}\right) + (1-\beta)\boldsymbol{g}^{t-\tau_i^t}\right)$. We have*

$$
\begin{aligned}
\frac{1}{T}\sum_{t=0}^{T-1}\boldsymbol{\omega}^t \leq &5\eta^2 K^2 e^{2\eta\beta KL}\left(2\beta^2\gamma^2 L^2(1 + 2\tau_{\max}^2) + 1\right)\frac{1}{T}\sum_{t=0}^{T-1}\left(\mathbb{E}\|\mathcal{E}^t\|^2 + \mathbb{E}\left\|\nabla f(\boldsymbol{\theta}^t)\right\|^2\right) \\
&+ 10\eta^2\beta^2 K^2 e^{2\eta\beta KL}\frac{1}{T}\sum_{t=0}^{T-1}\left(L^2\boldsymbol{\omega}^t + \phi^t\right) + (10e^{2\eta\beta KL} + 1)K\eta^2\beta^2\sigma^2 \\
&+ (1 + \eta\beta L)K^3\eta^3\beta^3 L\sigma^2 + \frac{\boldsymbol{\omega}^0}{T},
\end{aligned}
$$

*where $\boldsymbol{\omega}^0 \leq 3\eta^2 K^2 e^{2\eta\beta KL}\left\|\nabla f\left(\boldsymbol{\theta}^0\right)\right\|^2 + \left(3e^{2\eta\beta KL}\left(\beta^2 + \frac{1}{N}\right) + \beta^2\right)K\eta^2\sigma^2 + (1 + \eta\beta L)K^3\eta^3\beta^3 L\sigma^2$.*

*Proof.* From the definition of $\boldsymbol{\zeta}_i^{t,k}$, we have

$$
\begin{aligned}
\mathbb{E}\left\|\boldsymbol{\zeta}_i^{t,k} - \boldsymbol{\zeta}_i^{t,k-1}\right\|^2 &= \eta^2\beta^2\mathbb{E}\left\|\nabla f_i\left(\boldsymbol{\theta}_i^{t,k}\right) - \nabla f_i\left(\boldsymbol{\theta}_i^{t,k-1}\right)\right\|^2 \\
&\leq \eta^2\beta^2 L^2\mathbb{E}\left\|\boldsymbol{\theta}_i^{t,k} - \boldsymbol{\theta}_i^{t,k-1}\right\|^2 \\
&= \eta^2\beta^2 L^2\left(\mathbb{E}\left\|\boldsymbol{\zeta}_i^{t,k-1}\right\|^2 + \eta^2\beta^2\mathbb{E}\left\|\nabla f_i\left(\boldsymbol{\theta}_i^{t,k}\right) - \nabla F\left(\boldsymbol{\theta}_i^{t,k}; \boldsymbol{\xi}_i^{k,t}\right)\right\|^2\right) \\
&\leq \eta^2\beta^2 L^2\left(\mathbb{E}\left\|\boldsymbol{\zeta}_i^{t,k-1}\right\|^2 + \eta^2\beta^2\sigma^2\right).
\end{aligned}
$$

$$\mathbb{E}\left\|\boldsymbol{\zeta}_i^{t,j}\right\|^2 \leq \left(1 + \frac{1}{\eta\beta L}\right)\mathbb{E}\left\|\boldsymbol{\zeta}_i^{t,j} - \boldsymbol{\zeta}_i^{t,j-1}\right\|^2 + (1 + \eta\beta L)\mathbb{E}\left\|\boldsymbol{\zeta}_i^{t,j-1}\right\|^2$$

$$\leq \eta\beta L(1 + \eta\beta L)\mathbb{E}\left\|\boldsymbol{\zeta}_i^{t,j-1}\right\|^2 + (1 + \eta\beta L)\mathbb{E}\left\|\boldsymbol{\zeta}_i^{t,j-1}\right\|^2 + (1 + \eta\beta L)\eta^3\beta^3 L\sigma^2$$

$$= (1 + \eta\beta L)^2\mathbb{E}\left\|\boldsymbol{\zeta}_i^{t,j-1}\right\|^2 + (1 + \eta\beta L)\eta^3\beta^3 L\sigma^2$$

$$\leq (1 + \eta\beta L)^{2j}\mathbb{E}\left\|\boldsymbol{\zeta}_i^{t,0}\right\|^2 + j(1 + \eta\beta L)\eta^3\beta^3 L\sigma^2$$

$$\leq e^{2j\eta\beta L}\mathbb{E}\left\|\boldsymbol{\zeta}_i^{t,0}\right\|^2 + j(1 + \eta\beta L)\eta^3\beta^3 L\sigma^2,$$

where we use $(1 + \eta\beta L)^{\frac{1}{\eta\beta L}} \leq e$. Then, we have

$$\mathbb{E}\left\|\boldsymbol{\theta}_i^{t,k} - \boldsymbol{\theta}^{t-\tau_i^t}\right\|^2 \leq 2\mathbb{E}\left\|\sum_{j=0}^{k-1}\boldsymbol{\zeta}_i^{t,j}\right\|^2 + 2k\eta^2\beta^2\sigma^2$$

$$\leq 2k\sum_{j=0}^{k-1}\mathbb{E}\left\|\boldsymbol{\zeta}_i^{t,j}\right\|^2 + 2k\eta^2\beta^2\sigma^2$$

$$\leq 2k\sum_{j=0}^{k-1}\left(e^{2j\eta\beta L}\mathbb{E}\left\|\boldsymbol{\zeta}_i^{t,0}\right\|^2 + j(1 + \eta\beta L)\eta^3\beta^3 L\sigma^2\right) + 2k\eta^2\beta^2\sigma^2$$

$$\leq 2k^2 e^{2\eta\beta KL}\mathbb{E}\left\|\boldsymbol{\zeta}_i^{t,0}\right\|^2 + k^3(1 + \eta\beta L)\eta^3\beta^3 L\sigma^2 + 2k\eta^2\beta^2\sigma^2.$$

Summing up the above inequality over $k$ yields

$$\frac{1}{K}\sum_k \mathbb{E}\left\|\boldsymbol{\theta}_i^{t,k} - \boldsymbol{\theta}^{t-\tau_i^t}\right\|^2 \leq K^2 e^{2\eta\beta KL}\mathbb{E}\left\|\boldsymbol{\zeta}_i^{t,0}\right\|^2 + (1 + \eta\beta L)K^3\eta^3\beta^3 L\sigma^2 + K\eta^2\beta^2\sigma^2. \tag{5}$$

Based on the definination of $\boldsymbol{\zeta}_i^{t,k}$, $\boldsymbol{\zeta}_i^{t,0} = -\eta\left(\beta\left(\nabla f_i\left(\boldsymbol{\theta}^{t-\tau_i^t}\right) - \widetilde{\boldsymbol{c}}_i^t + \boldsymbol{c}^{t-\tau_i^t}\right) + (1-\beta)\boldsymbol{g}^{t-\tau_i^t}\right)$. Then, we have

$$\mathbb{E}\left\|\boldsymbol{\zeta}_i^{t,0}\right\|^2 = \eta^2\mathbb{E}\left\|\beta\left(\nabla f_i\left(\boldsymbol{\theta}^{t-\tau_i^t}\right) \mp \nabla f_i\left(\boldsymbol{\theta}^{t-\tau_i^t-1}\right) - \widetilde{\boldsymbol{c}}_i^t\right) + \beta\left(\boldsymbol{c}^{t-\tau_i^t} - \nabla f\left(\boldsymbol{\theta}^{t-\tau_i^t-1}\right)\right)\right.$$

$$\left. + (1-\beta)\left(\boldsymbol{g}^{t-\tau_i^t} - \nabla f\left(\boldsymbol{\theta}^{t-\tau_i^t-1}\right)\right) + \nabla f\left(\boldsymbol{\theta}^{t-\tau_i^t-1}\right)\right\|^2$$

$$\leq 5\eta^2\beta^2\left(L^2\mathbb{E}\left\|\boldsymbol{\theta}^{t-\tau_i^t} - \boldsymbol{\theta}^{t-\tau_i^t-1}\right\|^2 + \mathbb{E}\left\|\nabla f_i\left(\boldsymbol{\theta}^{t-\tau_i^t-1}\right) - \widetilde{\boldsymbol{c}}_i^t\right\|^2 + \mathbb{E}\left\|\nabla f\left(\boldsymbol{\theta}^{t-\tau_i^t-1}\right) - \boldsymbol{c}^{t-\tau_i^t}\right\|^2\right)$$

$$+ 5\eta^2(1-\beta)^2\mathbb{E}\left\|\mathcal{E}^{t-\tau_i^t-1}\right\|^2 + 5\eta^2\mathbb{E}\left\|\nabla f\left(\boldsymbol{\theta}^{t-\tau_i^t-1}\right)\right\|^2. \tag{6}$$

We know that $\mathbb{E}\left\|\boldsymbol{\theta}^{t-\tau_i^t} - \boldsymbol{\theta}^{t-\tau_i^t-1}\right\|^2 \leq 2\gamma^2\left(\mathbb{E}\left\|\mathcal{E}^{t-\tau_i^t-1}\right\|^2 + \mathbb{E}\left\|\nabla f\left(\boldsymbol{\theta}^{t-\tau_i^t-1}\right)\right\|^2\right)$. Additionally, for any $i, t$, we have

$$\mathbb{E}\left\|\nabla f_i\left(\boldsymbol{\theta}^{t-\tau_i^t}\right) - \widetilde{\boldsymbol{c}}_i^{t+1}\right\|^2 = \mathbb{E}\left\|\nabla f_i\left(\boldsymbol{\theta}^{t-\tau_i^t}\right) \mp \frac{1}{K}\sum_k \nabla F\left(\boldsymbol{\theta}_i^{t,k}\right) - \frac{1}{K}\sum_k \nabla F\left(\boldsymbol{\theta}_i^{t,k}; \boldsymbol{\xi}_i^{t,k}\right)\right\|^2$$

$$\leq 2L^2\boldsymbol{\omega}_i^t + \frac{2\sigma^2}{K}.$$

Further, for any $t$, we have

$$
\begin{aligned}
\mathbb{E}\left\|\nabla f\left(\boldsymbol{\theta}^t\right)-\boldsymbol{c}^{t+1}\right\|^2 &= \mathbb{E}\left\|\nabla f(\boldsymbol{\theta}^t) \mp \frac{1}{N}\sum_i \nabla f_i\left(\boldsymbol{\theta}^{t-\tau_i^t}\right)-\frac{1}{N}\sum_i \boldsymbol{c}_i^{t+1}\right\|^2 \\
&\leq 2\frac{1}{N}\sum_i \mathbb{E}\left\|\nabla f_i\left(\boldsymbol{\theta}^t\right)-\nabla f_i\left(\boldsymbol{\theta}^{t-\tau_i^t}\right)\right\|^2 + 2\phi^t \\
&\leq \frac{2L^2}{N}\sum_i \tau_i^t \sum_{d=t-\tau_i^t}^{t-1}\mathbb{E}\left\|\boldsymbol{\theta}^{d+1}-\boldsymbol{\theta}^d\right\|^2 + 2\phi^t \\
&\leq 4\gamma^2 L^2 \tau_{\max}\sum_{d=t-\tau_{\max}}^{t-1}\left(\mathbb{E}\|\mathcal{E}^d\|^2 + \mathbb{E}\left\|\nabla f\left(\boldsymbol{\theta}^d\right)\right\|^2\right) + 2\phi^t.
\end{aligned}
$$

Plugging the above results into (6), we have

$$
\begin{aligned}
\mathbb{E}\left\|\boldsymbol{\zeta}_i^{t,0}\right\|^2 \leq{}& 5\eta^2\beta^2\left(2\gamma^2 L^2\left(\mathbb{E}\left\|\mathcal{E}^{t-\tau_i^t-1}\right\|^2 + \mathbb{E}\left\|\nabla f\left(\boldsymbol{\theta}^{t-\tau_i^t-1}\right)\right\|^2\right) + 2L^2\boldsymbol{\omega}_i^{t-1} + \frac{2\sigma^2}{K}\right) \\
&+ 20\eta^2\beta^2\gamma^2 L^2\frac{\tau_{\max}}{N}\sum_i \sum_{d=t-\tau_i^t-1-\tau_{\max}}^{t-\tau_i^t-1-1}\left(\mathbb{E}\|\mathcal{E}^d\|^2 + \mathbb{E}\left\|\nabla f\left(\boldsymbol{\theta}^d\right)\right\|^2\right) + 10\eta^2\beta^2\phi^t \\
&+ 5\eta^2\mathbb{E}\left\|\mathcal{E}^{t-\tau_i^t-1}\right\|^2 + 5\eta^2\mathbb{E}\left\|\nabla f\left(\boldsymbol{\theta}^{t-\tau_i^t-1}\right)\right\|^2.
\end{aligned}
$$

Plugging the above inequality into (5) and summing over $i, t$ yields

$$
\begin{aligned}
\frac{1}{T}\sum_{t=0}^{T-1}\boldsymbol{\omega}^t \leq{}& 5\eta^2 K^2 e^{2\eta\beta KL}\left(2\beta^2\gamma^2 L^2(1+2\tau_{\max}^2)+1\right)\frac{1}{T}\sum_{t=0}^{T-1}\left(\mathbb{E}\|\mathcal{E}^t\|^2 + \mathbb{E}\left\|\nabla f(\boldsymbol{\theta}^t)\right\|^2\right) \\
&+ 10\eta^2\beta^2 K^2 e^{2\eta\beta KL}\frac{1}{T}\sum_{t=0}^{T-1}\left(L^2\boldsymbol{\omega}^t + \phi^t\right) + (10e^{2\eta\beta KL}+1)K\eta^2\beta^2\sigma^2 \\
&+ (1+\eta\beta L)K^3\eta^3\beta^3 L\sigma^2 + \frac{\boldsymbol{\omega}^0}{T}.
\end{aligned}
$$

Since $\boldsymbol{c}_i^0 = \frac{1}{K}\sum_{k=0}^{K-1}\nabla F\left(\boldsymbol{\theta}^0; \boldsymbol{\xi}_i^{-1,k}\right)$ for any $i$, $\boldsymbol{c}^0 = \frac{1}{N}\sum_i \boldsymbol{c}_i^0$, and $\boldsymbol{g}^0 = \boldsymbol{c}^0$, we have

$$
\begin{aligned}
\mathbb{E}\left\|\boldsymbol{\zeta}_i^{0,0}\right\|^2 &= \eta^2\mathbb{E}\left\|\beta\left(\nabla f_i\left(\boldsymbol{\theta}^0\right)-\boldsymbol{c}_i^0+\boldsymbol{c}^0\right)+(1-\beta)\boldsymbol{g}^0\right\|^2 \\
&= \eta^2\mathbb{E}\left\|\beta\left(\nabla f_i\left(\boldsymbol{\theta}^0\right)-\frac{1}{K}\sum_k \nabla F\left(\boldsymbol{\theta}^0; \boldsymbol{\xi}_i^{-1,k}\right)\right)+\frac{1}{NK}\sum_{i,k}\nabla F\left(\boldsymbol{\theta}^0; \boldsymbol{\xi}_i^{-1,k}\right)\mp\nabla f\left(\boldsymbol{\theta}^0\right)\right\|^2 \\
&\leq 3\eta^2\left(\frac{\beta^2\sigma^2}{K}+\frac{\sigma^2}{NK}+\left\|\nabla f\left(\boldsymbol{\theta}^0\right)\right\|^2\right).
\end{aligned}
$$

$$
\begin{aligned}
\boldsymbol{\omega}_i^0 &\leq K^2 e^{2\eta\beta KL}\frac{1}{N}\sum_i \mathbb{E}\left\|\boldsymbol{\zeta}_i^{0,0}\right\|^2 + (1+\eta\beta L)K^3\eta^3\beta^3 L\sigma^2 + K\eta^2\beta^2\sigma^2 \\
&\leq 3\eta^2 K^2 e^{2\eta\beta KL}\left\|\nabla f\left(\boldsymbol{\theta}^0\right)\right\|^2 + \left(3e^{2\eta\beta KL}\left(\beta^2+\frac{1}{N}\right)+\beta^2\right)K\eta^2\sigma^2 + (1+\eta\beta L)K^3\eta^3\beta^3 L\sigma^2.
\end{aligned}
$$

$\square$

## C. Proof of Theorem 3.1

Based on the $L$-smoothness of $f(\cdot)$ in Assumption 2.1 and Line 6 in Algorithm 1, we have

$$
\begin{aligned}
f(\boldsymbol{\theta}^{t+1}) - f(\boldsymbol{\theta}^t) &\leq \left\langle \nabla f(\boldsymbol{\theta}^t), \boldsymbol{\theta}^{t+1} - \boldsymbol{\theta}^t \right\rangle + \frac{L}{2} \left\| \boldsymbol{\theta}^{t+1} - \boldsymbol{\theta}^t \right\|^2 \\
&= -\gamma \|\nabla f(\boldsymbol{\theta}^t)\|^2 + \gamma \left\langle \nabla f(\boldsymbol{\theta}^t), \nabla f(\boldsymbol{\theta}^t) - \boldsymbol{g}^{t+1} \right\rangle + \frac{\gamma^2 L}{2} \left\| \boldsymbol{g}^{t+1} \mp \nabla f(\boldsymbol{\theta}^t) \right\|^2 \\
&\leq -\frac{\gamma}{2} \|\nabla f(\boldsymbol{\theta}^t)\|^2 + \frac{\gamma}{2} \left\| \nabla f(\boldsymbol{\theta}^t) - \boldsymbol{g}^{t+1} \right\|^2 + L\gamma^2 \left\| \nabla f(\boldsymbol{\theta}^t) - \boldsymbol{g}^{t+1} \right\|^2 + \gamma^2 L \|\nabla f(\boldsymbol{\theta}^t)\|^2 \\
&= -\frac{\gamma}{2}(1 - 2\gamma L)\|\nabla f(\boldsymbol{\theta}^t)\|^2 + \frac{\gamma}{2}(1 + 2\gamma L)\|\nabla f(\boldsymbol{\theta}^t) - \boldsymbol{g}^{t+1}\|^2.
\end{aligned}
$$

Define $\mathcal{E}^t := \nabla f(\boldsymbol{\theta}^t) - \boldsymbol{g}^{t+1}$ and $\Delta := f(\boldsymbol{\theta}^0) - f^*$, where $f^* := \min_{\boldsymbol{\theta}} f(\boldsymbol{\theta}) > -\infty$. Then, $f(\boldsymbol{\theta}^0) - f(\boldsymbol{\theta}^T) \leq f(\boldsymbol{\theta}^0) - f^* = \Delta$. Summing the above inequality over $t = 0, 1, \ldots, T-1$ yields

$$
\frac{1}{T} \sum_{t=0}^{T-1} \|\nabla f(\boldsymbol{\theta}^t)\|^2 \leq \frac{2\Delta}{\gamma(1 - 2\gamma L)T} + \frac{1 + 2\gamma L}{1 - 2\gamma L} \frac{1}{T} \sum_{t=0}^{T-1} \|\mathcal{E}^t\|^2. \tag{7}
$$

From the update rule of $\boldsymbol{g}^{t+1}$ given in Line 5 in Algorithm 1, we have

$$
\begin{aligned}
\mathbb{E}\|\mathcal{E}^t\|^2 &= \mathbb{E} \left\| (1-\beta)\left(\nabla f(\boldsymbol{\theta}^t) - \boldsymbol{g}^t\right) + \beta \underbrace{\left( \nabla f(\boldsymbol{\theta}^t) - \boldsymbol{c}^t - \frac{1}{S} \sum_{i \in \mathcal{S}_t} \left( \boldsymbol{c}_i^{t+1} - \boldsymbol{c}_i^t \right) \right)}_{=:\boldsymbol{v}^t} \right\|^2 \\
&= (1-\beta)^2 \mathbb{E}\left\| \nabla f(\boldsymbol{\theta}^t) - \boldsymbol{g}^t \right\|^2 + \beta^2 \mathbb{E}\|\boldsymbol{v}^t\|^2 + 2\beta \mathbb{E}\left\langle (1-\beta)\left(\nabla f(\boldsymbol{\theta}^t) - \boldsymbol{g}^t\right), \boldsymbol{v}^t \right\rangle \\
&\stackrel{(a)}{=} (1-\beta)^2 \mathbb{E}\left\| \nabla f(\boldsymbol{\theta}^t) - \boldsymbol{g}^t \right\|^2 + \beta^2 \mathbb{E}\|\boldsymbol{v}^t\|^2 + 2\beta \mathbb{E}\left\langle (1-\beta)\left(\nabla f(\boldsymbol{\theta}^t) - \boldsymbol{g}^t\right), \underbrace{\nabla f(\boldsymbol{\theta}^t) - \frac{1}{NK} \sum_{i,k} \nabla f_i\left(\boldsymbol{\theta}_i^{t,k}\right)}_{=:\boldsymbol{\psi}^t} \right\rangle \\
&\leq (1-\beta)^2 \left(1 + \frac{\beta}{2}\right) \mathbb{E}\left\| \nabla f(\boldsymbol{\theta}^t) - \boldsymbol{g}^t \right\|^2 + \beta^2 \mathbb{E}\|\boldsymbol{v}^t\|^2 + 2\beta \mathbb{E}\left\|\boldsymbol{\psi}^t\right\|^2,
\end{aligned} \tag{8}
$$

where $(a)$ is based on the fact that

$$
\begin{aligned}
\mathbb{E}[\boldsymbol{v}^t | \mathcal{F}^t] &= \mathbb{E}_{\{\boldsymbol{\xi}_i^{t,k}\}_{\forall i,k}, \mathcal{S}_t}[\boldsymbol{v}^t] \\
&= \mathbb{E}_{\{\boldsymbol{\xi}_i^{t,k}\}_{\forall i,k}} \left[ \nabla f(\boldsymbol{\theta}^t) - \boldsymbol{c}^t - \frac{1}{NK} \sum_{i,k} \left( \nabla F\left(\boldsymbol{\theta}_i^{t,k}; \boldsymbol{\xi}_i^{t,k}\right) - \boldsymbol{c}_i^t \right) \right] \\
&= \nabla f(\boldsymbol{\theta}^t) - \frac{1}{NK} \sum_{i,k} \nabla f_i\left(\boldsymbol{\theta}_i^{t,k}\right) =: \boldsymbol{\psi}^t.
\end{aligned}
$$

To upper bound $(1 - \beta)^2 \left(1 + \frac{\beta}{2}\right) \mathbb{E} \left\|\nabla f(\boldsymbol{\theta}^t) - \boldsymbol{g}^t\right\|^2$ in (8), we have

$$(1 - \beta)^2 \left(1 + \frac{\beta}{2}\right) \mathbb{E} \left\|\nabla f(\boldsymbol{\theta}^t) - \boldsymbol{g}^t\right\|^2$$

$$\leq (1 - \beta)^2 \left(1 + \frac{\beta}{2}\right) \mathbb{E} \left[\left(1 + \frac{\beta}{2}\right) \left\|\mathcal{E}^{t-1}\right\|^2 + \left(1 + \frac{2}{\beta}\right) \left\|\nabla f(\boldsymbol{\theta}^t) - \nabla f\left(\boldsymbol{\theta}^{t-1}\right)\right\|^2\right]$$

$$\leq (1 - \beta)\mathbb{E} \left\|\mathcal{E}^{t-1}\right\|^2 + \frac{2L^2}{\beta} \left\|\boldsymbol{\theta}^t - \boldsymbol{\theta}^{t-1}\right\|^2$$

$$\leq (1 - \beta)\mathbb{E} \left\|\mathcal{E}^{t-1}\right\|^2 + \frac{4L^2\gamma^2}{\beta} \left(\mathbb{E} \left\|\mathcal{E}^{t-1}\right\|^2 + \left\|\nabla f\left(\boldsymbol{\theta}^{t-1}\right)\right\|^2\right)$$

$$= \left(1 - \beta + \frac{4L^2\gamma^2}{\beta}\right) \mathbb{E} \left\|\mathcal{E}^{t-1}\right\|^2 + \frac{4L^2\gamma^2}{\beta} \mathbb{E} \left\|\nabla f\left(\boldsymbol{\theta}^{t-1}\right)\right\|^2.$$

Define $\phi_i^t := \mathbb{E} \left\|\nabla f_i\left(\boldsymbol{\theta}^{t-\tau_i^t}\right) - \boldsymbol{c}_i^{t+1}\right\|^2$, $\phi^t := \frac{1}{N} \sum_i \phi_i^t$, $\omega_i^t := \frac{1}{K} \sum_k \mathbb{E} \left\|\boldsymbol{\theta}_i^{t,k} - \boldsymbol{\theta}^{t-\tau_i^t}\right\|^2$, and $\omega^t := \frac{1}{N} \sum_i \omega_i^t$. To upper bound $\mathbb{E}\|\boldsymbol{v}^t\|^2$ in (8), it follows from Lemma B.1 that

$$\mathbb{E}\|\boldsymbol{v}^t\|^2$$

$$\leq \mathbb{E} \left\|\nabla f(\boldsymbol{\theta}^t) - \frac{1}{N} \sum_i \boldsymbol{c}_i^{t+1}\right\|^2 + \frac{1}{S} \frac{1}{N} \sum_i \mathbb{E} \left\|(\boldsymbol{c}_i^{t+1} - \boldsymbol{c}_i^t) - \sum_{i'} (\boldsymbol{c}_{i'}^{t+1} - \boldsymbol{c}_{i'}^t)\right\|^2$$

$$\leq \mathbb{E} \left\|\nabla f(\boldsymbol{\theta}^t) - \frac{1}{NK} \sum_{i,k} \nabla F\left(\boldsymbol{\theta}_i^{t,k}; \boldsymbol{\xi}_i^{t,k}\right)\right\|^2 + \frac{1}{S} \frac{1}{N} \sum_i \mathbb{E} \left\|\frac{1}{K} \sum_k \nabla F\left(\boldsymbol{\theta}_i^{t,k}; \boldsymbol{\xi}_i^{t,k}\right) - \boldsymbol{c}_i^t\right\|^2$$

$$\leq 2\mathbb{E} \left\|\boldsymbol{\psi}^t\right\|^2 + \frac{2\sigma^2}{NK} + \frac{1}{SN} \sum_i \mathbb{E} \left\|\frac{1}{K} \sum_k \left(\nabla F\left(\boldsymbol{\theta}_i^{t,k}; \boldsymbol{\xi}_i^{t,k}\right) \mp \nabla f_i\left(\boldsymbol{\theta}_i^{t,k}\right)\right) \mp \nabla f_i\left(\boldsymbol{\theta}^{t-\tau_i^t}\right) \mp \nabla f_i\left(\boldsymbol{\theta}^{t-\tau_i^t-1}\right) - \boldsymbol{c}_i^t\right\|^2$$

$$\leq 2\mathbb{E} \left\|\boldsymbol{\psi}^t\right\|^2 + \frac{6\sigma^2}{SK} + \frac{4L^2}{S} \omega^t + \frac{4L^2}{SN} \sum_i \left\|\boldsymbol{\theta}^{t-\tau_i^t} - \boldsymbol{\theta}^{t-\tau_i^t-1}\right\|^2 + \frac{4}{S} \phi^{t-1}$$

$$\leq \mathbb{E} \left\|\boldsymbol{\psi}^t\right\|^2 + \frac{6\sigma^2}{SK} + \frac{4L^2}{S} \omega^t + \frac{8\gamma^2 L^2}{SN} \sum_i \left(\mathbb{E} \left\|\mathcal{E}^{t-\tau_i^t-1}\right\|^2 + \mathbb{E} \left\|\nabla f\left(\boldsymbol{\theta}^{t-\tau_i^t-1}\right)\right\|^2\right) + \frac{4}{S} \phi^{t-1}.$$

Plugging the above results into (8), we have

$$\mathbb{E}\|\mathcal{E}^t\|^2 \leq \left(1 - \beta + \frac{4L^2\gamma^2}{\beta}\right) \mathbb{E} \left\|\mathcal{E}^{t-1}\right\|^2 + \frac{4L^2\gamma^2}{\beta} \mathbb{E} \left\|\nabla f\left(\boldsymbol{\theta}^{t-1}\right)\right\|^2 + 2\beta(\beta + 1)\mathbb{E} \left\|\boldsymbol{\psi}^t\right\|^2 + \frac{6\beta^2\sigma^2}{SK}$$

$$+ \frac{4\beta^2 L^2}{S} \omega^t + \frac{8\beta^2\gamma^2 L^2}{SN} \sum_i \left(\mathbb{E} \left\|\mathcal{E}^{t-\tau_i^t-1}\right\|^2 + \mathbb{E} \left\|\nabla f\left(\boldsymbol{\theta}^{t-\tau_i^t-1}\right)\right\|^2\right) + \frac{4\beta^2}{S} \phi^{t-1}.$$

Additionally, for any $t$, we have

$$\mathbb{E} \left\|\boldsymbol{\psi}^t\right\|^2 = \mathbb{E} \left\|\nabla f(\boldsymbol{\theta}^t) \mp \frac{1}{N} \sum_i \nabla f_i\left(\boldsymbol{\theta}^{t-\tau_i^t}\right) - \frac{1}{NK} \sum_{i,k} \nabla f_i\left(\boldsymbol{\theta}_i^{t,k}\right)\right\|^2$$

$$\leq 2\mathbb{E} \left\|\frac{1}{N} \sum_i \left(\nabla f_i(\boldsymbol{\theta}^t) - \nabla f_i\left(\boldsymbol{\theta}^{t-\tau_i^t}\right)\right)\right\|^2 + 2L^2 \omega^t$$

$$\leq 2L^2 \frac{1}{N} \sum_i \tau_i^t \sum_{d=t-\tau_i^t}^{t-1} \mathbb{E} \left\|\boldsymbol{\theta}^{d+1} - \boldsymbol{\theta}^d\right\|^2 + 2L^2 \omega^t$$

$$\leq 4\tau_{\max}\gamma^2 L^2 \sum_{d=t-\tau_{\max}}^{t-1} \left(\mathbb{E}\|\mathcal{E}^d\|^2 + \mathbb{E} \left\|\nabla f\left(\boldsymbol{\theta}^d\right)\right\|^2\right) + 2L^2 \omega^t.$$

Thus,

$$
\mathbb{E}\|\mathcal{E}^t\|^2 \leq \left(1 - \beta + \frac{4L^2\gamma^2}{\beta}\right) \mathbb{E}\left\|\mathcal{E}^{t-1}\right\|^2 + \frac{4L^2\gamma^2}{\beta}\mathbb{E}\left\|\nabla f\left(\boldsymbol{\theta}^{t-1}\right)\right\|^2 + 4\beta L^2\left(1 + \beta + \frac{\beta}{S}\right)\boldsymbol{\omega}^t
$$

$$
+ \frac{8\beta^2\gamma^2 L^2}{SN}\sum_i \left(\mathbb{E}\left\|\mathcal{E}^{t-\tau_i^t-1}\right\|^2 + \mathbb{E}\left\|\nabla f\left(\boldsymbol{\theta}^{t-\tau_i^t-1}\right)\right\|^2\right) + \frac{4\beta^2}{S}\phi^{t-1}
$$

$$
+ 16\tau_{\max}\beta\gamma^2 L^2 \sum_{d=t-\tau_{\max}}^{t-1}\left(\mathbb{E}\|\mathcal{E}^d\|^2 + \mathbb{E}\left\|\nabla f\left(\boldsymbol{\theta}^d\right)\right\|^2\right) + \frac{6\beta^2\sigma^2}{SK}.
$$

Summing up the above inequality over $t$ yields

$$
\left(1 - \frac{4L^2\gamma^2}{\beta^2}\right)\frac{1}{T}\sum_{t=0}^{T-1}\mathbb{E}\|\mathcal{E}^t\|^2
$$

$$
\leq \frac{\mathbb{E}\|\mathcal{E}^0\|^2}{\beta T} + \frac{4L^2\gamma^2}{\beta^2 T}\sum_{t=0}^{T-1}\mathbb{E}\left\|\nabla f(\boldsymbol{\theta}^t)\right\|^2 + 4L^2\frac{1+\beta+\frac{\beta}{S}}{T}\sum_{t=0}^{T-1}\boldsymbol{\omega}^t + \frac{4\beta}{ST}\sum_{t=0}^{T-1}\phi^t
$$

$$
+ 8\gamma^2 L^2\left(2\tau_{\max}^2 + \frac{\beta}{S}\right)\frac{1}{T}\sum_{t=0}^{T-1}\left(\mathbb{E}\left\|\mathcal{E}^t\right\|^2 + \mathbb{E}\left\|\nabla f\left(\boldsymbol{\theta}^t\right)\right\|^2\right) + \frac{6\beta\sigma^2}{SK}. \tag{9}
$$

From Lemma B.2, we know that

$$
\frac{1}{T}\sum_{t=0}^{T-1}\phi^t \leq \frac{4N\sigma^2}{SKT} + \frac{4L^2\boldsymbol{\omega}^0}{T} + \frac{4\sigma^2}{K} + \frac{4L^2}{T}\sum_{t=0}^{T-1}\boldsymbol{\omega}^t
$$

$$
+ \frac{4N^2\gamma^2 L^2}{S^2 T}\sum_{t=0}^{T-1}\left(\mathbb{E}\left\|\mathcal{E}^t\right\|^2 + \mathbb{E}\left\|\nabla f\left(\boldsymbol{\theta}^t\right)\right\|^2\right).
$$

From Lemma B.3, we know that

$$
\frac{1}{T}\sum_{t=0}^{T-1}\boldsymbol{\omega}^t \leq 5\eta^2 K^2 e^{2\eta\beta KL}\left(2\beta^2\gamma^2 L^2(1+2\tau_{\max}^2)+1\right)\frac{1}{T}\sum_{t=0}^{T-1}\left(\mathbb{E}\|\mathcal{E}^t\|^2 + \mathbb{E}\left\|\nabla f(\boldsymbol{\theta}^t)\right\|^2\right)
$$

$$
+ 10\eta^2\beta^2 K^2 e^{2\eta\beta KL}\frac{1}{T}\sum_{t=0}^{T-1}\left(L^2\boldsymbol{\omega}^t + \phi^t\right) + (10e^{2\eta\beta KL}+1)K\eta^2\beta^2\sigma^2
$$

$$
+ (1+\eta\beta L)K^3\eta^3\beta^3 L\sigma^2 + \frac{\boldsymbol{\omega}^0}{T}.
$$

Plugging the bound of $\frac{1}{T}\sum_{t=0}^{T-1}\phi^t$ into $\frac{1}{T}\sum_{t=0}^{T-1}\boldsymbol{\omega}^t$ yields

$$
\frac{1}{T}\sum_{t=0}^{T-1}\boldsymbol{\omega}^t \leq 5\eta^2 K^2 e^{2\eta\beta KL}\left(2\beta^2\gamma^2 L^2\left(1+2\tau_{\max}^2+\frac{4N^2}{S^2}\right)+1\right)\frac{1}{T}\sum_{t=0}^{T-1}\left(\mathbb{E}\|\mathcal{E}^t\|^2 + \mathbb{E}\left\|\nabla f(\boldsymbol{\theta}^t)\right\|^2\right)
$$

$$
+ 50\eta^2\beta^2 K^2 L^2 e^{2\eta\beta KL}\frac{1}{T}\sum_{t=0}^{T-1}\boldsymbol{\omega}^t + (50e^{2\eta\beta KL}+1)K\eta^2\beta^2\sigma^2
$$

$$
+ (1+\eta\beta L)K^3\eta^3\beta^3 L\sigma^2 + \frac{\boldsymbol{\omega}^0}{T} + 10\eta^2\beta^2 K^2 e^{2\eta\beta KL}\left(\frac{4N\sigma^2}{SKT} + \frac{4L^2\boldsymbol{\omega}^0}{T}\right).
$$

Set $\beta = \mathcal{O}\left(\frac{1}{\sqrt{T}}\right)$, $\gamma = \frac{\beta}{4L}$, and $\eta KL \leq \mathcal{O}(1)$. Then, $e^{2\eta\beta KL} \leq e^2$. Thus, we have

$$
\frac{1}{T}\sum_{t=0}^{T-1}\boldsymbol{\omega}^t \lesssim 5\eta^2 K^2 e^2\frac{1}{T}\sum_{t=0}^{T-1}\left(\mathbb{E}\|\mathcal{E}^t\|^2 + \mathbb{E}\left\|\nabla f(\boldsymbol{\theta}^t)\right\|^2\right) + 50e^2 K\eta^2\beta^2\sigma^2 + \frac{\boldsymbol{\omega}^0}{T}.
$$

where $\lesssim$ means "less than or asymptotically equal to" in order sense.

Similarly, plugging the above bound back into $\frac{1}{T}\sum_{t=0}^{T-1}\phi^t$, we have

$$\frac{1}{T}\sum_{t=0}^{T-1}\phi^t \lesssim \frac{8L^2\boldsymbol{\omega}^0}{T} + \frac{4\sigma^2}{K} + 4L^2\left(5\eta^2K^2e^2 + \frac{N^2}{S^2}\gamma^2\right)\frac{1}{T}\sum_{t=0}^{T-1}\left(\mathbb{E}\left\|\mathcal{E}^t\right\|^2 + \mathbb{E}\left\|\nabla f\left(\boldsymbol{\theta}^t\right)\right\|^2\right).$$

Plugging the above bounds on $\frac{1}{T}\sum_{t=0}^{T-1}\boldsymbol{\omega}^t$ and $\frac{1}{T}\sum_{t=0}^{T-1}\phi^t$ into (9), we have

$$\frac{3}{4}\frac{1}{T}\sum_{t=0}^{T-1}\mathbb{E}\|\mathcal{E}^t\|^2 \lesssim \frac{\mathbb{E}\|\mathcal{E}^0\|^2}{\beta T} + \frac{1}{4T}\sum_{t=0}^{T-1}\mathbb{E}\left\|\nabla f(\boldsymbol{\theta}^t)\right\|^2 + \frac{22\beta\sigma^2}{SK} + 200e^2K\eta^2\beta^2L^2\sigma^2 + \frac{4L^2\boldsymbol{\omega}^0}{T}$$

$$+ \left(\tau_{\max}^2\beta^2 + 20\eta^2K^2L^2e^2\right)\frac{1}{T}\sum_{t=0}^{T-1}\left(\mathbb{E}\left\|\mathcal{E}^t\right\|^2 + \mathbb{E}\left\|\nabla f\left(\boldsymbol{\theta}^t\right)\right\|^2\right).$$

Since $\boldsymbol{c}_i^0 = \frac{1}{K}\sum_{k=0}^{K-1}\nabla F\left(\boldsymbol{\theta}^0;\boldsymbol{\xi}_i^{-1,k}\right)$ for any $i$, $\boldsymbol{c}^0 = \frac{1}{N}\sum_i\boldsymbol{c}_i^0$, and $\boldsymbol{g}^0 = \boldsymbol{c}^0$, we have

$$\mathbb{E}\left\|\mathcal{E}^0\right\|^2 = \mathbb{E}\left\|\nabla f(\boldsymbol{\theta}^0) - \beta\left(\frac{1}{S}\sum_{i\in\mathcal{S}_0}\left(\boldsymbol{c}_i^1 - \boldsymbol{c}_i^0\right) + \boldsymbol{c}^0\right) - (1-\beta)\boldsymbol{g}^0\right\|^2$$

$$= \mathbb{E}\left\|\nabla f(\boldsymbol{\theta}^0) - \frac{1}{NK}\sum_{i,k}\nabla F\left(\boldsymbol{\theta}^0;\boldsymbol{\xi}_i^{-1,k}\right)\right.$$

$$\left.+ \frac{\beta}{SK}\sum_{i\in\mathcal{S}_0,k}\left(\nabla F\left(\boldsymbol{\theta}_i^0;\boldsymbol{\xi}_i^{-1,k}\right) \mp \nabla f_i\left(\boldsymbol{\theta}_i^0\right) \mp \nabla f_i\left(\boldsymbol{\theta}_i^{0,k}\right) - \nabla F\left(\boldsymbol{\theta}_i^{0,k};\boldsymbol{\xi}_i^{0,k}\right)\right)\right\|^2$$

$$\leq \frac{4\sigma^2}{NK} + \frac{8\beta^2\sigma^2}{SK} + \frac{\beta^2}{S}\sum_{i\in\mathcal{S}_0}\boldsymbol{\omega}_i^0.$$

From Lemma B.3, we know that, for any $i$,

$$\boldsymbol{\omega}_i^0 \leq 3\eta^2K^2e^{2\eta\beta KL}\left\|\nabla f\left(\boldsymbol{\theta}^0\right)\right\|^2 + \left(3e^{2\eta\beta KL}\left(\beta^2 + \frac{1}{N}\right) + \beta^2\right)K\eta^2\sigma^2 + (1+\eta\beta L)K^3\eta^3\beta^3L\sigma^2$$

$$\lesssim 3\eta^2K^2e^2\left\|\nabla f\left(\boldsymbol{\theta}^0\right)\right\|^2 + \frac{3e^2K\eta^2\sigma^2}{N}.$$

Thus, we have

$$\frac{1}{T}\sum_{t=0}^{T-1}\mathbb{E}\|\mathcal{E}^t\|^2 \lesssim \frac{1}{3T}\sum_{t=0}^{T-1}\mathbb{E}\left\|\nabla f(\boldsymbol{\theta}^t)\right\|^2 + \left(\frac{2}{\beta T} + 11\beta\right)\frac{8\sigma^2}{3SK} + \frac{800e^2\beta^2\sigma^2}{3K} + \frac{64L^2\sigma^2}{3NKT}$$

$$+ \frac{4}{3}\left(\tau_{\max}^2\beta^2 + 20\eta^2K^2L^2e^2\right)\frac{1}{T}\sum_{t=0}^{T-1}\left(\mathbb{E}\left\|\mathcal{E}^t\right\|^2 + \mathbb{E}\left\|\nabla f\left(\boldsymbol{\theta}^t\right)\right\|^2\right).$$

Denote by $a := \tau_{\max}^2\beta^2 + 20\eta^2K^2L^2e^2$, We require $a < \frac{1}{4}$. Then, we have

$$\frac{1}{T}\sum_{t=0}^{T-1}\mathbb{E}\|\mathcal{E}^t\|^2 \lesssim \frac{1+4a}{3-4a}\frac{1}{T}\sum_{t=0}^{T-1}\mathbb{E}\left\|\nabla f\left(\boldsymbol{\theta}^t\right)\right\|^2 + \left(\frac{2}{\beta T} + 11\beta\right)\frac{4\sigma^2}{SK} + \frac{400e^2\beta^2\sigma^2}{K} + \frac{16L^2\sigma^2}{NKT}.$$

where we use $3 - 4a \geq 2$. Plugging the above bound into (7) yields

$$\frac{1}{T}\sum_{t=0}^{T-1}\mathbb{E}\|\nabla f(\boldsymbol{\theta}^t)\|^2 \lesssim \frac{2\Delta}{\gamma(1-2\gamma L)T} + \frac{1+2\gamma L}{1-2\gamma L}\frac{1+4a}{3-4a}\frac{1}{T}\sum_{t=0}^{T-1}\mathbb{E}\left\|\nabla f\left(\boldsymbol{\theta}^t\right)\right\|^2$$

$$+ \frac{1+2\gamma L}{1-2\gamma L}\left(\left(\frac{2}{\beta T} + 11\beta\right)\frac{4\sigma^2}{SK} + \frac{400e^2\beta^2\sigma^2}{K} + \frac{16L^2\sigma^2}{NKT}\right).$$

$$\left(1 - \frac{(1+2\gamma L)(1+4a)}{(1-2\gamma L)(3-4a)}\right) \frac{1}{T} \sum_{t=0}^{T-1} \mathbb{E}\|\nabla f(\boldsymbol{\theta}^t)\|^2 \lesssim \frac{2\Delta}{\gamma(1-2\gamma L)T} + \frac{1+2\gamma L}{1-2\gamma L}\left(\frac{2}{\beta T}+11\beta\right)\frac{4\sigma^2}{SK}$$

$$+ \frac{1+2\gamma L}{1-2\gamma L}\left(\frac{400e^2\beta^2\sigma^2}{K} + \frac{16L^2\sigma^2}{NKT}\right).$$

When $2 - 8a - 8\gamma L \geq 0$ holds, let $\beta = \sqrt{\frac{SK}{T}}$ and $\gamma = \frac{\beta}{4L} = \frac{1}{4L}\sqrt{\frac{SK}{T}}$. Then, we have

$$\frac{1}{T} \sum_{t=0}^{T-1} \mathbb{E}\|\nabla f(\boldsymbol{\theta}^t)\|^2 \lesssim \frac{(3-4a)(4L\Delta+26\sigma^2)}{\left(1-4a-\sqrt{\frac{SK}{T}}\right)\sqrt{SKT}} + \frac{3-4a}{1-4a-\sqrt{\frac{SK}{T}}}\left(\frac{200e^2S\sigma^2}{T} + \frac{8L^2\sigma^2}{NKT}\right).$$

Note that we require $1 - 4a - \sqrt{\frac{SK}{T}} \geq 0$, i.e., $a := \frac{SK}{T}\tau_{\max}^2 + 20\eta^2 K^2 L^2 e^2 \leq \frac{1}{4}\left(1-\sqrt{\frac{SK}{T}}\right)$.

## D. Proof of Theorem 4.1

We first present the following inequalities that are frequently used in our analysis.

**Lemma D.1.** *For any $i, t$, we have*

$$\frac{1}{K} \sum_k \left\|\boldsymbol{\theta}_i^{t,k} - \boldsymbol{\theta}^{t-\tau_i^t}\right\|^2 \leq \frac{1}{3}\eta^2 K^2 \quad and \quad \frac{1}{K} \sum_k \left\|\boldsymbol{\theta}_i^{t,k} - \boldsymbol{\theta}^{t-\tau_i^t}\right\| \leq \frac{1}{2}\eta K.$$

*Proof.* From the update rule of local model, for any $i, k$ and $t$, we have

$$\left\|\boldsymbol{\theta}_i^{t,k+1} - \boldsymbol{\theta}_i^{t,k}\right\| = \eta \left\|\frac{\boldsymbol{g}_i^{t,k}}{\left\|\boldsymbol{g}_i^{t,k}\right\|}\right\| \leq \eta.$$

Then,

$$\left\|\boldsymbol{\theta}_i^{t,k} - \boldsymbol{\theta}^{t-\tau_i^t}\right\|^2 = \left\|\sum_{j=0}^{k-1}\left(\boldsymbol{\theta}_i^{t,j+1} - \boldsymbol{\theta}_i^{t,j}\right)\right\|^2 \leq k\sum_{j=0}^{k-1}\left\|\boldsymbol{\theta}_i^{t,j+1} - \boldsymbol{\theta}_i^{t,j}\right\|^2 \leq \eta^2 k^2.$$

Summing the above inequality over $i$ and $k$ yields

$$\frac{1}{K} \sum_k \left\|\boldsymbol{\theta}_i^{t,k} - \boldsymbol{\theta}^{t-\tau_i^t}\right\|^2 \leq \frac{\eta^2}{K}\sum_{k=0}^{K-1}k^2 \leq \frac{\eta^2}{6K}(K-1)K(2K-1) \leq \frac{1}{3}\eta^2 K^2.$$

Similarly, we have

$$\frac{1}{K} \sum_k \left\|\boldsymbol{\theta}_i^{t,k} - \boldsymbol{\theta}^{t-\tau_i^t}\right\| = \frac{1}{K} \sum_k \left(\left\|\boldsymbol{\theta}_i^{t,k} - \boldsymbol{\theta}^t\right\|^2\right)^{\frac{1}{2}} \leq \frac{\eta}{K}\sum_{k=0}^{K-1}k \leq \frac{1}{2}\eta K.$$

$\square$

Moreover, we have

$$\left\|\boldsymbol{\theta}^t - \boldsymbol{\theta}^{t-1}\right\| = \left\|\frac{\gamma}{SK}\sum_{i\in\mathcal{S}_t,k}\frac{\boldsymbol{g}_i^{t,k}}{\left\|\boldsymbol{g}_i^{t,k}\right\|}\right\| \leq \frac{\gamma}{SK}\sum_{i\in\mathcal{S}_t,k}\left\|\frac{\boldsymbol{g}_i^{t,k}}{\left\|\boldsymbol{g}_i^{t,k}\right\|}\right\| \leq \gamma,$$

and correspondingly $\|\boldsymbol{\theta}^t - \boldsymbol{\theta}^{t-\tau_i^t}\| \leq \sum_{\tau=t-\tau_i^t+1}^{t}\|\boldsymbol{\theta}^\tau - \boldsymbol{\theta}^{\tau-1}\| \leq \tau_i^t\gamma.$

From the $L$-smoothness of $f(\cdot)$ in Assumption 2.1, we have

$$
\begin{aligned}
& f(\boldsymbol{\theta}^{t+1}) - f(\boldsymbol{\theta}^t) \\
& \leq \nabla f(\boldsymbol{\theta}^t)^\top (\boldsymbol{\theta}^{t+1} - \boldsymbol{\theta}^t) + \frac{L}{2} \left\| \boldsymbol{\theta}^{t+1} - \boldsymbol{\theta}^t \right\|^2 \\
& = -\gamma \nabla f(\boldsymbol{\theta}^t)^\top \left( \frac{1}{SK} \sum_{i \in \mathcal{S}_{t,k}} \frac{\boldsymbol{g}_i^{t,k}}{\left\| \boldsymbol{g}_i^{t,k} \right\|} \right) + \frac{\gamma^2 L}{2} \\
& = -\gamma \left( \nabla f(\boldsymbol{\theta}^t) - \boldsymbol{g}^{t+1} \right)^\top \left( \frac{1}{SK} \sum_{i \in \mathcal{S}_{t,k}} \frac{\boldsymbol{g}_i^{t,k}}{\left\| \boldsymbol{g}_i^{t,k} \right\|} \right) - \gamma \left( \boldsymbol{g}^{t+1} \right)^\top \left( \frac{1}{SK} \sum_{i \in \mathcal{S}_{t,k}} \frac{\boldsymbol{g}_i^{t,k}}{\left\| \boldsymbol{g}_i^{t,k} \right\|} \right) + \frac{\gamma^2 L}{2} \\
& \leq \gamma \left\| \nabla f(\boldsymbol{\theta}^t) - \boldsymbol{g}^{t+1} \right\| - \gamma \left( \boldsymbol{g}^{t+1} \right)^\top \left( \frac{1}{SK} \sum_{i \in \mathcal{S}_{t,k}} \frac{\boldsymbol{g}_i^{t,k}}{\left\| \boldsymbol{g}_i^{t,k} \right\|} - \frac{\boldsymbol{g}^{t+1}}{\left\| \boldsymbol{g}^{t+1} \right\|} \right) - \gamma \left\| \boldsymbol{g}^{t+1} \right\| + \frac{\gamma^2 L}{2} \\
& \overset{(a)}{\leq} 2\gamma \left\| \nabla f(\boldsymbol{\theta}^t) - \boldsymbol{g}^{t+1} \right\| - \gamma \left\| \nabla f(\boldsymbol{\theta}^t) \right\| + \gamma \left\| \boldsymbol{g}^{t+1} \right\| \left\| \frac{1}{SK} \sum_{i \in \mathcal{S}_{t,k}} \frac{\boldsymbol{g}_i^{t,k}}{\left\| \boldsymbol{g}_i^{t,k} \right\|} - \frac{\boldsymbol{g}^{t+1}}{\left\| \boldsymbol{g}^{t+1} \right\|} \right\| + \frac{\gamma^2 L}{2} \\
& \overset{(b)}{\leq} 2\gamma \left\| \nabla f(\boldsymbol{\theta}^t) - \boldsymbol{g}^{t+1} \right\| - \gamma \left\| \nabla f(\boldsymbol{\theta}^t) \right\| + \frac{2\gamma}{SK} \sum_{i \in \mathcal{S}_{t,k}} \left\| \boldsymbol{g}_i^{t,k} - \boldsymbol{g}^{t+1} \right\| + \frac{\gamma^2 L}{2}.
\end{aligned}
$$

$$(10)$$

$$(11)$$

where $(a)$ is based on $\gamma \left\| \nabla f(\boldsymbol{\theta}^t) \right\| - \gamma \left\| \boldsymbol{g}^{t+1} \right\| \leq \gamma \left\| \nabla f(\boldsymbol{\theta}^t) - \boldsymbol{g}^{t+1} \right\|$ and $(b)$ is based on the following result.

$$
\begin{aligned}
& \left\| \boldsymbol{g}^{t+1} \right\| \left\| \frac{1}{SK} \sum_{i \in \mathcal{S}_{t,k}} \frac{\boldsymbol{g}_i^{t,k}}{\left\| \boldsymbol{g}_i^{t,k} \right\|} - \frac{\boldsymbol{g}^{t+1}}{\left\| \boldsymbol{g}^{t+1} \right\|} \right\| \\
& = \left\| \boldsymbol{g}^{t+1} \right\| \left\| \frac{1}{SK} \sum_{i \in \mathcal{S}_{t,k}} \left( \frac{\boldsymbol{g}_i^{t,k}}{\left\| \boldsymbol{g}_i^{t,k} \right\|} \mp \frac{\boldsymbol{g}_i^{t,k}}{\left\| \boldsymbol{g}^{t+1} \right\|} \right) - \frac{\boldsymbol{g}^{t+1}}{\left\| \boldsymbol{g}^{t+1} \right\|} \right\| \\
& \leq \frac{\left\| \boldsymbol{g}^{t+1} \right\|}{SK} \left\| \sum_{i \in \mathcal{S}_{t,k}} \frac{\boldsymbol{g}_i^{t,k}}{\left\| \boldsymbol{g}_i^{t,k} \right\|} - \frac{\boldsymbol{g}_i^{t,k}}{\left\| \boldsymbol{g}^{t+1} \right\|} \right\| + \left\| \boldsymbol{g}^{t+1} \right\| \left\| \frac{1}{SK} \sum_{i \in \mathcal{S}_{t,k}} \frac{\boldsymbol{g}_i^{t,k}}{\left\| \boldsymbol{g}^{t+1} \right\|} - \frac{\boldsymbol{g}^{t+1}}{\left\| \boldsymbol{g}^{t+1} \right\|} \right\| \\
& = \frac{\left\| \boldsymbol{g}^{t+1} \right\|}{SK} \left\| \sum_{i \in \mathcal{S}_{t,k}} \frac{\left\| \boldsymbol{g}^{t+1} \right\| - \left\| \boldsymbol{g}_i^{t,k} \right\|}{\left\| \boldsymbol{g}^{t+1} \right\| \left\| \boldsymbol{g}_i^{t,k} \right\|} \boldsymbol{g}_i^{t,k} \right\| + \left\| \frac{1}{SK} \sum_{i \in \mathcal{S}_{t,k}} \boldsymbol{g}_i^{t,k} - \boldsymbol{g}^{t+1} \right\| \\
& \leq \frac{\left\| \boldsymbol{g}^{t+1} \right\|}{SK} \sum_{i \in \mathcal{S}_{t,k}} \frac{\left| \left\| \boldsymbol{g}^{t+1} \right\| - \left\| \boldsymbol{g}_i^{t,k} \right\| \right|}{\left\| \boldsymbol{g}^t \right\| \left\| \boldsymbol{g}_i^{t,k} \right\|} \left\| \boldsymbol{g}_i^{t,k} \right\| + \frac{1}{SK} \sum_{i \in \mathcal{S}_{t,k}} \left\| \boldsymbol{g}_i^{t,k} - \boldsymbol{g}^{t+1} \right\| \\
& \leq \frac{2}{SK} \sum_{i \in \mathcal{S}_{t,k}} \left\| \boldsymbol{g}_i^{t,k} - \boldsymbol{g}^{t+1} \right\|.
\end{aligned}
$$

Summing up (11) over $t$ yields

$$
\begin{aligned}
\frac{1}{T} \sum_{t=0}^{T-1} \mathbb{E} \left\| \nabla f(\boldsymbol{\theta}^t) \right\| \leq & \frac{\Delta}{\gamma T} + \frac{2}{T} \sum_{t=0}^{T-1} \mathbb{E} \left\| \nabla f(\boldsymbol{\theta}^t) - \boldsymbol{g}^{t+1} \right\| \\
& + \frac{2}{SKT} \sum_{t=0}^{T-1} \mathbb{E} \left[ \sum_{i \in \mathcal{S}_{t,k}} \left\| \boldsymbol{g}_i^{t,k} - \boldsymbol{g}^{t+1} \right\| \right] + \frac{\gamma L}{2}.
\end{aligned}
$$

**D.1. Analysis of $\frac{1}{T}\sum_{t=0}^{T-1}\mathbb{E}\left\|\nabla f(\boldsymbol{\theta}^t)-\boldsymbol{g}^{t+1}\right\|$**

Define $\mathcal{E}^t := \nabla f(\boldsymbol{\theta}^t) - \boldsymbol{g}^{t+1}$. $\boldsymbol{u}^t := \nabla f\left(\boldsymbol{\theta}^t\right) - \nabla f\left(\boldsymbol{\theta}^{t-1}\right)$ and $\boldsymbol{v}^t := \left(\nabla f(\boldsymbol{\theta}^t) - \boldsymbol{c}^t - \frac{1}{S}\sum_{i\in\mathcal{S}_t}\left(\boldsymbol{c}_i^{t+1} - \boldsymbol{c}_i^t\right)\right)$. From the update rule of $\boldsymbol{g}^{t+1}$ given in Step 9 of Algorithm 3, we have

$$
\begin{aligned}
\mathcal{E}^t &= (1-\beta)\left(\nabla f(\boldsymbol{\theta}^t) - \boldsymbol{g}^t\right) + \beta\left(\nabla f(\boldsymbol{\theta}^t) - \boldsymbol{c}^t - \frac{1}{S}\sum_{i\in\mathcal{S}_t}\left(\boldsymbol{c}_i^{t+1} - \boldsymbol{c}_i^t\right)\right)\\
&= (1-\beta)\mathcal{E}^{t-1} + (1-\beta)\boldsymbol{u}^t + \beta\boldsymbol{v}^t\\
&= (1-\beta)^t\mathcal{E}^0 + \sum_{d=1}^t \boldsymbol{u}^d(1-\beta)^{t+1-d} + \sum_{d=1}^t \beta\boldsymbol{v}^d(1-\beta)^{t-d}.
\end{aligned}
$$

Based on the triangle inequality of $\ell_2$ norm and the concavity of the square root $(\cdot)^{\frac{1}{2}}$, we have

$$
\mathbb{E}\left\|\mathcal{E}^t\right\| \leq (1-\beta)^t\mathbb{E}\left\|\mathcal{E}^0\right\| + \sum_{d=1}^t \mathbb{E}\left\|\boldsymbol{u}^d\right\|(1-\beta)^{t+1-d} + \left(\mathbb{E}\left\|\sum_{d=1}^t \beta\boldsymbol{v}^d(1-\beta)^{t-d}\right\|^2\right)^{\frac{1}{2}}. \tag{12}
$$

Since $\boldsymbol{c}_i^0 = \frac{1}{K}\sum_{k=0}^{K-1}\nabla F\left(\boldsymbol{\theta}^0;\boldsymbol{\xi}_i^{-1,k}\right)$ for any $i$, $\boldsymbol{c}^0 = \frac{1}{N}\sum_i \boldsymbol{c}_i^0$, and $\boldsymbol{g}^0 = \boldsymbol{c}^0$, we have

$$
\begin{aligned}
\mathbb{E}\left\|\mathcal{E}^0\right\| &= \mathbb{E}\left\|\nabla f(\boldsymbol{\theta}^0) - \frac{1}{NK}\sum_{i,k}\nabla F\left(\boldsymbol{\theta}^0;\boldsymbol{\xi}_i^{-1,k}\right) + \frac{\beta}{SK}\sum_{i\in\mathcal{S}_0,k}\left(\nabla F\left(\boldsymbol{\theta}_i^0;\boldsymbol{\xi}_i^{-1,k}\right) - \nabla F\left(\boldsymbol{\theta}_i^{0,k};\boldsymbol{\xi}_i^{0,k}\right)\right)\right\|\\
&\leq \frac{\sigma}{\sqrt{NK}} + \mathbb{E}\left\|\frac{\beta}{SK}\sum_{i\in\mathcal{S}_0,k}\left(\nabla F\left(\boldsymbol{\theta}_i^0;\boldsymbol{\xi}_i^{-1,k}\right) \mp \nabla f_i\left(\boldsymbol{\theta}_i^0\right) \mp \nabla f_i\left(\boldsymbol{\theta}_i^{0,k}\right) - \nabla F\left(\boldsymbol{\theta}_i^{0,k};\boldsymbol{\xi}_i^{0,k}\right)\right)\right\|\\
&\leq \frac{\sigma}{\sqrt{NK}} + \frac{\beta\sigma}{\sqrt{SK}} + \frac{\beta}{SK}\mathbb{E}\left[\sum_{i\in\mathcal{S}_0,k}\left\|\nabla f_i\left(\boldsymbol{\theta}^0\right) - \nabla f_i\left(\boldsymbol{\theta}_i^{0,k}\right)\right\|\right] + \frac{\beta\sigma}{\sqrt{SK}}\\
&\leq \frac{\beta L}{NK}\sum_{i,k}\mathbb{E}\left\|\boldsymbol{\theta}_i^{0,k} - \boldsymbol{\theta}^0\right\| + \frac{3\sigma}{\sqrt{SK}}\\
&\leq \frac{1}{2}\eta\beta KL + \frac{3\sigma}{\sqrt{SK}}. \tag{13}
\end{aligned}
$$

Additionally, for any $t$, we have

$$
\left\|\boldsymbol{u}^t\right\| = \left\|\nabla f(\boldsymbol{\theta}^{t+1}) - \nabla f(\boldsymbol{\theta}^t)\right\| \leq L\left\|\boldsymbol{\theta}^{t+1} - \boldsymbol{\theta}^t\right\| \leq \gamma L\left\|\frac{1}{SK}\sum_{i\in\mathcal{S}_t,k}\frac{\boldsymbol{g}_i^{t,k}}{\|\boldsymbol{g}_i^{t,k}\|}\right\| \leq \gamma L. \tag{14}
$$

Further, we have

$$
\begin{aligned}
\mathbb{E}\left\|\sum_{d=1}^t \beta\boldsymbol{v}^d(1-\beta)^{t-d}\right\|^2 &= \sum_{d=1}^t \beta^2\mathbb{E}\left\|\boldsymbol{v}^d\right\|^2(1-\beta)^{2(t-d)}\\
&\quad + \sum_{1\leq d_1,d_2\leq t, d_1\neq d_2}\mathbb{E}\left\langle\beta\boldsymbol{v}^{d_1}(1-\beta)^{t-d_1}, \beta\boldsymbol{v}^{d_2}(1-\beta)^{t-d_2}\right\rangle.
\end{aligned}
$$

Since $\boldsymbol{c}_i^{t+1} = \frac{1}{K}\sum_{k=1}^{K}\nabla F\left(\boldsymbol{\theta}_i^{t,k};\boldsymbol{\xi}_i^{t,k}\right)$ for $i \in \mathcal{S}_t$ and $\boldsymbol{c}^t = \frac{1}{N}\sum_{i=1}^{N}\boldsymbol{c}_i^t$, for any $t$, we have

$$\mathbb{E}[\boldsymbol{v}^t|\mathcal{F}^t] = \mathbb{E}_{\left\{\boldsymbol{\xi}_i^{t,k}\right\}_{\forall i,k},\mathcal{S}_t}[\boldsymbol{v}^t]$$

$$= \mathbb{E}_{\left\{\boldsymbol{\xi}_i^{t,k}\right\}_{\forall i,k}}\left[\nabla f(\boldsymbol{\theta}^t) - \boldsymbol{c}^t - \frac{1}{NK}\sum_{i,k}\left(\nabla F\left(\boldsymbol{\theta}_i^{t,k};\boldsymbol{\xi}_i^{t,k}\right) - \boldsymbol{c}_i^t\right)\right]$$

$$= \nabla f(\boldsymbol{\theta}^t) - \frac{1}{NK}\sum_{i,k}\nabla f_i\left(\boldsymbol{\theta}_i^{t,k}\right) := \boldsymbol{\psi}^t.$$

$$\mathbb{E}\left\|\boldsymbol{\psi}^t\right\| = \frac{1}{N}\sum_i\mathbb{E}\left\|\nabla f(\boldsymbol{\theta}^t) \mp \nabla f\left(\boldsymbol{\theta}^{t-\tau_i^t}\right) - \frac{1}{NK}\sum_{i,k}\nabla f_i\left(\boldsymbol{\theta}_i^{t,k}\right)\right\|$$

$$\leq \frac{L}{N}\sum_i\mathbb{E}\left\|\boldsymbol{\theta}^t - \boldsymbol{\theta}^{t-\tau_i^t}\right\| + \frac{L}{NK}\sum_{i,k}\mathbb{E}\left\|\boldsymbol{\theta}^{t-\tau_i^t} - \boldsymbol{\theta}_i^{t,k}\right\|$$

$$\leq \frac{\gamma L}{N}\sum_i\tau_i^t + \frac{1}{2}\eta KL$$

$$\leq \gamma\tau_{\max}L + \frac{1}{2}\eta KL.$$

We also have

$$\mathbb{E}\left\|\nabla f(\boldsymbol{\theta}^t) - \frac{1}{SK}\sum_{i\in\mathcal{S}_t,k}\nabla f_i\left(\boldsymbol{\theta}_i^{t,k}\right)\right\|^2$$

$$\leq \frac{2}{S}\sum_{i\in\mathcal{S}_t}\mathbb{E}\left\|\nabla f(\boldsymbol{\theta}^t) - \nabla f\left(\boldsymbol{\theta}^{t-\tau_i^t}\right)\right\|^2 + \frac{2L^2}{SK}\sum_{i\in\mathcal{S}_t,k}\mathbb{E}\left\|\boldsymbol{\theta}^{t-\tau_i^t} - \boldsymbol{\theta}_i^{t,k}\right\|^2$$

$$\leq 2\gamma^2\tau_{\max}^2 L^2 + \eta^2 K^2 L^2.$$

Similarly, we have $\mathbb{E}\left\|\boldsymbol{\psi}^t\right\|^2 = \mathbb{E}\left\|\nabla f(\boldsymbol{\theta}^t) - \frac{1}{NK}\sum_{i,k}\nabla f_i\left(\boldsymbol{\theta}_i^{t,k}\right)\right\|^2 \leq 2\gamma^2\tau_{\max}^2 L^2 + \eta^2 K^2 L^2$.

For any $0 \leq t_1 < t_2 \leq T-1$, we have

$$\mathbb{E}\left\langle\boldsymbol{v}^{t_1},\boldsymbol{v}^{t_2}\right\rangle = \mathbb{E}\left\langle\boldsymbol{v}^{t_1},\mathbb{E}\left[\boldsymbol{v}^{t_2}|\mathcal{F}^{t_2}\right]\right\rangle$$

$$= \mathbb{E}\left\langle\nabla f\left(\boldsymbol{\theta}^{t_1}\right) - \frac{1}{SK}\sum_{i\in\mathcal{S}_t,k}\nabla f_i\left(\boldsymbol{\theta}_i^{t_1,k}\right),\boldsymbol{\psi}^{t_2}\right\rangle + \mathbb{E}\left\langle\mathbb{E}_{\mathcal{S}_t}\left[\boldsymbol{c}^t - \frac{1}{S}\sum_{i\in\mathcal{S}_t}\boldsymbol{c}_i^t\right],\boldsymbol{\psi}^{t_2}\right\rangle$$

$$+ \mathbb{E}\left\langle\frac{1}{SK}\sum_{i\in\mathcal{S}_t,k}\left(\nabla f_i\left(\boldsymbol{\theta}_i^{t_1,k}\right) - \nabla F\left(\boldsymbol{\theta}_i^{t_1,k};\boldsymbol{\xi}_i^{t_1,k}\right)\right),\boldsymbol{\psi}^{t_2}\right\rangle$$

$$\leq \frac{1}{2}\mathbb{E}\left\|\nabla f\left(\boldsymbol{\theta}^{t_1}\right) - \frac{1}{SK}\sum_{i\in\mathcal{S}_{t_1},k}\nabla f_i\left(\boldsymbol{\theta}_i^{t_1,k}\right)\right\|^2 + \frac{1}{2}\mathbb{E}\left\|\boldsymbol{\psi}^{t_2}\right\|^2 + \frac{\sigma}{\sqrt{SK}}\mathbb{E}\left\|\boldsymbol{\psi}^{t_2}\right\|$$

$$\leq 2\gamma^2\tau_{\max}^2 L^2 + \eta^2 K^2 L^2 + \frac{L\sigma}{\sqrt{SK}}\left(\gamma\tau_{\max} + \frac{1}{2}\eta K\right).$$

Further, based on Lemma B.1, we have

$$
\begin{aligned}
\mathbb{E}\left\|\boldsymbol{v}^t\right\|^2 =& \mathbb{E}\left\|\nabla f(\boldsymbol{\theta}^t) - \boldsymbol{c}^t - \frac{1}{S}\sum_{i\in\mathcal{S}_t}\left(\boldsymbol{c}_i^{t+1} - \boldsymbol{c}_i^t\right)\right\|^2 \\
\leq& \mathbb{E}\left\|\nabla f(\boldsymbol{\theta}^t) - \frac{1}{N}\sum_i \boldsymbol{c}_i^{t+1}\right\|^2 + \frac{1}{S}\frac{1}{N}\sum_i \mathbb{E}\left\|(\boldsymbol{c}_i^{t+1} - \boldsymbol{c}_i^t) - \frac{1}{N}\sum_i\left(\boldsymbol{c}_i^{t+1} - \boldsymbol{c}_i^t\right)\right\|^2 \\
\leq& \mathbb{E}\left\|\nabla f(\boldsymbol{\theta}^t) - \frac{1}{N}\sum_i \boldsymbol{c}_i^{t+1}\right\|^2 + \frac{1}{S}\frac{1}{N}\sum_i \mathbb{E}\left\|\boldsymbol{c}_i^{t+1} - \boldsymbol{c}_i^t\right\|^2.
\end{aligned}
$$

Note that $\boldsymbol{c}_i^{t+1} = \frac{1}{K}\sum_k \nabla F\left(\boldsymbol{\theta}_i^{t,k};\boldsymbol{\xi}_i^{t,k}\right)$, i.e., client $i$ is selected in the $t$th round. We have

$$
\begin{aligned}
\mathbb{E}&\left\|\nabla f(\boldsymbol{\theta}^t) - \frac{1}{NK}\sum_{i,k}\nabla F\left(\boldsymbol{\theta}_i^{t,k};\boldsymbol{\xi}_i^{t,k}\right)\right\|^2 \\
=& \mathbb{E}\left\|\frac{1}{NK}\sum_{i,k}\left(\nabla f_i\left(\boldsymbol{\theta}^t\right) \mp \nabla f_i\left(\boldsymbol{\theta}^{t-\tau_i^t}\right) \mp \nabla f_i\left(\boldsymbol{\theta}_i^{t,k}\right) - \nabla F\left(\boldsymbol{\theta}_i^{t,k};\boldsymbol{\xi}_i^{t,k}\right)\right)\right\|^2 \\
\leq& 3L^2\mathbb{E}\left\|\boldsymbol{\theta}^t - \boldsymbol{\theta}^{t-\tau_{\max}}\right\|^2 + \frac{3L^2}{NK}\sum_{i,k}\mathbb{E}\left\|\boldsymbol{\theta}_i^{t,k} - \boldsymbol{\theta}^{t-\tau_i^t}\right\|^2 + \frac{3\sigma^2}{NK} \\
\leq& 3\gamma^2\tau_{\max}^2 L^2 + \eta^2 K^2 L^2 + \frac{3\sigma^2}{NK}, \forall t.
\end{aligned}
$$

From Lemma D.2, we know that

$$
\mathbb{E}\left\|\boldsymbol{c}_i^{t+1} - \boldsymbol{c}_i^t\right\|^2 \leq \frac{28\sigma^2}{K} + \frac{28}{3}\eta^2 K^2 L^2 + 4\gamma^2 L^2\left(1 + \frac{4N^2}{S^2}\right).
$$

Thus, we have

$$
\begin{aligned}
\mathbb{E}\|\boldsymbol{v}^t\|^2 \leq& 3\gamma^2\tau_{\max}^2 L^2 + \eta^2 K^2 L^2 + \frac{3\sigma^2}{NK} + \frac{28\sigma^2}{SK} + \frac{28}{3S}\eta^2 K^2 L^2 + \frac{4\gamma^2 L^2}{S}\left(1 + \frac{4N^2}{S^2}\right) \\
\leq& \gamma^2 L^2\left(3\tau_{\max}^2 + \frac{4}{S} + \frac{4N^3}{S^3}\right) + \frac{31\sigma^2}{SK} + \left(1 + \frac{28}{3S}\right)\eta^2 K^2 L^2.
\end{aligned}
$$

Then, we have

$$
\begin{aligned}
\mathbb{E}\left\|\sum_{d=1}^t \beta\boldsymbol{v}^d(1-\beta)^{t-d}\right\|^2 \leq& \beta\mathbb{E}\|\boldsymbol{v}^d\|^2 + \left\langle\boldsymbol{v}^{d_1},\boldsymbol{v}^{d_2}\right\rangle \\
\leq& \beta\left(\gamma^2 L^2\left(3\tau_{\max}^2 + \frac{4}{S} + \frac{4N^3}{S^3}\right) + \frac{31\sigma^2}{SK} + \left(1 + \frac{28}{3S}\right)\eta^2 K^2 L^2\right) \\
& + 2\gamma^2\tau_{\max}^2 L^2 + \eta^2 K^2 L^2 + \frac{L\sigma}{\sqrt{SK}}\left(\gamma\tau_{\max} + \frac{1}{2}\eta K\right).
\end{aligned}
$$

$$
\begin{aligned}
\left(\mathbb{E}\left\|\sum_{d=1}^t \beta\boldsymbol{v}^d(1-\beta)^{t-d}\right\|^2\right)^{\frac{1}{2}} \leq& \gamma L\left((\sqrt{2} + \sqrt{3\beta})\tau_{\max} + 2\sqrt{\frac{\beta}{S}}\left(1 + \frac{N\sqrt{N}}{S}\right)\right) + \sqrt{\frac{31\beta\sigma^2}{SK}} \\
& + \eta KL\left(1 + \sqrt{\beta} + \sqrt{\frac{28\beta}{3S}}\right) + \sqrt{\frac{L\sigma(2\gamma\tau_{\max} + \eta K)}{2\sqrt{SK}}}. \quad (15)
\end{aligned}
$$

Plugging (13), (14) and (15) into (12), we have

$$
\begin{aligned}
\mathbb{E}\left\|\mathcal{E}^t\right\| \leq & (1-\beta)^t \left(\frac{1}{2}\eta\beta KL + \frac{3\sigma}{\sqrt{SK}}\right) + \frac{\gamma L}{\beta} + \gamma L\left((\sqrt{2}+\sqrt{3\beta})\tau_{\max} + 2\sqrt{\frac{\beta}{S}}\left(1+\frac{N\sqrt{N}}{S}\right)\right) \\
& + \sqrt{\frac{31\beta\sigma^2}{SK}} + \eta KL\left(1+\sqrt{\beta}+\sqrt{\frac{28\beta}{3S}}\right) + \sqrt{\frac{L\sigma(2\gamma\tau_{\max}+\eta K)}{2\sqrt{SK}}}.
\end{aligned}
\tag{16}
$$

Summing up the above inequality over $t$ yields

$$
\begin{aligned}
\frac{1}{T}\sum_{t=0}^{T-1}\mathbb{E}\left\|\mathcal{E}^t\right\| \leq & \frac{\eta KL}{2T} + \frac{3\sigma}{\beta T\sqrt{SK}} + \frac{\gamma L}{\beta} + \gamma L\left((\sqrt{2}+\sqrt{3\beta})\tau_{\max} + 2\sqrt{\frac{\beta}{S}}\left(1+\frac{N\sqrt{N}}{S}\right)\right) \\
& + \sqrt{\frac{31\beta\sigma^2}{SK}} + \eta KL\left(1+\sqrt{\beta}+\sqrt{\frac{28\beta}{3S}}\right) + \sqrt{\frac{L\sigma(2\gamma\tau_{\max}+\eta K)}{2\sqrt{SK}}}.
\end{aligned}
$$

### D.2. Analysis of $\frac{1}{SKT}\mathbb{E}\left[\sum_{i\in\mathcal{S}_t,k,t}\left\|g_i^{t,k}-g^{t+1}\right\|\right]$

Recall that $g_i^{t,k}$ and $g^{t+1}$ are respectively updated by $g_i^{t,k} = \beta\left(\nabla F\left(\theta_i^{t,k};\xi_i^{t,k}\right) - \tilde{c}_i^t + c^{t-\tau_i^t}\right) + (1-\beta)g^{t-\tau_i^t}$ and $g^{t+1} = \beta\left(\frac{1}{S}\sum_{i\in\mathcal{S}_t}\left(c_i^{t+1}-c_i^t\right)+c^t\right)+(1-\beta)g^t$. Then, we have

$$
\begin{aligned}
g_i^{t,k}-g^{t+1} = & \beta\left(\nabla F\left(\theta_i^{t,k};\xi_i^{t,k}\right)-\tilde{c}_i^t\right) - \frac{\beta}{S}\sum_{i\in\mathcal{S}_t}\left(c_i^{t+1}-c_i^t\right) \\
& + \beta\left(c^{t-\tau_i^t}-c^t\right) + (1-\beta)\left(g^{t-\tau_i^t}-g^t\right).
\end{aligned}
$$

$$
\begin{aligned}
\left\|g_i^{t,k}-g^{t+1}\right\| \leq & \beta\left\|\nabla F\left(\theta_i^{t,k};\xi_i^{t,k}\right)-\tilde{c}_i^t\right\| + \frac{\beta}{S}\sum_{i\in\mathcal{S}_t}\left\|c_i^{t+1}-c_i^t\right\| \\
& + \beta\left\|c^{t-\tau_i^t}-c^t\right\| + (1-\beta)\left\|g^{t-\tau_i^t}-g^t\right\| \\
\leq & \beta\left\|\nabla F\left(\theta_i^{t,k};\xi_i^{t,k}\right)-\tilde{c}_i^t\right\| + \frac{\beta}{S}\sum_{i\in\mathcal{S}_t}\left\|c_i^{t+1}-c_i^t\right\| \\
& + \frac{\beta}{N}\sum_i\sum_{d=t-\tau_i^t}^{t-1}\left\|c_i^{d+1}-c_i^d\right\| + \sum_{d=t-\tau_i^t}^{t-1}\left\|g^{d+1}-g^d\right\|.
\end{aligned}
\tag{17}
$$

For any $i,t$, we have

$$
\begin{aligned}
& \frac{1}{K}\sum_k\mathbb{E}\left\|\nabla F\left(\theta_i^{t,k};\xi_i^{t,k}\right)-\tilde{c}_i^t\right\| \\
= & \frac{1}{K}\sum_k\mathbb{E}\left\|\nabla F\left(\theta_i^{t,k};\xi_i^{t,k}\right)\mp f_i\left(\theta_i^{t,k}\right)\mp f_i\left(\theta^{t-\tau_i^t}\right)\mp f_i\left(\theta^{t-\tau_i^t-1-\tau_i^{t-\tau_i^t-1}}\right)\right. \\
& \left.\mp\frac{1}{K}\sum_k\nabla F\left(\theta_i^{t-\tau_i^t-1,k}\right)-\frac{1}{K}\sum_k\nabla F\left(\theta_i^{t-\tau_i^t-1,k};\xi_i^{t-\tau_i^t-1,k}\right)\right\| \\
\leq & \sigma + \frac{L}{K}\sum_k\left\|\theta_i^{t,k}-\theta^{t-\tau_i^t}\right\| + (\tau_{\max}+1)\gamma L + \frac{L}{K}\sum_k\left\|\theta_i^{t-\tau_i^t-1,k}-\theta^{t-\tau_i^t-1-\tau_i^{t-\tau_i^t-1}}\right\| + \frac{\sigma}{\sqrt{K}} \\
\leq & 2\sigma + \eta KL + (\tau_{\max}+1)\gamma L.
\end{aligned}
\tag{18}
$$

Moreover, we have

$$\mathbb{E}\left\|\boldsymbol{g}^{t+1} - \boldsymbol{g}^t\right\| = \beta \mathbb{E}\left\|\frac{1}{S}\sum_{i \in \mathcal{S}_t}\left(\boldsymbol{c}_i^{t+1} - \boldsymbol{c}_i^t\right) + \boldsymbol{c}^t \mp \nabla f\left(\boldsymbol{\theta}^{t-1}\right) - \boldsymbol{g}^t\right\|$$

$$\leq \frac{\beta}{S}\sum_{i \in \mathcal{S}_t}\mathbb{E}\left\|\boldsymbol{c}_i^{t+1} - \boldsymbol{c}_i^t\right\| + \beta\mathbb{E}\left\|\nabla f\left(\boldsymbol{\theta}^{t-1}\right) - \boldsymbol{c}^t\right\| + \beta\mathbb{E}\|\mathcal{E}^{t-1}\|. \tag{19}$$

From Lemma D.2, for any $t, i \in \mathcal{S}_t$, we have

$$\mathbb{E}\left\|\boldsymbol{c}_i^{t+1} - \boldsymbol{c}_i^t\right\| \leq 2\sigma\sqrt{\frac{7}{K}} + 2\sqrt{\frac{7}{3}}\eta KL + 2\gamma L\left(1 + \frac{2N}{S}\right) \tag{20}$$

$$\mathbb{E}\left\|\nabla f\left(\boldsymbol{\theta}^t\right) - \boldsymbol{c}^{t+1}\right\| \leq 2\sigma\sqrt{\frac{6}{K}} + 2\gamma L\left(\sum_i\frac{\tau_i^t}{N} + \frac{2N}{S}\right) + 2\sqrt{2}\eta KL. \tag{21}$$

Plugging (18), (19), (20), and (21) into (17), we have

$$\frac{1}{K}\sum_k\mathbb{E}\left\|\boldsymbol{g}_i^{t,k} - \boldsymbol{g}^{t+1}\right\| \leq \beta(2\sigma + \eta KL + \gamma(\tau_{\max} + 1)L)$$

$$+ (1 + 2\tau_i^t)\beta\left(2\sqrt{\frac{7}{K}}\sigma + 2\sqrt{\frac{7}{3}}\eta KL + 2\gamma L\left(1 + \frac{2N}{S}\right)\right)$$

$$+ \beta\tau_i^t\left(2\sigma\sqrt{\frac{6}{K}} + 2\gamma L\left(\sum_i\frac{\tau_i^t}{N} + \frac{2N}{S}\right) + 2\sqrt{2}\eta KL\right) + \beta\tau_i^t\mathbb{E}\|\mathcal{E}^{t-1}\|.$$

$$\frac{1}{SKT}\mathbb{E}\left[\sum_{i \in \mathcal{S}_t, k, t}\left\|\boldsymbol{g}_i^{t,k} - \boldsymbol{g}^{t+1}\right\|\right]$$

$$\leq \beta(2\sigma + \eta KL + \gamma(\tau_{\max} + 1)L) + (1 + 2\bar{\tau})\beta\left(2\sqrt{\frac{7}{K}}\sigma + 2\sqrt{\frac{7}{3}}\eta KL + 2\gamma L\left(1 + \frac{2N}{S}\right)\right)$$

$$+ \beta\bar{\tau}\left(2\sigma\sqrt{\frac{6}{K}} + 2\gamma L\left(\tau_{\max} + \frac{2N}{S}\right) + 2\sqrt{2}\eta KL\right) + \frac{\beta}{NT}\sum_{i,t}\tau_i^t\mathbb{E}\|\mathcal{E}^{t-1}\|.$$

where $\bar{\tau} := \frac{1}{NT}\sum_{i,t}\tau_i^t$.

From (16), we have

$$\frac{\beta}{NT}\sum_{i,t}\tau_i^t\mathbb{E}\left\|\mathcal{E}^{t-1}\right\| \leq \frac{\beta\tau_{\max}}{T}\sum_t(1 - \beta)^t\left(\frac{1}{2}\eta\beta KL + \frac{3\sigma}{\sqrt{SK}}\right) + \beta\gamma\bar{\tau}L\left((\sqrt{2} + \sqrt{3\beta})\tau_{\max} + 2\sqrt{\frac{\beta}{S}}\left(1 + \frac{N\sqrt{N}}{S}\right)\right)$$

$$+ \gamma\bar{\tau}L + \beta\bar{\tau}\sqrt{\frac{31\beta\sigma^2}{SK}} + \beta\eta\bar{\tau}KL\left(1 + \sqrt{\beta} + \sqrt{\frac{28\beta}{3S}}\right) + \beta\bar{\tau}\sqrt{\frac{L\sigma(2\gamma\tau_{\max} + \eta K)}{2\sqrt{SK}}}$$

$$\leq \frac{\tau_{\max}}{2}\beta\eta KL + \frac{3\tau_{\max}\sigma}{T\sqrt{SK}} + \gamma\beta\bar{\tau}L\left((\sqrt{2} + \sqrt{3\beta})\tau_{\max} + 2\sqrt{\frac{\beta}{S}}\left(1 + \frac{N\sqrt{N}}{S}\right)\right)$$

$$+ \gamma\bar{\tau}L + \beta\bar{\tau}\sqrt{\frac{31\beta\sigma^2}{SK}} + \beta\eta\bar{\tau}KL\left(1 + \sqrt{\beta} + \sqrt{\frac{28\beta}{3S}}\right) + \beta\bar{\tau}\sqrt{\frac{L\sigma(2\gamma\tau_{\max} + \eta K)}{2\sqrt{SK}}}.$$

### D.3. Combination

Let $\beta = \sqrt{\frac{SK}{T}}$, $\gamma = \frac{(SK)^{\frac{1}{4}}}{T^{\frac{3}{4}}}$ and $\eta = \frac{1}{K\sqrt{T}}$, we have

$$\frac{1}{T}\sum_{t=0}^{T-1}\mathbb{E}\left\|\mathcal{E}^t\right\| \lesssim \frac{\gamma L}{\beta} + \sqrt{2}\tau_{\max}\gamma L + \sqrt{\frac{31\beta\sigma^2}{SK}} + \sqrt{\frac{L\sigma(2\gamma\tau_{\max}+\eta K)}{2\sqrt{SK}}}$$

$$\lesssim \frac{L}{(SKT)^{\frac{1}{4}}} + \frac{\tau_{\max}L(SK)^{\frac{1}{4}}}{T^{\frac{3}{4}}} + \frac{\sigma}{(SKT)^{\frac{1}{4}}} + \frac{\sqrt{L\sigma\tau_{\max}}}{T^{\frac{3}{8}}(SK)^{\frac{1}{8}}} + \frac{\sqrt{L\sigma}}{(SKT)^{\frac{1}{4}}}.$$

Similarly, we have

$$\frac{1}{SKT}\mathbb{E}\left[\sum_{i\in\mathcal{S}_t,k,t}\left\|\boldsymbol{g}_i^{t,k}-\boldsymbol{g}^t\right\|\right]$$

$$\lesssim \beta\sigma\left(1+\frac{\bar{\tau}}{\sqrt{K}}\right) + \frac{\beta}{NT}\sum_{i,t}\tau_i^t\mathbb{E}\left\|\mathcal{E}^t\right\|$$

$$\lesssim \left(1+\frac{\bar{\tau}}{\sqrt{K}}\right)\sigma\sqrt{\frac{SK}{T}} + \bar{\tau}\tau_{\max}L\frac{(SK)^{\frac{3}{4}}}{T^{\frac{5}{4}}} + \left(\sigma+\sqrt{L\sigma}\right)\frac{\bar{\tau}(SK)^{\frac{1}{4}}}{T^{\frac{3}{4}}} + \bar{\tau}\sqrt{L\sigma\tau_{\max}}\frac{(SK)^{\frac{3}{8}}}{T^{\frac{7}{8}}}.$$

Then, we have

$$\frac{1}{T}\sum_{t=0}^{T-1}\mathbb{E}\left\|\nabla f(\boldsymbol{\theta}^t)\right\| \leq \frac{\Delta}{\gamma T} + \frac{2}{T}\sum_{t=0}^{T-1}\mathbb{E}\left\|\mathcal{E}^t\right\| + \frac{2}{SKT}\sum_{t=0}^{T-1}\mathbb{E}\left[\sum_{i\in\mathcal{S}_t,k}\left\|\boldsymbol{g}_i^{t,k}-\boldsymbol{g}^{t+1}\right\|\right] + \frac{\gamma L}{2}$$

$$\lesssim \frac{\Delta+L+\sigma+\sqrt{L\sigma}}{(SKT)^{\frac{1}{4}}} + \sigma\sqrt{\frac{SK}{T}} + \frac{\sqrt{L\sigma\tau_{\max}}}{T^{\frac{3}{8}}(SK)^{\frac{1}{8}}} + \bar{\tau}\sigma\sqrt{\frac{S}{T}}$$

$$+ \tau_{\max}L\frac{(SK)^{\frac{1}{4}}}{T^{\frac{3}{4}}} + \bar{\tau}\sqrt{L\sigma\tau_{\max}}\frac{(SK)^{\frac{3}{8}}}{T^{\frac{7}{8}}} + \bar{\tau}\tau_{\max}L\frac{(SK)^{\frac{3}{4}}}{T^{\frac{5}{4}}}.$$

**Lemma D.2.** *Assume that the delay of client $i$ at the $t$th round is $\tau_i^t$. For any $t, i \in \mathcal{S}_t$, we have*

*i)* $\mathbb{E}\left\|\boldsymbol{c}_i^{t+1}-\boldsymbol{c}_i^t\right\|^2 \leq \frac{28\sigma^2}{K} + \frac{28}{3}\eta^2 K^2 L^2 + 4\gamma^2 L^2\left(1+\frac{4N^2}{S^2}\right);$

*ii)* $\mathbb{E}\left\|\boldsymbol{c}_i^{t+1}-\boldsymbol{c}_i^t\right\| \leq 2\sqrt{\frac{7}{K}}\sigma + 2\sqrt{\frac{7}{3}}\eta KL + 2\gamma L\left(1+\frac{2N}{S}\right);$

*iii)* $\mathbb{E}\left\|\nabla f\left(\boldsymbol{\theta}^t\right)-\boldsymbol{c}^{t+1}\right\| \leq 2\sigma\sqrt{\frac{6}{K}} + 2\gamma L\left(\sum_i\frac{\tau_i^t}{N}+\frac{2N}{S}\right) + 2\sqrt{2}\eta KL.$

*Proof.* Define $\phi_i^t := \mathbb{E}\left\|\nabla f_i\left(\boldsymbol{\theta}^{t-\tau_i^t}\right)-\boldsymbol{c}_i^{t+1}\right\|^2$. Since for any $t$, the $S$ elements in $\mathcal{S}_t$ are uniformly sampled from $\{1,\cdots,N\}$, we have

$$\boldsymbol{c}_i^{t+1} = \begin{cases} \boldsymbol{c}_i^t & \text{if } i\in\mathcal{S}_t \text{ (w.p. } 1-\frac{S}{N}) \\ \frac{1}{K}\sum_k\nabla F\left(\boldsymbol{\theta}_i^{t,k};\boldsymbol{\xi}_i^{t,k}\right) & \text{if } i\notin\mathcal{S}_t \text{ (w.p. } \frac{S}{N}). \end{cases}$$

Using Young's inequality repeatedly, we have

$$
\begin{aligned}
\phi_i^t &= \left(1 - \frac{S}{N}\right) \mathbb{E} \left\| \nabla f_i \left(\boldsymbol{\theta}^{t-\tau_i^t}\right) - \boldsymbol{c}_i^t \right\|^2 + \frac{S}{N} \mathbb{E} \left\| \frac{1}{K} \sum_k \left(\nabla f_i \left(\boldsymbol{\theta}^{t-\tau_i^t}\right) - \nabla F \left(\boldsymbol{\theta}_i^{t,k}; \boldsymbol{\xi}_i^{t,k}\right)\right) \right\|^2 \\
&\leq \left(1 - \frac{S}{N}\right) \mathbb{E} \left\| \nabla f_i \left(\boldsymbol{\theta}^{t-\tau_i^t}\right) \mp \nabla f_i \left(\boldsymbol{\theta}^{t-\tau_i^t-1}\right) - \boldsymbol{c}_i^t \right\|^2 + \frac{S}{N} \left(\frac{2\sigma^2}{K} + \frac{2L^2}{K} \sum_k \mathbb{E} \left\| \boldsymbol{\theta}_i^{t,k} - \boldsymbol{\theta}^{t-\tau_i^t} \right\|^2\right) \\
&\leq \left(1 - \frac{S}{N}\right) \mathbb{E} \left[\left(1 + \frac{S}{2N}\right) \phi_i^{t-1} + \left(1 + \frac{2N}{S}\right) \gamma^2 L^2\right] + \frac{2S}{N} \left(\frac{\sigma^2}{K} + \frac{1}{3} \eta^2 K^2 L^2\right) \\
&\leq \left(1 - \frac{S}{2N}\right) \phi_i^{t-1} + \frac{2N}{S} \gamma^2 L^2 + \frac{2S}{N} \left(\frac{\sigma^2}{K} + \frac{1}{3} \eta^2 K^2 L^2\right) \\
&\leq \left(1 - \frac{S}{2N}\right)^t \phi_i^0 + \left(\frac{2N}{S} \gamma^2 L^2 + \frac{2S}{N} \left(\frac{\sigma^2}{K} + \frac{1}{3} \eta^2 K^2 L^2\right)\right) \sum_{\tau=0}^{t-1} \left(1 - \frac{S}{2N}\right)^\tau \\
&\leq \left(1 - \frac{S}{2N}\right)^t \phi_i^0 + 4 \left(\frac{N^2}{S^2} \gamma^2 L^2 + \frac{\sigma^2}{K} + \frac{1}{3} \eta^2 K^2 L^2\right).
\end{aligned}
$$

Since $\boldsymbol{c}_i^0 = \frac{1}{K} \sum_{k=0}^{K-1} \nabla F \left(\boldsymbol{\theta}^0; \boldsymbol{\xi}_i^{-1,k}\right)$, we have

$$
\begin{aligned}
\phi_i^0 &= \left(1 - \frac{S}{N}\right) \mathbb{E} \left\| \nabla f_i \left(\boldsymbol{\theta}^0\right) - \boldsymbol{c}_i^0 \right\|^2 + \frac{S}{N} \mathbb{E} \left\| \nabla f_i \left(\boldsymbol{\theta}^0\right) - \frac{1}{K} \sum_k \nabla F \left(\boldsymbol{\theta}_i^{0,k}; \boldsymbol{\xi}_i^{0,k}\right) \right\|^2 \\
&\leq \left(1 - \frac{S}{N}\right) \frac{\sigma^2}{K} + \frac{2S}{N} \left(L^2 \mathbb{E} \left\| \boldsymbol{\theta}^0 - \boldsymbol{\theta}_i^{0,k} \right\| + \frac{\sigma^2}{K}\right) \\
&\leq \left(1 + \frac{S}{N}\right) \frac{\sigma^2}{K} + \frac{2S}{3N} \eta^2 K^2 L^2 \\
&\leq \frac{2\sigma^2}{K} + \frac{2S}{3N} \eta^2 K^2 L^2.
\end{aligned}
$$

Then, we have

$$
\begin{aligned}
\phi_i^t &\leq \left(\frac{2\sigma^2}{K} + \frac{2S}{3N} \eta^2 K^2 L^2\right) \left(1 - \frac{S}{2N}\right)^t + 4 \left(\frac{N^2}{S^2} \gamma^2 L^2 + \frac{\sigma^2}{K} + \frac{1}{3} \eta^2 K^2 L^2\right) \\
&\leq \frac{6\sigma^2}{K} + 2 \eta^2 K^2 L^2 + \frac{4N^2}{S^2} \gamma^2 L^2, \forall i, t.
\end{aligned} \tag{22}
$$

Since $i \in \mathcal{S}_t$, we have $\boldsymbol{c}_i^{t+1} = \frac{1}{K} \sum_k \nabla F \left(\boldsymbol{\theta}_i^{t,k}; \boldsymbol{\xi}_i^{t,k}\right)$. Then,

$$
\begin{aligned}
&\mathbb{E} \left\| \boldsymbol{c}_i^{t+1} - \boldsymbol{c}_i^t \right\|^2 \\
&= \mathbb{E} \left\| \frac{1}{K} \sum_k \nabla F \left(\boldsymbol{\theta}_i^{t,k}; \boldsymbol{\xi}_i^{t,k}\right) \mp \nabla f_i \left(\boldsymbol{\theta}_i^{t,k}\right) \mp \nabla f_i \left(\boldsymbol{\theta}^{t-\tau_i^t}\right) \mp \nabla f_i \left(\boldsymbol{\theta}^{t-\tau_i^t-1}\right) - \boldsymbol{c}_i^t \right\|^2 \\
&\leq \frac{4\sigma^2}{K} + \frac{4L^2}{K} \sum_k \mathbb{E} \left\| \boldsymbol{\theta}_i^{t,k} - \boldsymbol{\theta}^{t-\tau_i^t} \right\|^2 + 4L^2 \mathbb{E} \left\| \boldsymbol{\theta}^{t-\tau_i^t} - \boldsymbol{\theta}^{t-\tau_i^t-1} \right\|^2 + 4\mathbb{E} \left\| \nabla f_i \left(\boldsymbol{\theta}^{t-\tau_i^t-1}\right) - \boldsymbol{c}_i^t \right\|^2 \\
&\leq \frac{4\sigma^2}{K} + \frac{4}{3} \eta^2 K^2 L^2 + 4\gamma^2 L^2 + 4\phi_i^{t-1}.
\end{aligned} \tag{23}
$$

Plugging (22) into (23), we have

$$
\mathbb{E} \left\| \frac{1}{K} \sum_k \nabla F \left(\boldsymbol{\theta}_i^{t,k}; \boldsymbol{\xi}_i^{t,k}\right) - \boldsymbol{c}_i^t \right\|^2 \leq \frac{28\sigma^2}{K} + \frac{28}{3} \eta^2 K^2 L^2 + 4\gamma^2 L^2 \left(1 + \frac{4N^2}{S^2}\right),
$$

which proves $i$). Moreover, based on the concavity of function $(\cdot)^{\frac{1}{2}}$, taking square root on the both sides of $i$) directly yields $ii$) that

$$\mathbb{E}\left\|\frac{1}{K}\sum_k \nabla F\left(\boldsymbol{\theta}_i^{t,k};\boldsymbol{\xi}_i^{t,k}\right) - \boldsymbol{c}_i^t\right\| \leq \left(\mathbb{E}\left\|\frac{1}{K}\sum_k \nabla F\left(\boldsymbol{\theta}_i^{t,k};\boldsymbol{\xi}_i^{t,k}\right) - \boldsymbol{c}_i^t\right\|^2\right)^{\frac{1}{2}}$$

$$\leq 2\sqrt{\frac{7}{K}}\sigma + 2\sqrt{\frac{7}{3}}\eta K L + 2\gamma L\left(1 + \frac{2N}{S}\right).$$

Further, we have

$$\mathbb{E}\left\|\nabla f\left(\boldsymbol{\theta}^t\right) - \boldsymbol{c}^{t+1}\right\| \leq \frac{1}{N}\sum_i \mathbb{E}\left\|\nabla f_i\left(\boldsymbol{\theta}^t\right) - \boldsymbol{c}_i^{t+1}\right\|$$

$$\leq \frac{2}{N}\sum_i \mathbb{E}\left\|\nabla f_i\left(\boldsymbol{\theta}^t\right) - \nabla f_i\left(\boldsymbol{\theta}^{t-\tau_i^t}\right)\right\| + \frac{2}{N}\sum_i \mathbb{E}\left\|\nabla f_i\left(\boldsymbol{\theta}^{t-\tau_i^t}\right) - \boldsymbol{c}_i^{t+1}\right\|$$

$$\leq \frac{2}{N}\sum_i \tau_i^t\gamma L + \frac{2}{N}\sum_i \sqrt{\phi_i^t}$$

$$\leq 2\sigma\sqrt{\frac{6}{K}} + 2\gamma L\left(\sum_i \frac{\tau_i^t}{N} + \frac{2N}{S}\right) + 2\sqrt{2}\eta K L.$$

$\square$

## E. Additional Numerical Results

### E.1. Simulation Setup

**Tasks and Networks.** We evaluate the performance of our algorithms on the image classification task with two real-world dateset: CIFAR-10 (Li et al., 2017) and FMNIST (Xiao et al., 2017). For FMNIST, we utilize a convolutional neural network (CNN) consisting of three convolutional layers and two fully connected layers. For CIFAR-10, we adopt a ResNet-18 architecture.

**Federated Setting.** Our experiments consider a federated learning system with $N = 100$ clients cooperating to train a shared model. At each global communication round, a fraction of $0.1$ clients are randomly selected to participate, resulting in $S = 10$ active clients per round. Each selected client trains locally for 2 epochs with a batch size of 100 using stochastic gradient descent (SGD) as the optimization algorithm. These settings are consistent across all experiments unless otherwise stated.

**Data Heterogeneity.** To evaluate the robustness of our algorithms under varying data distributions, we consider both i.i.d. and non-i.i.d. data settings. For the i.i.d. case, data samples are uniformly distributed across the clients. For the non-i.i.d. case, we simulate realistic data heterogeneity using a Dirichlet distribution $Dir(\alpha)$. Specifically, we set $\alpha = 0.5$, where smaller $\alpha$ values correspond to higher degrees of non-i.i.d. distribution.

**Asynchronous Setting.** We simulate asynchronous environments using FedBuff (Nguyen et al., 2022) as the baseline framework. The concurrency level is set to $M_c = 20$, meaning that the server can aggregate results from up to 20 clients concurrently. The delay time for each client is sampled from a uniform distribution $\mathcal{U}(0, T_{\max})$, where $T_{\max} = 20$ seconds by default. The delayed communication round $\tau$ correlates with the concurrency $M_c$. Specifically, when $M_c = 20$, the average delay $\tau_{\text{avg}} = 0.9184$, and the maximum delay $\tau_{\max} = 4$, as analyzed in CA$^2$FL. We run all experiments for a total of $T = 600$ global communication rounds.

**Baselines.** We compare our proposed algorithms against two state-of-the-art asynchronous federated learning baselines: CA$^2$FL (Wang et al., 2023) and FADAS (Wang et al., 2024c). For the main results, we adopt the hyperparameter settings recommended by their respective papers. In cases where these settings fail under challenging asynchronous conditions, we adjust them appropriately to ensure functionality.

### E.2. Additional Results

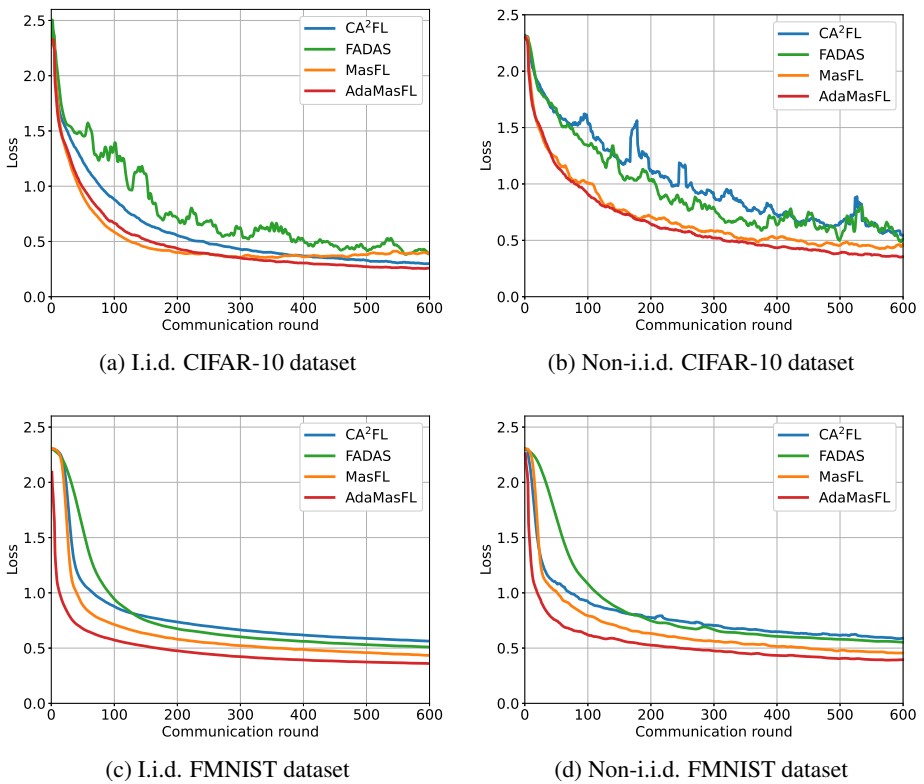

(a) I.i.d. CIFAR-10 dataset       (b) Non-i.i.d. CIFAR-10 dataset

(c) I.i.d. FMNIST dataset       (d) Non-i.i.d. FMNIST dataset

Figure 4: Test loss versus communication round on different datasets with i.i.d./non-i.i.d. data.

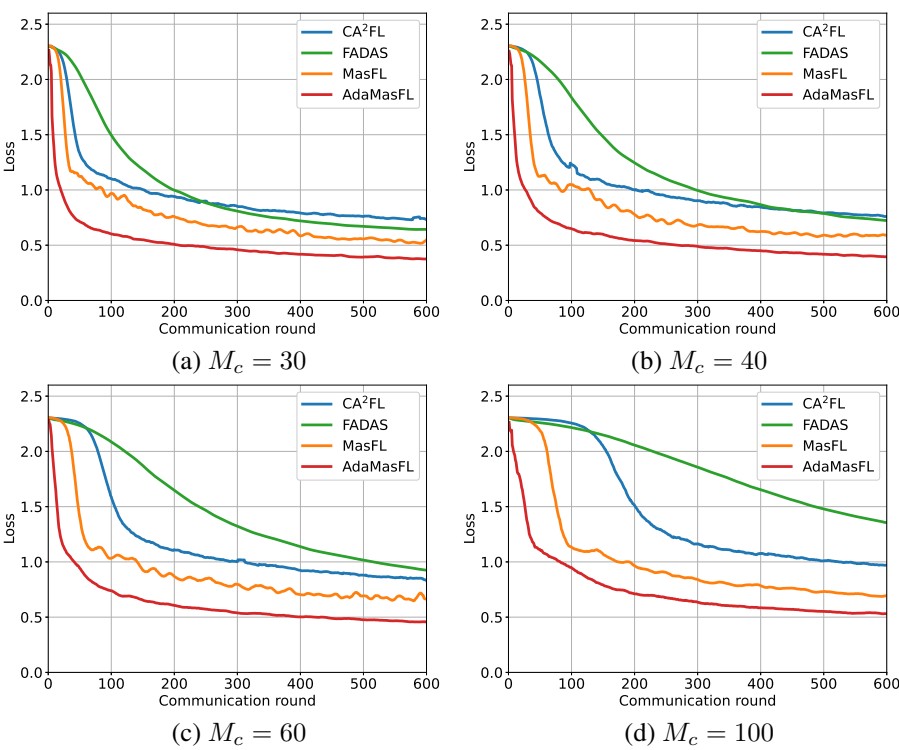

(a) $M_c = 30$       (b) $M_c = 40$

(c) $M_c = 60$       (d) $M_c = 100$

Figure 5: Test loss versus communication round on non-i.i.d. FMNIST dataset under varying levels of asynchrony.

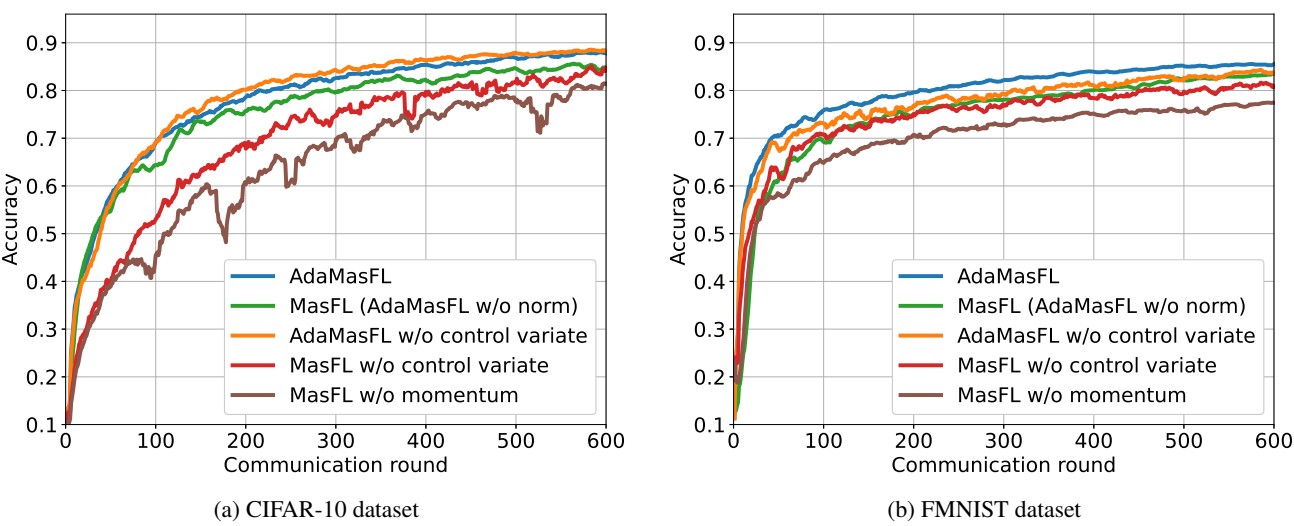

(a) CIFAR-10 dataset

(b) FMNIST dataset

Figure 6: Test accuracy versus communication round for ablation studies on non-i.i.d. CIFAR-10/FMNIST dataset.

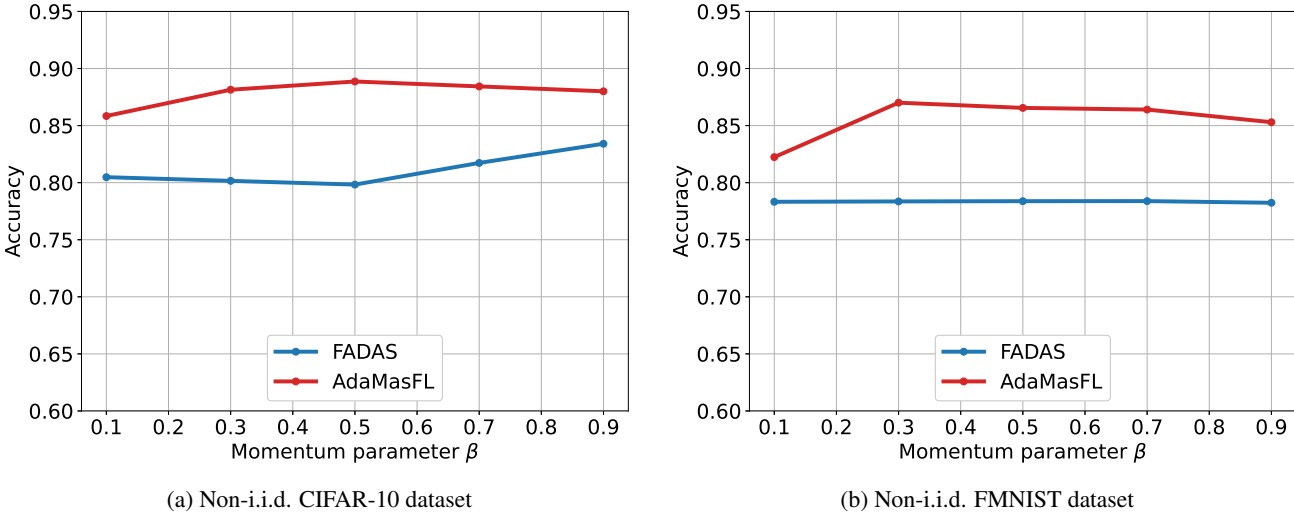

(a) Non-i.i.d. CIFAR-10 dataset

(b) Non-i.i.d. FMNIST dataset

Figure 7: Test accuracy compared with baselines using momentum under different momentum parameter $\beta$.

