# OpenReview forum: "Momentum-Driven Adaptivity: Towards Tuning-Free Asynchronous Federated Learning"
_ICML.cc/2025/Conference — ICML 2025 poster_

### Official Review · Reviewer_mYF9 · 2025-02-27

**Overall Recommendation:** 3

**Summary:**

This paper propose to adopt the server-client momentum with variance reduction technique in the Async FL framework. By using VR technology, the proof eliminates the heterogeneity assumption while maintaining the same efficiency as sync FL such as SCAFFOLD. The experiments validate the algorithm's efficiency in terms of test accuracy.

**Claims And Evidence:**

The theoretical proofs are complete, but the experiments are insufficient, lacking sensitivity studies on multiple variables and comprehensive ablation experiments. There is also no use of methods such as changing random seeds to verify robustness. For specific details, refer to 'Experimental Designs Or Analyses‘.

**Essential References Not Discussed:**

The discussion is sufficient.

**Experimental Designs Or Analyses:**

1. The experiments are partially tested on FMNIST and CIFAR10 on ResNet-18, with some improvements on the test accuracy, but there is a lack of systematic experiments to validate the findings. There is insufficient experimental validation for related variables, such as different local length $K$, data heterogeneity levels, and the duration of delay $\tau_i$. According to the description on page 29, most of the hyperparameters are selected as fixed values without additional sensitivity experiments, etc. I recommend that the authors conduct further extensive ablation experiments to determine the robustness of the proposed algorithm.

2. An experiment for a core variable is missing. One of the core contributions of this paper is the use of momentum-based weighting at both the server and client ends. Therefore, ablation experiments related to this scheme are essential. Necessary experiments, such as the benefits brought by only local momentum, global momentum, or variance reduction, should be included in the main text.

3. The experiments related to $\beta$ are missing. As an important hyperparameter, $\beta$ should be tested to determine the range in which it is effective. Although the theory sets it to be related to $T$, the practical role of $\beta$ in the experiments is still unclear.

4. The authors need to include a table to report the communication bits and wall-clock time of all methods in the baselines.

**Methods And Evaluation Criteria:**

1. The approach has some originality, but its novelty is not high. Both server-client momentum and variance reduction techniques like scaffold have been widely studied. The approach adopts in this paper combines these two methods within the asynchronous FL framework without making significant modifications.

2. AdaMasFL requires to communcate an extra variable $\Delta$, this will increase the communication overhead.

**Other Comments Or Suggestions:**

see above for more details

**Other Strengths And Weaknesses:**

Other strengths:

Table 1 presents the convergence complexity and detailed conditions, which is great as it allows one to easily see the source of theoretical progress and the additional assumptions. Although the choice of $\beta$ depends on $T$ rather than being a constant, which makes the theory somewhat thin, the corresponding conclusions are still very clear in the comparison shown in Table 1.

**Questions For Authors:**

Main issues seen above.

1. What's the meaning of variables in line 153, e.g. $\sigma_h$? I did not see the relevant definition.

2. Can $\beta$ be set as a constant? If so, what effect does it have?

**Relation To Broader Scientific Literature:**

It is helpful for the design of asynchronous federated learning frameworks.

**Theoretical Claims:**

1. In theorem 3.1, the selection of momentum coefficient $\beta=\sqrt{SK/T}$ is not a constant. This usually does not occur in previous algorithms, because the number of training rounds $T$ can approach infinity, which also means that $\beta$ will consequently approach 0. This condition is stronger than previous methods.

2. The description in line 271 is inaccurate. On the general assumptions, the original FedAvg cannot achieve the full acceleration bound under partial participation, as discussed in reference [1]. The full acceleration bound $O(1/\sqrt{SKT})$ depends on variance reduction or additional bounded gradient assumptions. The related description needs to be revised and further discussed.

[1] Achieving linear speedup with partial worker participation in non-iid federated learning

---

> ### Author Rebuttal · Authors · 2025-03-26
>
> **We sincerely appreciate the reviewer for recognizing our contributions and for the constructive comments. Our point-to-point responses to concerns on Weaknesses and Questions are given below.**
>
> **Reply to Methods And Evaluation Criteria:**
>
> 1. Thank you for your comment. Our work advances asynchronous federated learning (AFL) in the following key ways:
>
> - **Eliminating Data Heterogeneity Bounds in the Asynchronous Regime:** While server-client momentum has been explored in some prior AFL works, all existing approaches rely on data dissimilarity bounds to address non-iid data. In this work, we introduce a novel *two-level momentum* technique, making our approach the first to eliminate the need for such heterogeneity bounds in AFL. Although variance reduction technique has been explored in methods such as SCAFFOLD, they operate in synchronous frameworks, thus their analyses are inherently simpler compared to our asynchronous approach.
> - **Tuning-Free AFL Approach:** This paper also makes another significant contribution by introducing a completely tuning-free AFL approach, greatly simplifying algorithm deployment. As shown in Fig. 2, our method demonstrates exceptional robustness across varying levels of asynchrony. Based on the reviewer's suggestion, we have also included comparisons under different data heterogeneity levels and local training lengths ($K$). The results confirm that our method remains robust across diverse scenarios.
>
> These contributions go beyond combining existing techniques and addressing key challenges in AFL.
>
> 2. We agree with the reviewer that AdaMasFL requires the communication of an additional variable, $\Delta$, as the cost of achieving a tuning-free AFL. To address this communication overhead, information compression techniques can be incorporated as a potential solution. We acknowledge this as an important future work.
>
> **Reply to Theoretical Claims:**
>
> 1. Thanks for the comment. Using a $T$-dependent learning rate is standard in non-convex optimization to achieve theoretical guarantees. This predefined $T$ is reasonable since in real-world deployments, FL systems typically run for a pre-determined number of rounds due to resource constraints and practical considerations. The design also ensures momentum diminishes as $T$ increases, avoiding excessive staleness accumulation in asynchronous updates.
>
> 2. We sincerely thank the reviewer for pointing this out. We have removed the reference to the FedAvg algorithm in line 271.
>
> **Reply to Experimental Designs Or Analyses:**
> 1. Thanks for the comment. We have included sensitivity experiments with different local training lengths ($K$) as shown in https://github.com/anonymouslinkforrebuttal/for4415/blob/main/4-1-k.pdf, and data heterogeneity levels as shown in https://github.com/anonymouslinkforrebuttal/for4415/blob/main/4-1-noniid.pdf. Additionally, an ablation study on the impact of varying delay durations has already been presented in Fig. 2 of our main paper. Detailed information about the asynchronous setting can be found in our response to "Weakness 2" of reviewer JC6K. The results demonstrate the robustness of our approaches across various hyperparameter settings.
>
> 2. Thanks for your suggestion. We have included an ablation study analyzing the benefits of normalization, momentum, and control variate in our algorithms, shown as https://github.com/anonymouslinkforrebuttal/for4415/blob/main/4-2-ablation.pdf.
>
> 3. Thanks for the comment. We have included performance comparisons under different values of $\beta$ and compared with the momentum-based method FADAS as in https://github.com/anonymouslinkforrebuttal/for4415/blob/main/4-3-beta.pdf.
>
> 4. Thanks for the comment. We have added a table to report the wall-clock time of all baseline methods (in seconds) achieving a certain accuracy (CIFAR-10: 80%, FMNIST: 75%). Regarding information compression, as it goes beyond the scope of this work and we assume perfect communication in our paper, we leave this as an important direction for future research.
> |Dataset  |Heterogeneity|CA$^2$FL|FADAS|MasFL|AdaMasFL|
> |-|-|-|-|-|-|
> |CIFAR-10|iid|3485|4171|2033|2323|
> ||non-iid|12593|9372|6679|5333|
> |FMNIST|iid|1975|1238|832|394|
> ||non-iid|2311|1943|1067|514|
>
> **Reply to Questions For Authors:**
>
> 1. Thanks for pointing this out! $\sigma_h$ should be $\sigma_g$ defined in eq. (1), which characterizes the level of data heterogeneity.
> 2. For a predetermined $T$, $\beta = \sqrt{SK/T}$ is a constant in our methods. As this momentum coefficient $\beta$ balances historical and fresh updates, it plays a crucial role in mitigating staleness in AFL. A larger $T$ results in a smaller $\beta$, ensuring stability and convergence by reducing the influence of stale updates. This design enables linear speedup by effectively balancing momentum and new gradient updates, ensuring strong theoretical guarantees and scalability.
>
> **Thank you once again for your thoughtful review and constructive feedback.**

---

> > ### Comment · Reviewer_mYF9 · 2025-04-03
> >
> > I have read the author's rebuttal and most of the concerns have been addressed. I suggest authors
> >
> > 1) state a table to show the communication bits and calculations for the proposed method.
> >
> > 2) according to the experiments of $\beta$, it appears that choosing $\beta=0.2,0.3$ yields the best performance, which aligns with expectations. However, since $\beta$ is not a constant in the theoretical analysis, this conclusion appears quite weak (as a decayed $\beta$, like the learning rate, can artificially weaken the influence of certain constant upper bounds). The authors' claim that $\beta$ can be treated as a constant in practice after pre-selecting $T$ is incorrect, because $T$ is generally not considered a constant in optimization complexity analysis. The current version does not seem to resolve this issue.
> >
> > 3) In Table 4, besides reporting the overall speedup, the authors should also provide the computation time per round. While the total time speedup is meaningful, the number of communication rounds may vary depending on the dataset, and reporting wall-time per iteration would more accurately reflect computational efficiency. I understand that this might make the algorithm appear computationally expensive and inefficient, but it represents the true performance of the method.
> >
> > 4) The experiments on $\beta$ suggest that the use of momentum provides only a minor improvement, as the performance curves in the experiment are quite uniform (except for $\beta=0.1$, where the performance is significantly worse). I suggest that the authors include a table comparing the AdaMasFL method with the baseline FedAvg method, showing how much each module improves performance separately. This would help highlight the most critical components of the proposed approach. Currently, the experiments suggest that $\beta$ may not be very important. How much improvement does normalized SGD and the server-side operations contribute?
> >
> > I believe I have fully understand this paper, and I will adjust my score based on the answers to these unresolved issues.

---

> > > ### Author Response · Authors · 2025-04-04
> > >
> > > Thanks for the reply!  Our point-to-point responses are given below.
> > >
> > > 1. Thanks for the comment. Denote by $d$ the dimension of our training model $\theta$. At each global round $t$, our first algorithm, **MasFL**, requires uploading a vector $\mathbf{c}^t$ (of dimension $d$) from each client to the server and downloading two vectors ($\theta$ and $\beta\theta^t + (1-\beta)g^{t+1}$, both of dimension $d$) from the server to each client. For our second algorithm, **AdaMasFL**, an additional $d$-dimensional vector, $\frac{1}{\eta K}(\theta^{t-\tau_i^t} - \theta_t^{t,K})$, is required to upload, resulting in $2d$-dimensional information transformation for both uplink and downlink.
> > >
> > > We would like to clarify that this communication overhead is the same as that of SCAFFOLD-based algorithms, and our methods do not introduce any additional communication workload. For each real number, let $\delta$ represents the communication bits required to transmit each real number, the communication costs of our methods are as follows:
> > > |  MasFL | Uplink | Downlink |
> > > |-|-|-|
> > > |  MasFL| $d\delta$ | $2d\delta$ |
> > > | AdaMasFL$\quad$ | $2d\delta$ $\quad$| $2d\delta$ |
> > >
> > > In our experiments, the dimension of the training model for the CIFAR-10 dataset (ResNet 18) is $d=11689512$, and that for the FMNIST dataset (CNN) is $d=21840$. The calculation accuracy in of our experiment is $\delta=32$ bits.
> > >
> > > 2. Thanks for the comment. (1) In our work, the value of $\beta = \sqrt{SK/T}$ is preselected and remains constant for a given experiment. It is a fixed value and does not decay. Here, we would like to clarify that $T$ refers to the total number of algorithm iterations, not the variable $t$, which represents the current iteration.
> > >
> > > (2) Moreover, we would like to emphasize that this $T$-dependent step size setting is a common practice in nonconvex optimization, as seen in works such as [1] and [2].  For further details, please refer to the discussion on "SGD in nonconvex optimization" in the Related Work section of reference [2].
> > >
> > > (3) Most importantly, such constant step sizes are generally preferred in practice for nonconvex optimization, especialy in large-scale optimization tasks like deep learning.
> > >
> > > [1] Cheng, Ziheng, et al. "Momentum benefits non-iid federated learning simply and provably." arXiv preprint arXiv:2306.16504 (2023).
> > >
> > > [2] Yang, Junchi, et al. "Two sides of one coin: the limits of untuned sgd and the power of adaptive methods." Advances in Neural Information Processing Systems 36 (2023): 74257-74288.
> > >
> > > 3. Thank you for this suggestion. The per-iteration wall-times of our approaches and all baselines are provided below:
> > > |Dataset|CA$^2$FL|FADAS|MasFL|AdaMasFL|
> > > |-|-|-|-|-|
> > > |CIFAR-10|25.47|26.13|26.85|26.4|
> > > |FMNIST|5.47|6.61|6.36|6.35|
> > >
> > > From this table, the per-iteration wall-times of all algorithms are quite similar. This is because they are all based on the FedAvg framework and do not involve computationally expensive operations, such as high-dimensional matrix multiplications.
> > >
> > > However, we would like to note that the computation time is significantly influenced by the computational resources and hardware configurations of the machine used to run the algorithms. While this metric is informative, it may vary across different environments.
> > >
> > > 4. Thank for the comment. (1) According to the simulation results in https://github.com/anonymouslinkforrebuttal/for4415/blob/main/4-2-ablation.pdf, momentum significantly improves the performance of our first algorithm, MasFL, while contributing only marginally to the second algorithm, AdaMasFL. This may be attributed to the robustness provided by the adaptive step size design in AdaMasFL.
> > >
> > > (2) We would like to emphasize that this stability to varying $\beta$ is a key advantage of our algorithm, highlighting its robustness and tuning-free nature. Notably, across the entire range of $\beta$ values depicted, our algorithm consistently outperforms the state-of-the-art baseline (FADAS approach).
> > >
> > > (3) We agree with the reviewer that adding the performance comparison with the FedAvg algorithm would better showcase the performance improvement of our approach. We have added FedAvg in the ablation study of AdaMasFL: https://github.com/anonymouslinkforrebuttal/for4415/blob/main/24-fedavg.pdf. The corresponding test accuracies are listed below:
> > > |Method|CIFAR-10|FMNIST|
> > > |-|-|-|
> > > | FedAvg|0.837|0.819|
> > > | AdaMasFL|0.880|0.853|
> > > | AdaMasFL w/o norm|0.844|0.833|
> > > | AdaMasFL w/o control variate|0.885|0.835|
> > > | AdaMasFL w/o momentum|0.868|0.850|
> > >
> > > (4) Finally, we emphasize that the primary focus of this paper is to design a tuning-free algorithm to simplify the deployment of AFL, which has never been realized before. We have provided thorough theoretical analysis to validate our results. In our algorithm design, we incorporate momentum, normalized SGD, and control variates because they are all indispensable components to achieving this tuning-free objective, not out of the performance improvement purpose.

---

### Official Review · Reviewer_uM33 · 2025-03-14

**Overall Recommendation:** 3

**Summary:**

This paper studies asynchronous federated learning and proposes a novel momentum-driven asynchronous FL framework that eliminates the need for data heterogeneity bounds. The authors provide theoretical analysis and conduct experiments to verify the effectiveness of the proposed method.

**Claims And Evidence:**

I think the claims in the submission are mostly clear.

**Essential References Not Discussed:**

N/A

**Experimental Designs Or Analyses:**

1. How was the asynchronous delay simulated/achieved in the experiments?
2. What does $M_c$ represent in the experiments? Does $M_c$ control the asynchronous delay, and does it share a similar concept with  $S$ in your experiments?
3. I think providing more ablation studies on how the degree of asynchronous delay affects the proposed method would be helpful in verifying its effectiveness.

**Methods And Evaluation Criteria:**

1. I am wondering how asynchronicity is reflected in the proposed algorithm. In line 3 of Algorithm 1, the server randomly selects a set of clients  $S_t$ , and in line 7, the aggregation also happens within this set  $S_t$. This suggests that the clients in  $S_t$  participate in the update within the same round, which seems to contradict the asynchronous scheme where clients update at their own pace. Although Algorithm 2 includes notations related to delay  $\tau_i^t$, I believe this represents a semi-asynchronous approach. While clients may start from different global models, the server still waits for all aggregated information before proceeding.

2. Moreover, I am wondering whether the unbiasedness of the two-level momentum still holds when the correction term includes asynchronous delay (a similar issue arises regarding the sampling set $S_t$).

3. I think the proposed methods incur extra implicit memory on both the server and clients, i.e., the server need to maintain each clients' correction term $c$, and the client also need to maintain one copy of correct term $c$.

**Other Comments Or Suggestions:**

N/A

**Other Strengths And Weaknesses:**

Strengths:
1. The empirical results seem convincing.
2. The paper is well structured and easy to follow.

Most Weaknesses have been discussed in previous sections, particularly about the theoretical claims, the method, and the motivation.

**Questions For Authors:**

Please refer to previous sections. Addressing the issues in previous part would help me better understand and potentially change my evaluation of the paper.

**Relation To Broader Scientific Literature:**

This paper may contribute to the privacy-preserving related machine learning applications.

**Theoretical Claims:**

I have reviewed all the theoretical results in the main paper and briefly examined the theoretical results in the appendix. I have several questions regarding this:

1. Why is the left-hand side of Theorem 3.1 different from that of Theorem 4.1? What is the reason for using both $||\nabla f(\theta^t))||^2$ and $||\nabla f(\theta^t))||$ when presenting the convergence rate?

2. I think the constraint on the learning rate  $\eta$  in Theorem 3.1 is not convincing. When the delay is large, e.g., $\tau_{\max} \geq \sqrt{T}/\sqrt{SK}$, the numerator in the constraint on $\eta$, i.e., $T - 4SK\tau_{\max}^2 < 0$ becomes negative, making the square root term undefined. Moreover, other related theoretical analyses on asynchronous FL, such as CA2FL and FADAS, do not include such a "minus term" in their learning rate conditions.

---

> ### Author Rebuttal · Authors · 2025-03-26
>
> **We sincerely appreciate the reviewer for recognizing our contributions and for the constructive comments. Our point-to-point responses to concerns on Weaknesses and Questions are given below.**
>
> **Reply to Methods And Evaluation Criteria:**
>
> 1. The asynchronicity of our method is reflected in the following two aspects:
> - **Server-Side Asynchronicity:** In Line 4 of Algorithm 1, the server maintains a buffer to store the latest updates from each client. When a client $i$ is selected for global aggregation ($i\in\mathcal{S}_t$), the server updates its control variate using the buffered (outdated) update rather than waiting for the client’s most recent update. This ensures the server does not rely on synchronized updates, enabling asynchronous behavior.
> - **Client-Side Asynchronicity:** On the client side (Algorithm 2), each client performs local updates continuously and uploads its results after $K$ iterations, then immediately begins the next update. Clients operate independently, and the global round index corresponds to the round when the client finishes its update, including a delay ($\tau_i^t$). This delay is reflected in the momentum term ($g^{t-\tau_i^t}$), capturing the asynchronicity.
>
> **Semi-Asynchronous Behavior:** Although the server aggregates update from the selected set of clients ($i\in\mathcal{S}_t$), these updates are generated at different times due to the clients’ independent schedules, distinguishing our approach from fully synchronous methods. Such an asynchronous mechanism is adopted in almost all existing AFL work to our knowledge, such as CA2FL and FADAS.
>
> 2. (1) The unbiasedness property does not hold in our two-level momentum. However, we achieve state-of-the-art convergence for our algorithm by carefully controlling the bias throughout the iterations. (2) The unbiasedness holds for the participating client set $\mathcal{S}_t$ because we sample clients randomly and uniformly at each round.
>
> 3. We agree with the reviewer that our method requires extra memory to maintain the control variate $c$ (with the same dimensions as the learning model) on both the server and client sides. This memory overhead is inherent to all SCAFFOLD-based algorithms, including ours.
>
> **Reply to Theoretical Claims:**
>
> 1. The differences in the left-hand sides of Theorem 3.1 and Theorem 4.1 arise from the distinct proof techniques used in each case:
> - Theorem 3.1: This theorem adopts the widely used squared gradient norm metric, i.e., $||\nabla f(\theta^t)||^2$, which is standard in analyzing convergence rates for optimization algorithms.
> - Theorem 4.1: To achieve problem-parameter-free convergence guarantees, we employ a novel proof technique that directly analyzes the gradient norm, i.e., $||\nabla f(\theta^t)||$. This direct analysis is essential for establishing problem-parameter-free results and requires a different approach compared to Theorem 3.1.
>
> 2. We agree with the reviewer that Theorem 3.1 suggests a sufficiently large $T$ such that $T \geq \Omega(\tau_{\max}^2)$. However, we would like to emphasize that imposing a lower bound on the number of iterations is a common practice in the literature. For example, the convergence bound of asynchronous SGD [1, Corollary 4] is established under a similar condition, $T \geq \Omega(\tau_{\max}^2)$. For AFL methods such as CA$^2$FL and FADAS, their asymptotic convergence rates of $\mathcal{O}\left(\frac{1}{\sqrt{T}}\right)$ are achieved when $T \geq \Omega(\tau_{\max}^2\tau_{\text{avg}}^2)$. Most importantly, we highlight that, unlike CA$^2$FL and FADAS, the unique contribution of Theorem 3.1 is that it does not impose any restrictions on data heterogeneity.
>
> [1] Lian, Xiangru, et al. *"Asynchronous parallel stochastic gradient for nonconvex optimization."* NIPS 2015.
>
> **Reply to Experimental Designs Or Analyses:**
>
> 1. Our implementation simulates practical asynchronous conditions using FedBuff's delay mechanism to account for varying client arrival times. Specifically:
> - At any given time, there are $M_c$ clients perform local updates concurrently;
> - Each client's execution time is sampled from a uniform distribution;
> - These varying execution times naturally create different delays in global aggregation participation;
> - Global aggregation occurs simultaneously with local updates.
>
>     This design captures the realistic scenario of heterogeneous client completion times and asynchronous aggregation.
>
> 2. $M_c$ represents the number of clients performing local updates concurrently throughout the algorithm's runtime. Since the overall client participation rate remains fixed, a greater $M_c$ leads to larger system delays, and vice versa.
>
> 3. Thank you for this suggestion. An ablation study on the performance of our methods and all baselines under different asynchronous levels has already been presented in Fig. 2 of our original paper.
>
> **Thank you once again for your thoughtful review and constructive feedback.**

---

> > ### Comment · Reviewer_uM33 · 2025-04-07
> >
> > Thank you to the authors for the rebuttal. Sorry for my late reply, as I originally replied to the “Official Comment” button.
> >
> > However, I still find somewhere a bit unclear to me.
> >
> > First, about the asynchronicity,  as I point out in the review, "In line 3 of Algorithm 1, the server randomly selects a set of clients $S_t$ , and in line 7, the aggregation also happens within this $S_t$." There is no evidence showing that "these updates are generated at different times due to the clients’ independent schedules". According to the algorithm, **the clients selected at round $t$ are also the ones whose updates are aggregated at that same round**. I think this is not a fully asynchronous setting.
> >
> > About the learning rate conditions. I think the requirement $T\geq \Omega(\tau_{\max}^2)$ for achieving the desired convergence rate and the condition of $T-4SK\tau_{\max}^2 \geq 0$ should not be treated as equivalent, even though they share a similar direction, i.e., $T$ should be sufficiently large relative to $(\tau_{\max}^2)$. To me, it’s not clear that these conditions stem from the same underlying aspect, so I hope the authors can provide further clarification on this point.

---

> > > ### Author Response · Authors · 2025-04-08
> > >
> > > Thanks for the reply!
> > >
> > > 1. **Asynchronous Setting:** Thank you for the comment. In our algorithm, as described in Section 3.1 and Algorithm 1, the server maintains a buffer to store updates from clients that arrive asynchronously throughout the runtime. This means that at each global aggregation step, the updates used for aggregation from the selected client set $S_t$  are drawn from the buffer, not from their most recent updates. Instead, the updates could have been generated at different times due to the independent schedules of the clients. This is a key point that ensures the algorithm operates in an asynchronous manner. We follows the widely adopted asynchronous setting from the FedBuff [1], which captures practical scenarios with varying client availability and update arrival times.
> > >
> > > [1] Nguyen, J., Malik, K., Zhan, H., Yousefpour, A., Rabbat, M., Malek, M., and Huba, D. Federated learning with buffered asynchronous aggregation. In Proceedings of the 25th International Conference on Artificial Intelligence and Statistics, pp. 3581–3607. PMLR, 2022.
> > >
> > > 2. Thank you for the insightful comment. We agree with the reviewer that the requirement $T \geq \Omega(\tau_\max^2)$ and the condition $T - 4SK\tau_\max^2 \geq 0$ are not equivalent. Our Theorem 3 specifically relies on the condition $T \geq \Omega(\tau_\max^2)$, and we will clarify this distinction in the revised version of our paper.
> > >
> > > We would also like to emphasize that the requirement for a sufficiently large $T$ is a common condition for ensuring theoretical guarantees in asynchronous learning frameworks, as seen in works such as CA$^2$FL, FADAS, and [2]. Compared to these references, our requirement on $T$ is not particularly restrictive.
> > >
> > >
> > > [2] Lian, Xiangru, et al. "Asynchronous parallel stochastic gradient for nonconvex optimization." NIPS 2015.
> > >
> > > **Thank you again for your valuable feedback!**

---

### Official Review · Reviewer_jC6k · 2025-03-17

**Overall Recommendation:** 3

**Summary:**

This paper addressed the data heterogeneity and staleness in asynchronous federated learning (AsycnFL). The authors propose MasFL that introduces the control variates into AsyncFL to stabilize the model updates during local training and global aggregation. Further, they normalize the momentum-averaged gradients in local model training to eliminate the need for trial-and-error tuning or problem-parameter, e.g., the number of participating clients $S$, local update iterations $K$, and communication rounds $T$. The theoretical analysis is thorough, and the experimental results are promising, showing improvements over the state-of-the-art methods.

**Claims And Evidence:**

Yes

**Essential References Not Discussed:**

One highly-related work [a] was missing.

[a] Cheng Z, Huang X, Wu P, et al. Momentum benefits non-iid federated learning simply and provably[J]. arXiv preprint arXiv:2306.16504, 2023.

**Experimental Designs Or Analyses:**

Yes

**Methods And Evaluation Criteria:**

Yes

**Other Comments Or Suggestions:**

NA

**Other Strengths And Weaknesses:**

**Strength**
- **High Writing Quality**. The manuscript is well-organized and easy to read.
- **Technical Advantage**. The authors first introduce the control variates into AsyncFL to tackle the challenges of data heterogeneity, gradient staleness,  and complex hyperparameter tuning. Compared with another similar technique Scaffold, the authors additionally consider the global momentum to stabilize the model updates during model aggregation, improving the FL system's robustness to the stale clients in the context of asynchronous aggregation. Both theoretical and experimental results confirm the effectiveness of this strategy.
- **Theoritical Contribution**. The paper provides strong theoretical guarantees for both MasFL and AdaMasFL. The convergence analysis is detailed, and the authors successfully prove that their methods achieve state-of-the-art convergence rates with linear speedup concerning the number of participating clients and local updates.

**Weakness**
- **Overstatement**. The claim that AdaMasFL "completely eliminates the need for tuning-free convergence" might be overstated. While the theoretical framework provides explicit hyperparameter settings, practical implementation might still require adjustments based on system-specific factors.  In addition, the claim "tuning-free AFL approach" also suffers the risk of overclaiming since tuning the learning rate led to different empirical results as evidenced by the experiments. The authors should temper this claim and discuss the practical implications of their theoretical results.
- **Issues on Experiment Settings**. The authors conducted experiments by directly assuming the degree of round-wise delaying of clients, which eliminated the necessity to consider the aggregation addition of clients (e.g., every $K$ arrivals in FedBuff). This issue also prevents a more practical setting that uses a virtual clock [b] to conduct asynchronous experiments. It's recommended to clarify the practical aggregation conditions of the proposed methods.
- **Missing Related Work**. One important related work was missing. Although this work conducted the convergence of synchronous FL, **they are the first to remove the dependency on the assumption of bounded data heterogeneity (i.e., bound gradient dissimilarity)** and also confirm the impact of control variate in Scaffold. Since the main theoretical contribution of this work lies in removing the dependency of convergence on bounded data heterogeneity assumption and uses similar control variates, the authors should discuss their contributions and the relationship of this work to it.


[a] Cheng Z, Huang X, Wu P, et al. Momentum benefits non-iid federated learning simply and provably[J]. arXiv preprint arXiv:2306.16504, 2023.

[b] Lai F, Dai Y, Singapuram S, et al. Fedscale: Benchmarking model and system performance of federated learning at scale[C]//International conference on machine learning. PMLR, 2022: 11814-11827.

**Questions For Authors:**

Please see the weakness.

**Relation To Broader Scientific Literature:**

NA

**Theoretical Claims:**

Yes

---

> ### Author Rebuttal · Authors · 2025-03-26
>
> **We sincerely appreciate the reviewer for recognizing our contributions and for the constructive comments. Our point-to-point responses to concerns on Weaknesses and Questions are given below.**
>
> **Reply to Essential References Not Discussed:**  We sincerely thank the reviewer for bringing this conference to our attention. We have added a detailed comparison of our work with [a] in the revised manuscript, as cited below.
> > In the synchronous setting, [a] showed that momentum helps remove data heterogeneity bounds (i.e., bound gradient dissimilarity) for nonconvex FL, marking a first in the literature.
>
> However, integrating momentum into AFL introduces substantial challenges due to fundamental conflicts between momentum's historical gradient accumulation and the asynchronous nature of updates. In asynchronous settings, clients' delayed updates compromise the accuracy of momentum calculations, as stale gradients introduce biases into the optimization trajectory, as shown in [1]. To address this, we propose a two-level momentum mechanism: the server updates based on the latest global momentum, while each client performs local updates using an outdated global momentum. This inconsistency, while necessary to control bias, presents significant challenges in the theoretical analysis of our approach.
>
> [1] Yu, T., Song, C., Wang, J., and Chitnis, M. Momentum approximation in asynchronous private federated learning. arXiv preprint arXiv:2402.09247, 2024.
>
> **Reply to Weekness 1:** Thank you for pointing this out. We agree with the reviewer that different learning rates can influence the performance of our algorithm. However, we would like to clarify that the optimal learning rates for our algorithm can be directly calculated based solely on system-defined constants: the number of participating clients ($S$), local update iterations ($K$), and communication rounds ($T$). These are configuration parameters that are predetermined by the system administrator, rather than problem-specific factors that require estimation or tuning.
>
> That said, we acknowledge the reviewer's suggestion to adopt a more tempered statement. In response, we have revised the manuscript to replace "tuning-free AFL approach" with "problem-parameter-free" or "simplify the algorithm tuning process" throughout, which we believe better reflects the practical implications of our work.
>
>
> **Reply to Weekness 2:** Thank you for this important observation. We apologize for any confusion caused by not including the detailed experimental setup in the main paper. Due to space constraints, we placed the complete asynchronous implementation details in the supplementary material (page 29).
>
> We would like to clarify that our implementation does simulate practical asynchronous conditions using FedBuff's delay mechanism, which accounts for varying client arrival times rather than assuming fixed delay rounds. Specifically:
>
> - At any given time, there are a total of $M_c$ clients perform local updates concurrently;
> - Each client's execution time is sampled from a uniform distribution;
> - These varying execution times naturally create different delays in global aggregation participation;
> - Global aggregation occurs simultaneously with local updates.
>
> This design aligns with the virtual clock setting proposed in [b] to captures the realistic scenario of heterogeneous client completion times and asynchronous aggregation.
>
> We appreciate this feedback and will incorporate these important implementation details into the main paper for better clarity.
>
> **Reply to Weekness 3:** Thanks for pointing this out. The reference [a] has been properly cited in the revised manuscript, please refer to our response under "Reply to Essential References Not Discussed."
>
>
> **Thank you once again for your thoughtful review and constructive feedback.**

---

### Official Review · Reviewer_3Kyd · 2025-03-21

**Overall Recommendation:** 4

**Summary:**

Previous works on asynchronous federated learning put strong assumptions (like bounded gradient assumption) in order to get theoretical guarantees. However, these assumptions are usually not realistic. In this paper, the authors propose a new asynchronous FL algorithm which novelly combines with global and local momentum. Due to the use of momentum, the authors remove the needs of strong assumptions in convergence analysis. Furthermore, they also design an adaptive version, which allows auto learning rate tuning. Experiments validate the effectiveness of proposed algorithms.

**Claims And Evidence:**

The authors claimed that they propose two novel training approaches MasFL and AdaMasFL. These algorithms incorporates two-level momentum. They provide not only theoretical guarantees with weaker assumptions but also experimental results to show advantages. Overall I think these claims are solid and well supported.

**Essential References Not Discussed:**

No.

**Experimental Designs Or Analyses:**

Yes, the designs and analyses look good.

**Methods And Evaluation Criteria:**

The authors provide a standard convergence analysis to analyze their proposed method.

The benchmark datasets in experiments are cifar and fmnist, which are relatively small.

**Other Comments Or Suggestions:**

In the paper "Yu et al. Momentum approximation in asynchronous private federated learning." they also propose a method to improve async FL. The authors should also compare experiments with this paper.

**Other Strengths And Weaknesses:**

NA

**Questions For Authors:**

NA

**Relation To Broader Scientific Literature:**

This paper may make asynchronous FL algorithm to be more practical and powerful. The analysis technique of two-level momentum may be also interesting to some readers.

**Theoretical Claims:**

I didn't check full details.

---

> ### Author Rebuttal · Authors · 2025-03-26
>
> **We sincerely appreciate the reviewer for recognizing our contributions and for the constructive comments.**
>
> **Response to Comments Or Suggestions:**
>
> Thanks for the suggestion. We have already discussed the reference Yu et al. (2024) in the Related Work section of our original manuscript (see Line 95), as cited below:
>
> > Yu et al. (2024)  identified that asynchrony introduces implicit bias in momentum updates and proposed momentum approximation for AFL, which optimally weights historical model updates to approximate synchronous momentum behavior.
>
> In response to the reviewer's suggestion, we will include experimental comparisons of our work with [1] in the revised version.
>
>
>
> **Thank you once again for your thoughtful review and constructive feedback.**

---

### Decision · Program_Chairs · 2025-05-01

**Decision:**

Accept (poster)

**Comment:**

The paper proposes a novel algorithm for federated learning with asynchronous adaptive updates. The algorithm integrates local and global momentum, along with control variates, giving convergence results in the asynchronous setting that do not rely on data heterogeneity-related bounds. After rebuttal, all the reviewers are on the positive side.

A weakness of this work is that it appears to be a combination of multiple known techniques. It would be helpful if the final paper can highlight the novel and non-obvious technical and mathematical steps that are key to obtaining the solution.